



# Long-term variability of drought indices in the Czech Lands and effects of external forcings and large-scale climate variability modes

**Jiří Mikšovský**[1,4], **Rudolf Brázdil**[2,4], **Miroslav Trnka**[3,4], **and Petr Pišoft**[1]

[1]Department of Atmospheric Physics, Charles University, Prague, Czech Republic
[2]Institute of Geography, Masaryk University, Brno, Czech Republic
[3]Institute of Agrosystems and Bioclimatology, Mendel University, Brno, Czech Republic
[4]Global Change Research Institute, Czech Academy of Sciences, Brno, Czech Republic

**Correspondence:** Jiří Mikšovský (jiri@miksovsky.info)

**Abstract.** While a considerable number of records document the temporal variability of droughts for central Europe, the understanding of its underlying causes remains limited. In this contribution, time series of three drought indices (Standardized Precipitation Index – SPI; Standardized Precipitation Evapotranspiration Index – SPEI; Palmer Drought Severity Index – PDSI) are analyzed with regard to mid- to long-term drought variability in the Czech Lands and its potential links to external forcings and internal climate variability modes over the 1501–2006 period. Employing instrumental and proxy-based data characterizing the external climate forcings (solar and volcanic activity, greenhouse gases) in parallel with series representing the activity of selected climate variability modes (El Niño–Southern Oscillation – ENSO; Atlantic Multidecadal Oscillation – AMO; Pacific Decadal Oscillation – PDO; North Atlantic Oscillation – NAO), regression and wavelet analyses were deployed to identify and quantify the temporal variability patterns of drought indices and similarity between individual signals. Aside from a strong connection to the NAO, temperatures in the AMO and (particularly) PDO regions were disclosed as one of the possible drivers of inter-decadal variability in the Czech drought regime. Colder and wetter episodes were found to coincide with increased volcanic activity, especially in summer, while no clear signature of solar activity was found. In addition to identification of the links themselves, their temporal stability and structure of their shared periodicities were investigated. The oscillations at periods of approximately 60–100 years were found to be potentially relevant in establishing the teleconnections affecting the long-term variability of central European droughts.

## 1 Introduction

Droughts, among the most prominent manifestations of extreme weather and climate anomalies, are not only of great climatological interest but also constitute an essential factor to be considered in the assessment of the impacts of climate change (Stocker et al., 2013; Trnka et al., 2018; Wilhite and Pulwarty, 2018). This is also valid for the territory of the Czech Republic where droughts, apart from floods, embody the most important natural disasters, with significant impacts upon various sectors of the national economy, such as agriculture, forestry, water management, and tourism and recreation. Since the Czech Republic lies on a continental divide with rivers flowing out of its territory, it depends on atmospheric precipitation alone for its water supply. Although certain extreme droughts with important socioeconomic and political impacts are known from the past, such as the drought of 1947 (Brázdil et al., 2016b), studies performed in recent years show that the Czech climate has become increasingly dry in the past 2–3 decades, expressed in terms of a higher frequency of extreme droughts with significant consequences (e.g., Brázdil et al., 2015b; Zahradníček et al., 2015). The abundance of long-term instrumental meteorological observations has provided a basis for a number of recent drought-focused studies, revealing complex regional

drought patterns and a richness of features observed at various spatial and temporal scales, in the European area (e.g., van der Schrier, 2006, 2007; Brázdil et al., 2009; Briffa et al., 2009; Dubrovský et al., 2009; Sousa et al., 2011; Spinoni et al., 2015) as well as other regions of the world (e.g., Dai, 2011; Spinoni et al., 2014; Ryne and Forest, 2016; Wilhite and Pulwarty, 2018). Along with more rapid variations, these also include long-term variability, such as a distinct trend towards drier conditions, prominent especially during the late 20th and early 21st centuries (e.g., Trnka et al., 2009a; Brázdil et al., 2015b).

In addition to a substantial number of studies investigating drought indices for the instrumental period in Europe (e.g., van der Schrier et al., 2007, 2013; Briffa et al., 2009; Sousa et al., 2011; Todd et al., 2013; Spinoni et al., 2015; Haslinger and Blöschl, 2017) and other areas of the world (e.g., Dai, 2011; Spinoni et al., 2014; Ryne and Forest, 2016; Wilhite and Pulwarty, 2018), generally calculated from measured precipitation totals and temperatures, considerable attention has been devoted to pre-instrumental drought reconstructions. The longest high-resolution drought series are typically based on various tree-ring series, usually reconstructing drought indices (mainly Palmer Drought Severity Index – PDSI) for summer (JJA) or other combinations of months during the growing season (e.g., Büntgen et al., 2010a, b, 2011a, b; Cook et al., 2015; Dobrovolný et al., 2015). Natural proxy data (see PAGES Hydro2k Consortium, 2017) may be supplemented by the documentary records generally utilized in historical climatology (Brázdil et al., 2005, 2010) in drought reconstructions. These are usually represented as series for drought frequency covering the last few centuries, usually from the 16th century to the present time or a shorter time period (e.g., Piervitali and Colacino, 2001; Domínguez-Castro et al., 2008, 2012; Diodato and Bellocchi, 2011; Brázdil et al., 2013; Noone et al., 2017). However, reconstructions of long-term series of drought indices from documentary and instrumental data, as have been done for the Czech Lands from the 16th century (Brázdil et al., 2016a; Možný et al., 2016), still remain the exception.

Although the series above permit the study of drought variability at various temporal and spatial scales, only a few researchers have attempted to link such fluctuations with the effects of climate forcings and large-scale internal variability modes, usually within the instrumental period. Pongrácz et al. (2003) applied a fuzzy-rule-based technique to the analysis of droughts in Hungary. Hess–Brezowsky circulation types and El Niño–Southern Oscillation (ENSO) events were used and their influence on drought occurrence (monthly PDSI) documented. Using weekly Z-index and PDSI, Trnka et al. (2009b) showed that the increase in drought frequency toward the end of the 20th century during April–June period is linked to increased occurrence of Hess–Brezowsky circulation types that are conducive to drought conditions over central Europe. Brázdil et al. (2015a) used regression analysis to investigate the effects of various external forcings

and large-scale climate variability modes in series of drought indices in the Czech Lands during the 1805–2012 period. The authors demonstrated the importance of the North Atlantic Oscillation phase and of the aggregate effect of anthropogenic forcings. Baek et al. (2017) applied correlation analysis to investigate teleconnection patterns between PDSI in multiple regions across the Northern Hemisphere and the activity of several climate variability modes, with noteworthy responses of European droughts indicated especially for North Atlantic Oscillation (NAO), ENSO and Pacific Decadal Oscillation (PDO). Other examples include attribution analyses for the climatic variables in Croatia (Bice et al., 2012) and for temperature and precipitation instrumental series in the Czech Lands (Mikšovský et al., 2014). More recent papers addressing the influence of certain forcing factors on individual climate variables may be added to this overview (e.g., Anet et al., 2014; Gudmundsson and Seneviratne, 2016; Schwander et al., 2017). Even so, the exact causes of the variability detected in drought data remain only incompletely known, especially regarding variations at decadal and multi-decadal timescales.

The current paper focuses on the identification and quantitative attribution of drought variability expressed by series of three drought indices in the Czech Lands (modern Czech Republic) throughout the past 5 centuries (1501–2006). In addition to an analysis of potential drought-relevant links in the climate system, attention is paid to their temporal stability and (mis)match of results based on climate reconstruction data from different sources. Regression and wavelet analysis are employed (see Sect. 3) to identify links between series of the three drought indices (supplemented by corresponding temperature and precipitation series) and the activity of external climate forcings or internal climate variability modes (see Sect. 2). The results of these analyses are presented in Sect. 4 and discussed with respect to the effects and variability patterns of individual explanatory factors and their interaction in Sect. 5. The last section then delivers a number of concluding remarks. Additional materials are presented in the Supplement.

## 2  Data

### 2.1  Drought indices

Various drought indices are used to characterize the spatiotemporal variability of droughts (see, e.g., Heim, 2000). To capture the temporal patterns of historical Czech drought regime, three country-wide drought indices were employed, each of them embodying a different strategy for defining dry or wet conditions (Fig. 1):

   i. Standardized Precipitation Index (SPI; McKee et al., 1993), calculated as the standardized deviation of precipitation totals over a chosen time window from their long-term means. SPI is a purely precipitation-based drought descriptor that takes no account of the

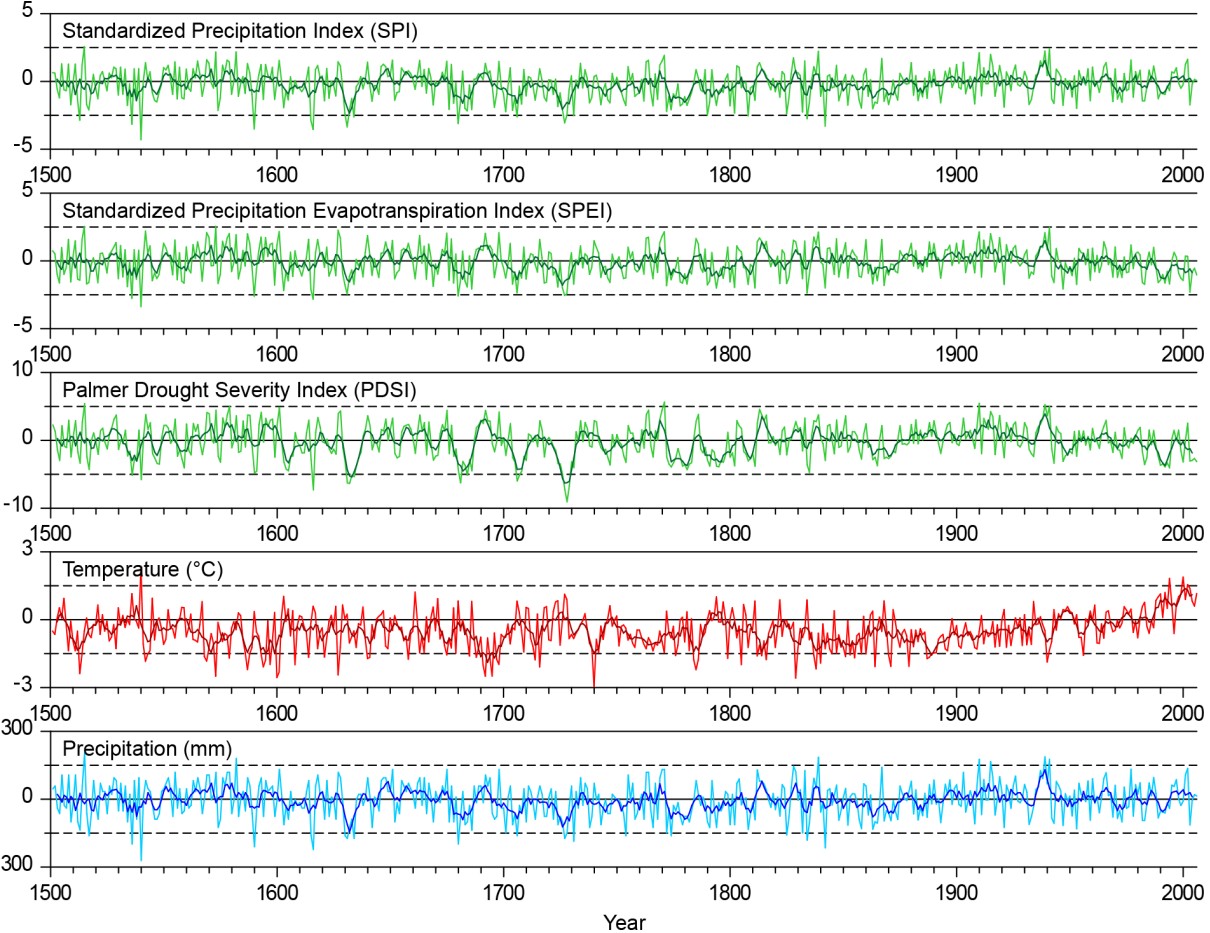

**Figure 1.** Annual series of drought indices, temperature and precipitation anomalies (with respect to 1961–1990 reference period) for the Czech Lands in the 1501–2006 period, also smoothed by 11-year running means (darker lines).

direct influence of temperature. As such, it is primarily representative of the factors altering precipitation in the target area.

ii. Standardized Precipitation Evapotranspiration Index (SPEI; Vicente-Serrano et al., 2010) is an index similar to SPI, but it considers potential evapotranspiration rather than precipitation alone, hence also reflecting temperature-related climate variations.

iii. Palmer Drought Severity Index (PDSI; Palmer, 1965) describes long-term soil moisture status. A self-calibrated version of PDSI was used in this contribution (Wells et al., 2004). Unlike SPI and SPEI, which are calculated from contemporaneous values of precipitation and temperature, PDSI also considers past drought status and effectively also storage capacity of the soil, thereby providing a better reflection of long-term drought behavior.

Long-term seasonal and annual series of these three indices, dating from 1501 CE in the Czech Lands (Brázdil et

al., 2016a), were used in the current study. They were derived from 500-year temperature and precipitation reconstructions based on a combination of documentary data and instrumental measurements. Documentary data comprised descriptions of weather and related phenomena from a variety of documentary evidence, some of it individual, some of it of an institutional character, such as annals, chronicles and memoirs, weather diaries (non-instrumental observations), financial and economic accounts, religious records, newspapers and journals, epigraphic sources, and more. Such data in the Czech Lands cover particularly, at varying degrees of density, the period from 1501 CE to the mid-19th century but continue even to the present time. The spatial density of such data changes over time, depending on the availability and extraction of existing documentary sources. All the data collected were critically evaluated with respect to possible errors in dating or spatial attribution and were used for interpretation of monthly weighted temperature and precipitation indices on a 7° scale, from which series of seasonal and annual indices were created (for more details see Brázdil et al., 2005, 2010). These data were further used as a

basic tool for temperature and precipitation reconstructions. First, Dobrovolný et al. (2010) reconstructed monthly, seasonal and annual central European temperature series, partly based on temperature indices derived from documentary data for Germany, Switzerland and the Czech Lands in the 1501–1854 period and partly on homogenized instrumental temperature series from 11 meteorological stations in central Europe (Germany, Austria, Switzerland, Bohemia) from 1760 onwards. Subsequently, seasonal and annual precipitation series for the Czech Lands were reconstructed from documentary-based precipitation indices in the 1501–1854 period and from mean precipitation series calculated from measured precipitation totals in the Czech Lands after 1804 (Dobrovolný et al., 2015).

Note that only seasonal and annual series of Czech SPI, SPEI and PDSI (Brázdil et al., 2016a), central European temperature series (Dobrovolný et al., 2010) and Czech precipitation series (Dobrovolný et al., 2015) were employed for further analysis in the current paper. Note also that despite the profound similarity in temporal variations in individual drought indices, manifested in their strong mutual correlation (Pearson correlation coefficient $r = 0.93$ for the annual values of SPI-SPEI, $r = 0.83$ for SPI-PDSI, $r = 0.88$ for SPEI-PDSI over the 1501–2006 period), each of them emphasizes different aspects of meteorological droughts, be they simple precipitation sums captured by SPI, evapotranspiration variations considered by SPEI or a longer-term soil moisture status embodied by PDSI. Hereinafter, results for multiple indices are therefore presented and discussed, with regard to their common features as well as mutual contrasts.

## 2.2 Explanatory variables

Due to the multitude of climate-defining influences and the complexity of their interactions, an essential part of statistical attribution analysis consists in the selection of the most relevant explanatory factors and the identification of the most appropriate quantifiable descriptors of their activity. For an analysis involving data from the pre-instrumental period, this task is further complicated by the limited amount of data suitable for direct quantitative analysis. Even so, reconstructions exist for most of the key climate drivers, be they external forcings or major modes of internal variability. In this analysis, several of these data sources were considered; brief descriptions of them appear below, while visualization of their fluctuations is provided in Figs. 2 (external forcings) and 3 (internal climate variability modes).

Of the external factors shaping the long-term climate evolution, a key role is played by the effects modifying radiative balance through changes in atmospheric composition. A large part of the observed changes may be attributed to variations in the concentrations of long-lived greenhouse gases (GHGs): carbon dioxide ($CO_2$) in particular, but also methane ($CH_4$) and nitrous oxide ($N_2O$) (Stocker et al., 2013). Since the past GHG concentrations are accessible to relatively accurate reconstruction using ice cores, their combined radiative forcing was considered as a potential formal descriptor of the long-term trends in the drought series. The time series of annual GHG forcing by Meinshausen et al. (2011) was used for the period since 1765 CE, and extended back to 1501 CE using the $CO_2$, $CH_4$ and $N_2O$ concentrations obtained from the online database of the Institute for Atmospheric and Climate Science, ETH Zurich, and approximate formulas provided in IPCC (2001, Table 6.2).

While variations in solar activity typically leave no or just a weak imprint on lower tropospheric observational time series during the instrumental era (e.g., Benestad, 2003; Gray et al., 2010; Brönnimann, 2015), their effects may become more noticeable over longer analysis periods, with major events such as the Maunder Minimum coming into play (e.g., Lohmann et al., 2004). In this contribution, a reconstruction of annual mean total solar irradiance (TSI) by Lean (2018) was used as the primary descriptor of solar activity. With data available from 850 CE onwards, the TSI values were used for the period 1501–2006 herein. An alternative TSI dataset by Coddington et al. (2016) was also employed for the 1610–2006 period.

Unlike variations in solar activity or concentrations of GHGs, the effects of major volcanic eruptions tend to be rather episodic, manifesting themselves in the lower troposphere as temporary global temperature drops (e.g., Canty et al., 2013), triggering summer cooling over Europe and winter warming over northern Europe (Fischer et al., 2007) but exhibiting just largely inconclusive imprints on local observational temperatures during the instrumental period (e.g., Mikšovský et al., 2016a). In this study, the volcanic activity descriptor was adapted from the stratospheric volcanic aerosol optical depth (AOD) series in the 30–90° N latitudinal band compiled by Crowley and Unterman (2013), based on sulfate records in the polar ice cores.

Aside from being the dominant climate mode in the equatorial Pacific, ENSO has been linked to various aspects of weather patterns in many regions around the globe (e.g., Brands, 2017; Dätwyler et al., 2019 and references therein). These teleconnections manifest themselves to some extent in the European climate (e.g., Brönnimann et al., 2007), and indications of an ENSO imprint have also been found in Czech temperature series (Mikšovský et al., 2014) as well as in the drought indices (Brázdil et al., 2015a). Two ENSO reconstructions were employed here: a reconstruction of interannual ENSO variability based on tree rings by Li et al. (2011) and a multi-proxy reconstruction of inter-decadal temperature variations in the Niño3 region by Mann et al. (2009). To limit the presence of long-term trends in the Mann et al. (2009) data, largely reflecting external forcing rather than a manifestation of internal climate dynamics, the series has been detrended by subtracting the 70-year moving average of the Northern Hemisphere mean temperature, also provided by Mann et al. (2009); the largely trend-free series by Li et al. (2011) was used in its original form.

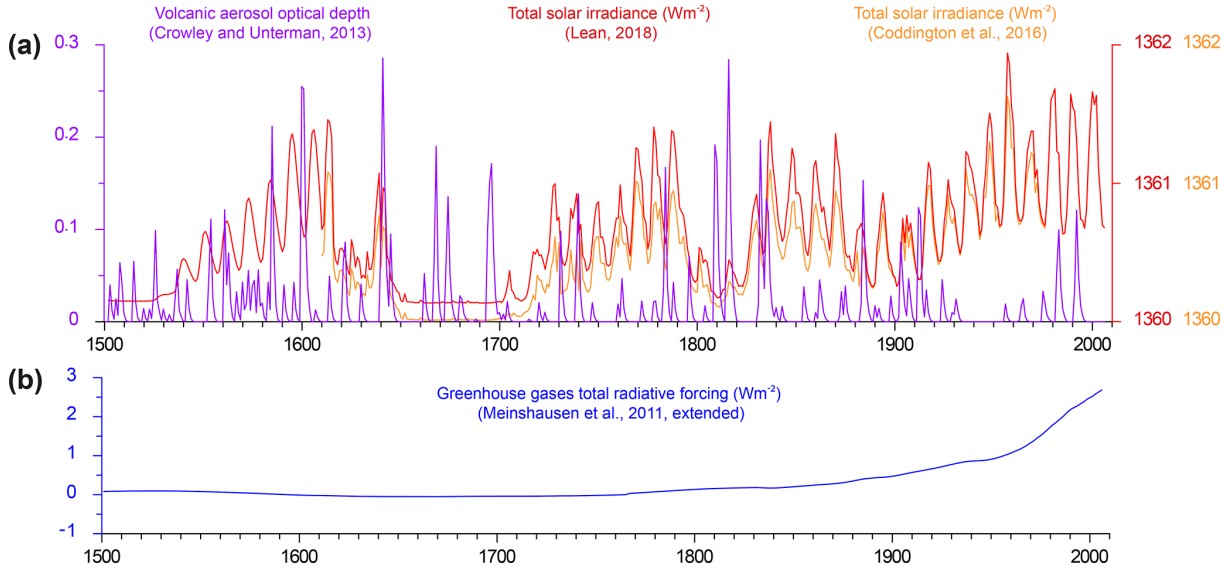

**Figure 2.** Annual series of external forcing descriptors: **(a)** volcanic aerosol optical depth and total solar irradiance; **(b)** greenhouse gases radiative forcing.

In the area of the northern Atlantic, the Atlantic Multidecadal Oscillation (AMO) provides the major source of inter-decadal variability, with an assumed main periodicity of about 70 years (e.g., Enfield et al., 2001). To analyze possible AMO influence over the last 5 centuries, multi-proxy reconstruction of annual temperatures in the AMO region by Mann et al. (2009) was employed for the 1501–2006 period as well as tree-ring-based AMO reconstruction by Gray et al. (2004), available for the 1567–1990 period (Fig. 3b). Again, due to the presence of strong long-term variations in the Mann et al. (2009) series, detrending by moving mean of the northern hemispheric temperature was applied during the preprocessing phase. The same treatment was also used in the case of the PDO, utilizing Mann et al. (2009) temperature data for the northern Pacific region. It is essential to note that this procedure does not fully conform to the usual definition of the PDO index, which is typically derived from the first principal component of sea surface temperatures in the northern Pacific, detrended by mean global sea surface temperature. For the sake of brevity, however, the PDO designation will hereafter be used for the signal obtained from Mann et al. (2009) data. Additional PDO index reconstructions by MacDonald and Case (2005) and Shen et al. (2006) were also included in our analysis for the 1501–1996 and 1501–1998 periods, respectively (Fig. 3c).

Unlike AMO, PDO and ENSO, dominated by mid- to long-term temporal variations, the North Atlantic Oscillation (NAO) constitutes a faster-oscillating climate mode, although the presence of long-term components has also been reported in some of its indices (e.g., Trouet et al., 2009; Ortega et al., 2015). For the analysis herein, three reconstructions of NAO activity were tested (Fig. 3d). The NAO index

series by Luterbacher et al. (2001), based on various European and Asian documentary and proxy data, is available for 1659–2001 CE at monthly time steps and for 1500–1658 CE at seasonal time steps. For the purposes of this study, it was also analyzed in the form of annual NAO index values, extended to the year 2006 by the instrumental NAO index data by Jones et al. (1997). The annually resolved multi-proxy winter NAO reconstruction by Ortega et al. (2015) was adopted for the 1501–1969 period. Finally, a reconstruction of decadal winter NAO variability by Trouet et al. (2009) was used for the 1501–1995 period.

## 3 Methods

Multiple linear regression was used to separate and quantify individual components in the series of drought indices, formally pertaining to individual explanatory variables. The statistical significance of the regression coefficients was evaluated by moving-block bootstrap, with the block size chosen to account for autocorrelations within the regression residuals (Politis and White, 2004). The series were analyzed at the annual time step, either as values constituting a mean for the entire year or as values pertaining to a single season of each year in the usual climatological sense: winter (DJF), spring (MAM), summer (JJA) or autumn (SON). The seasonal analysis was only carried out for SPI and SPEI indices, since PDSI definition involves long-term memory. The basic analyses were carried out for the whole 1501–2006 period; more limited time ranges were, however, used for some of the tests involving specific predictors with shorter temporal coverage. To investigate possible instabilities in the relations detected, regression analysis was also carried out for the subperiods

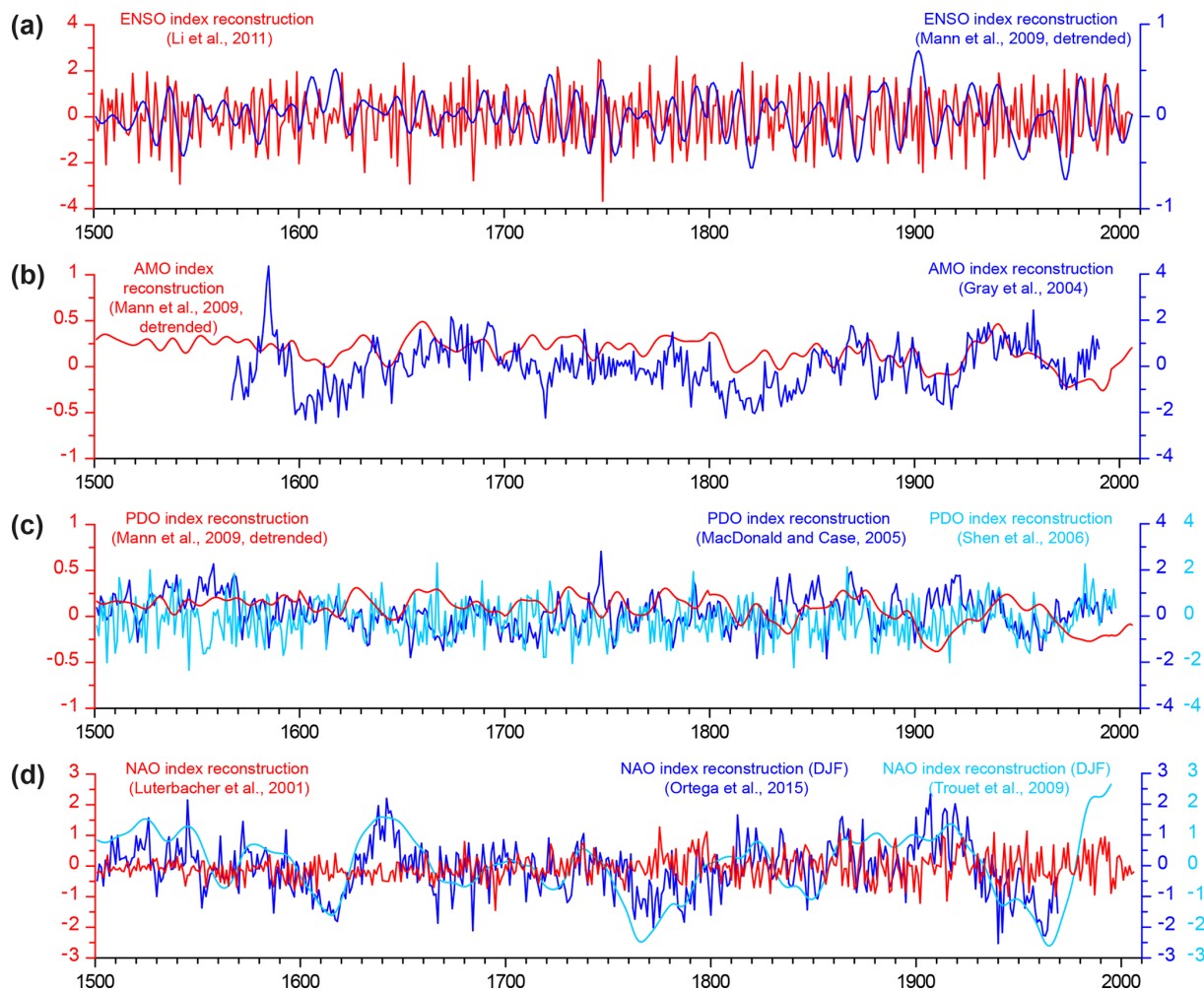

**Figure 3.** Reconstructed annual series of internal climate variability modes: **(a)** El Niño–Southern Oscillation (ENSO); **(b)** Atlantic Multi-decadal Oscillation (AMO); **(c)** Pacific Decadal Oscillation (PDO); **(d)** North Atlantic Oscillation (NAO).

1501 to 1850 and 1851 to 2006 (here considered approximately equivalent to the instrumental period). No time lag was applied to the predictors, except in the case of volcanic forcing at seasonal timescales, when a delay of 3 months was used. The results are presented in the form of standardized regression coefficients, i.e., equivalent to a setup with both predictand and predictor series converted in linear fashion to zero mean and unit variance.

Continuous wavelet transform, based on the Morlet-type mother wavelet, was employed to obtain a better picture of oscillatory components in the series of drought characteristics and explanatory variables. By providing transformation of the target signals into the time–frequency domain, the wavelet approach facilitates the investigation of (in)stability in the oscillatory components of the target signals and, through cross-wavelet spectra, their mutual coherence. This makes it possible to identify subperiods of activity associated with oscillations of interest, and their eventual similarity (and potential transfer) between individual signals.

The statistical significance of the wavelet coefficients was evaluated against the null hypothesis of a series generated by an autoregressive process of the first order (AR(1)), using the methodology described by Torrence and Compo (1998). Standardized and bias-corrected coefficients are presented for the wavelet (Liu et al., 2007) and cross-wavelet (Veleda et al., 2012) spectra.

## 4 Results

### 4.1 Regression-estimated drought responses

Standardized regression coefficients obtained by multiple linear regression between series of Czech drought indices, temperature or precipitation and a set of explanatory variables, representing external forcings and large-scale internal climate variability modes, are shown for annual values in Fig. 4 and for seasonal values in Fig. 5. The regression coefficients associated with the GHG forcing show a clear con-

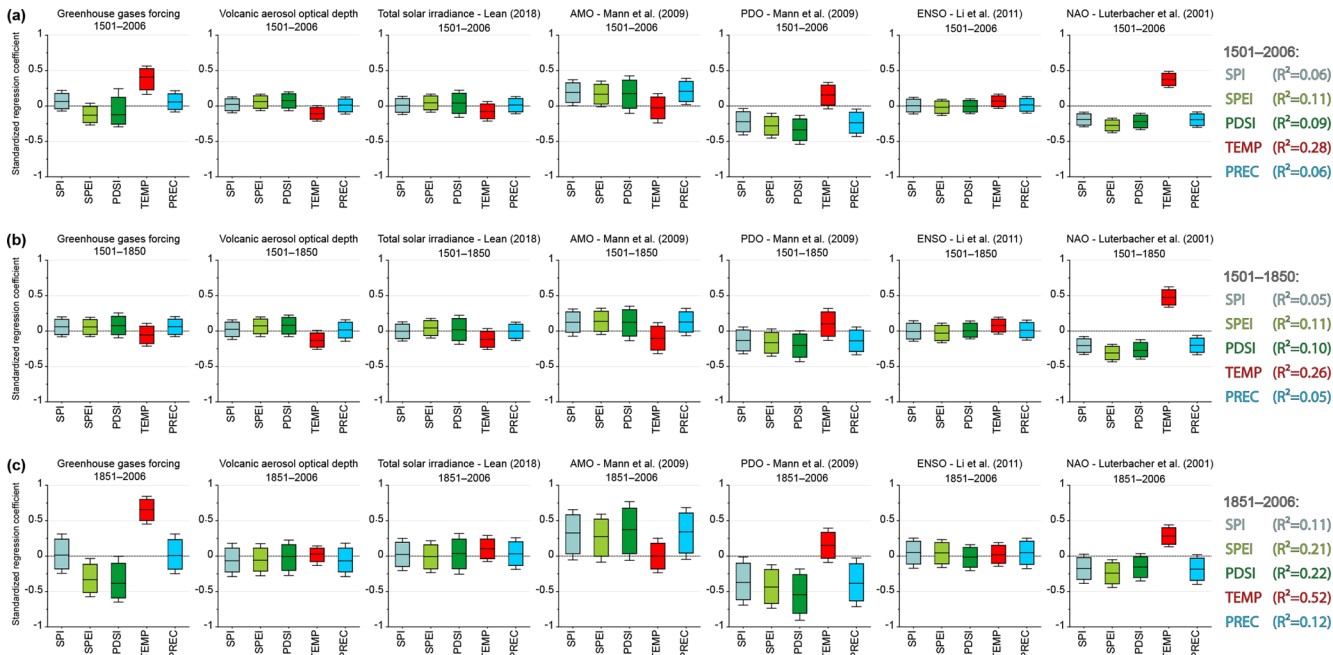

**Figure 4.** Standardized regression coefficients (central line, with 95 % confidence interval shown as the box and 99 % confidence interval as the whiskers), obtained by multiple linear regression between the predictands characterizing local Czech climate (drought indices, temperature and precipitation) and a set of explanatory variables representing external forcings and large-scale internal climate variability modes in various periods: **(a)** 1501–2006, **(b)** 1501–1850, **(c)** 1851–2006. $R^2$: fraction of variance explained by the regression mapping.

trast between the behavior of the Czech temperature (distinct, strongly significant link) and precipitation (statistically nonsignificant connection) series. This reflects a strong formal similarity in the shape of the temperature series and GHG concentration, sharing an increase in the later 20th and early 21st centuries. The connection becomes even more prominent for the 1851–2006 period (Fig. 4c) but does not manifest itself during the pre-instrumental 1501–1850 era (Fig. 4b), in which the GHG signal is mostly featureless. This pattern also appears for individual seasons, with the GHG–temperature link at its relative strongest during SON (Fig. 5). The formal association of GHG forcing with individual drought indices then conforms to their definition: while precipitation-only SPI behaves in a fashion very similar to precipitation itself, stronger (although not always statistically significant) links were indicated for SPEI and PDSI. It is also worthy of note that, due to very strong correlation between the respective time series, quite similar results would have been obtained if the GHG forcing series was replaced with a predictor representing just $CO_2$-related effects or by total anthropogenic forcing including the effects of man-made aerosols.

There is a lack of significant imprint from solar activity in our target series when Lean (2018) solar irradiance data are used for the 1501–2006 period (Fig. 4a). This not only applies to the drought and precipitation data but also to the temperature, despite the analysis period involving marked decreases in solar irradiance in the form of Maunder and Dalton minima. While a negative response borderline significant

at the 95 % level appears for temperature in the 1501–1850 period (Fig. 4b), it disappears when data for 1610–2006 are considered alone, i.e., the period when sunspot data are used in the reconstruction by Lean (2018) (whereas prior to 1610, a more indirect approach is used, utilizing cosmogenic irradiance indices); see Fig. 6e. A statistically significant solar-related signal was also absent in all individual seasons except for SON (Fig. 5), and nonsignificance of the imprint of solar activity was indicated when the Coddington et al. (2016) total solar irradiance series was used as a proxy for solar activity in the 1610–2006 period (Fig. 6d). Overall, our results do not seem to support the existence of a robust solar-induced component in the time series analyzed.

The cooling effect of major volcanic eruptions is clear in the Czech temperature series over the entire 1501–2006 period but becomes statistically nonsignificant when only the instrumental era (1851–2006) is considered. This contrast may stem from the limited amount of major volcanic events taking place after 1850, combined with the fact that individual eruptions, varied in their location and nature, do not form a sufficiently consistent sample for statistical analysis of local volcanism imprints. The volcanism effect on Czech precipitation series is nonsignificant regardless of the period analyzed. As a result, the volcanism-attributed component is negligible in precipitation-only SPI but somewhat more prominent (even though still nonsignificant) in temperature-sensitive SPEI and PDSI. The season-specific outcomes (Fig. 5) are largely consistent with those obtained

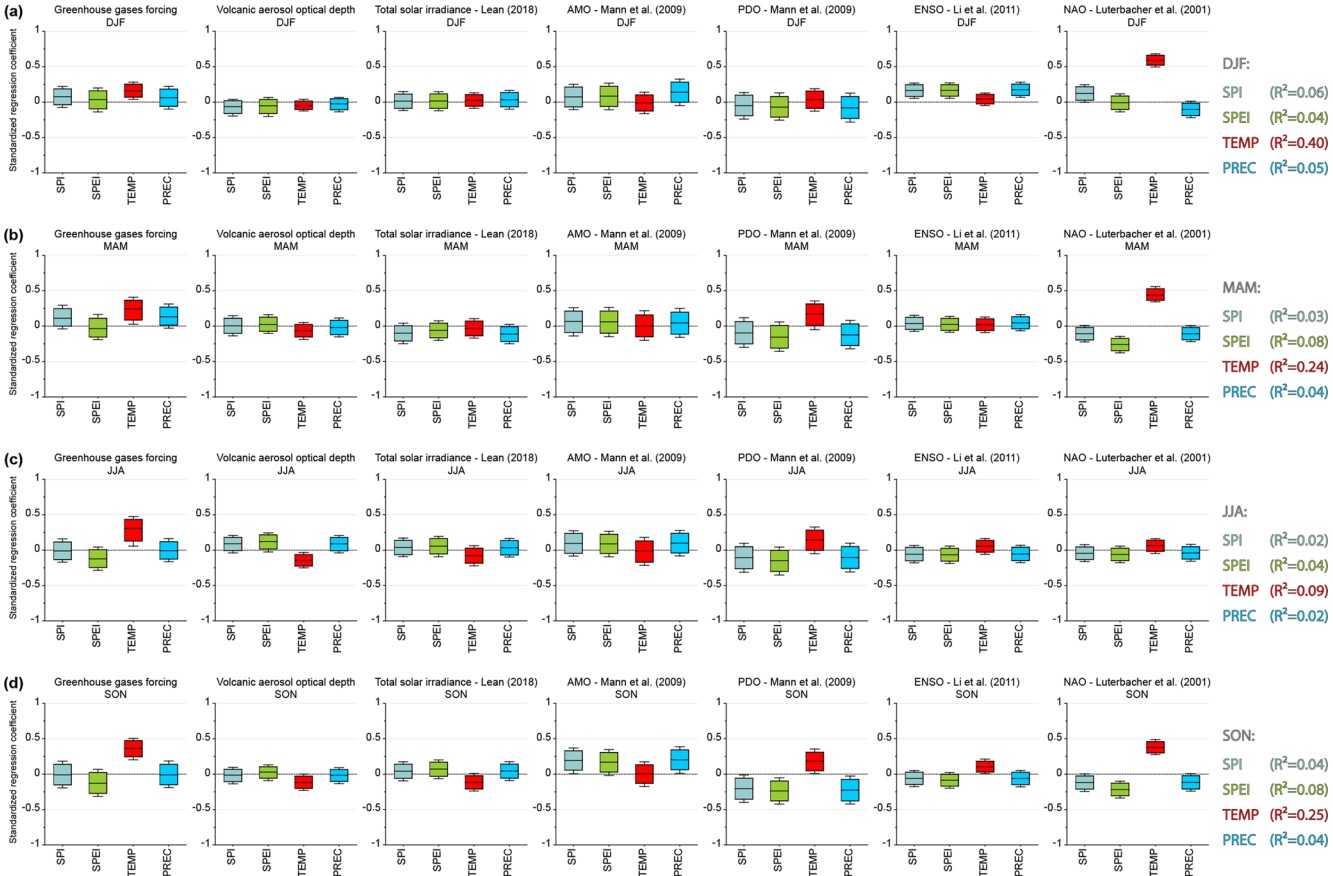

**Figure 5.** Standardized regression coefficients (central line, with 95 % confidence interval shown as the box and 99 % confidence interval as the whiskers), obtained by multiple linear regression between the predictands characterizing local Czech climate (drought indices, temperature and precipitation) and a set of explanatory variables representing external forcings and large-scale internal climate variability modes during individual seasons: **(a)** winter (DJF), **(b)** spring (MAM), **(c)** summer (JJA), **(d)** autumn (SON); 1501–2006 period. $R^2$: fraction of variance explained by the regression mapping.

for the year as a whole, with some degree of cooling indicated for all seasons and strongest during summer, when a response borderline significant at the 95 % level also appears for precipitation and both SPI and SPEI, indicating mildly wetter conditions following episodes of volcanism reaching the stratosphere.

Despite the previous indications of possible (albeit rather weak) links between the Czech drought regime and the activity of ENSO (Brázdil et al., 2015a), this analysis did not reveal any statistically significant associations within the annual data when the ENSO reconstruction by Li et al. (2011) was used, even though there was a slight tendency towards higher temperature during the positive ENSO phase (Fig. 4). This tendency was even stronger (and borderline statistically significant at the 95 % level) for the Mann et al. (2009) interdecadal ENSO variability (Fig. 6i). In the case of season-specific results, a significant tendency towards higher precipitation (and wetter conditions) was indicated for the positive ENSO phase in DJF for Li et al. (2011) data (Fig. 5) as well as a borderline significant tendency towards warmer and drier

conditions in SON. No such links appeared when the reconstruction by Mann et al. (2009) was used.

The AMO index based on the Mann et al. (2009) data was found to be linked to the variability of Czech precipitation, as well as all drought indices during the 1501–2006 period (Fig. 4a). However, its effect is somewhat less prominent prior to 1850 (Fig. 4b). A similar response also appears when the AMO reconstruction by Gray et al. (2004) is employed (Fig. 6f), with the statistical significance of the link lower than for the Mann et al. (2009) data. The existence of a robust connection is therefore uncertain, especially considering previously reported low AMO influence during the instrumental period (Brázdil et al., 2015b; Mikšovský et al., 2016b). There is a similarity between AMO-related links detected for the annual data and their season-specific counterparts (Fig. 5), with SON showing the highest relative degree of statistical significance.

The imprint of decadal and multi-decadal temperature variability in the northern Pacific area, associated with the activity of the PDO, was found to be quite distinct in all

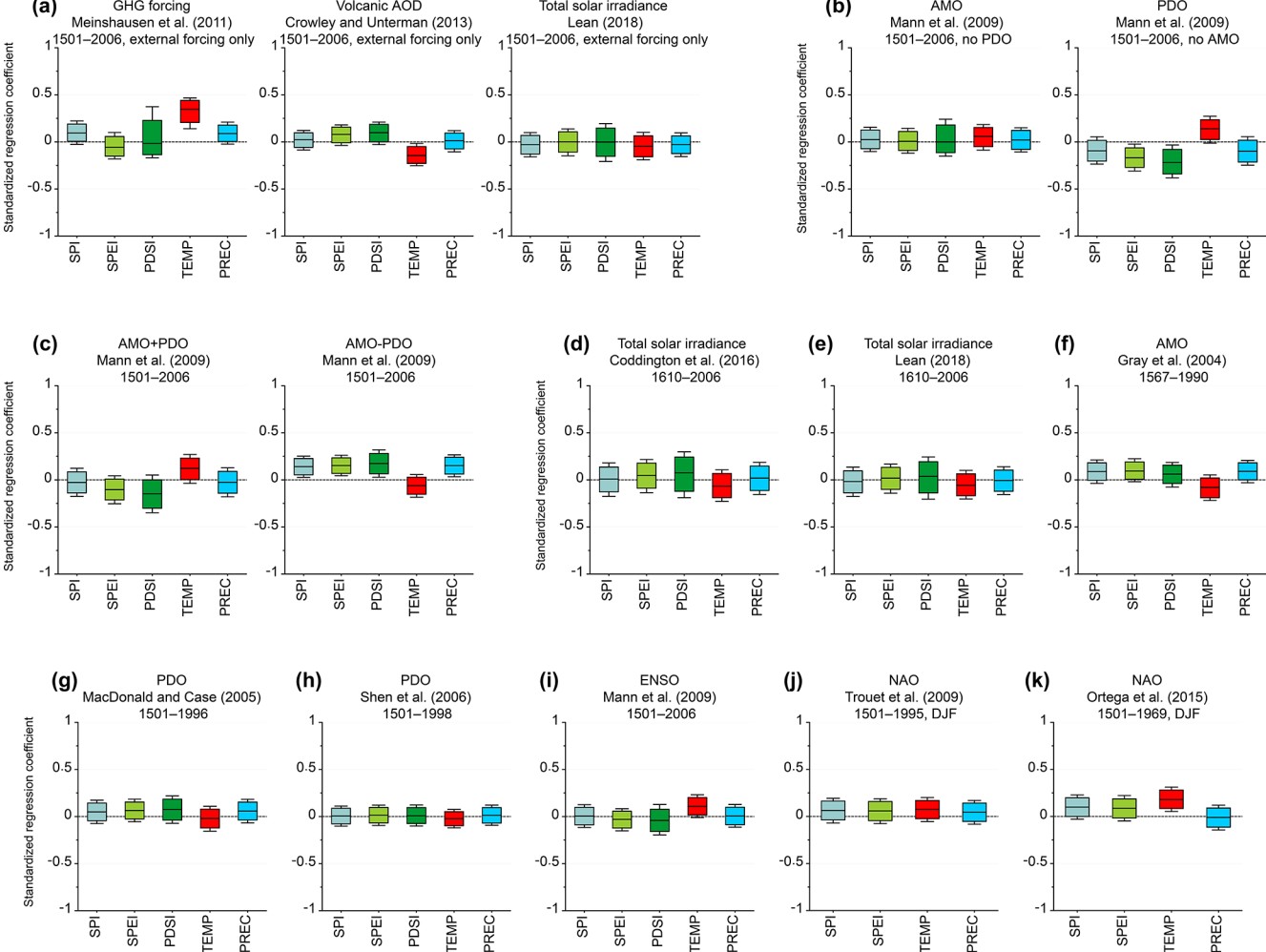

**Figure 6.** Standardized regression coefficients (central line, with 95 % confidence interval shown as the box and 99 % confidence interval as the whiskers), obtained by multiple linear regression between the predictands characterizing local Czech climate (drought indices, temperature and precipitation) and **(a)** a set of explanatory variables representing external forcings only (i.e., a three-predictor setup involving GHG forcing, volcanic aerosol optical depth and total solar irradiance only), **(b)** a setup from Fig. 4 with only AMO or only PDO predictor instead of both, **(c)** a setup from Fig. 4 with AMO and PDO predictors replaced by their mean (AMO + PDO) and difference (AMO − PDO), and **(d–k)** setups with one of the predictors replaced by its alternative version while keeping the rest the same as in Fig. 4 (only coefficients for the modified predictors are shown).

three drought indices, but especially in PDSI, when Mann et al. (2009) data were used as a PDO index source (Fig. 4). The influence of PDO is also strong in Czech precipitation data but less so in central European temperature series. While the relationships are formally stronger for the 1851–2006 period, the link is also significant in the earlier part of the series (1501–1850), especially for PDSI. On a seasonal basis, the strongest drought association with PDO was indicated for SON, whereas only nonsignificant links were found for DJF (Fig. 5). When the Mann et al. (2009) data are replaced with the reconstructions by MacDonald and Case (2005; Fig. 6g) or Shen et al. (2006; Fig. 6h), no significant response to the PDO index appears for any of the target variables.

Of the NAO index reconstructions employed here, the series by Luterbacher et al. (2001) was found to be associated with the strongest and statistically most significant responses in the 1501–2006 period as well as in both its sub-periods (Fig. 4). Positive NAO phase correlates with high temperatures and low precipitation totals and thus negative values of drought indices. In terms of season, this pattern is well-pronounced in MAM and SON, whereas in DJF the link to precipitation and SPI is only borderline significant at the 95 % level and no imprint appears for SPEI (Fig. 5). In JJA, the effect of NAO is nonsignificant regardless of the target variable. The effect of winter NAO was found to be just marginally similar for the Ortega et al. (2015) data, only statistically significant for temperature and without a

response in the case of precipitation (Fig. 6k). Finally, the winter NAO index by Trouet et al. (2009) was not associated with a statistically significant response in any of the target variables (Fig. 6j); note, however, that unlike the Luterbacher et al. (2001) and Ortega et al. (2015) series, this reconstruction only captures the long-term variations in NAO and thus foregoes most of the NAO variability spectrum.

## 4.2 Shared periodicities between drought indices and explanatory factors

Although Brázdil et al. (2015a) demonstrated well-pronounced interannual and inter-decadal variations in the Czech MAM–JJA drought data, these were predominantly irregular. As follows from Figs. 1 and 7, no persistent, dominant periodic or quasi-periodic component exists in any of the series of the Czech drought indices or in their temperature or precipitation counterparts. The same also holds when data for individual seasons are studied (not shown). While this finding is not surprising in the context of the central European climate, it also confirms only a limited direct influence for the factors of periodic nature, such as the 11-year solar cycle or the approximately 70-year periodicity of the North Atlantic sea surface temperature, typically ascribed to AMO (note also that although this periodicity is noticeable in the wavelet spectra of both AMO series here, it tests as statistically significant only from the 18th century onwards in the Mann et al., 2009, data; Fig. 7d). Nonetheless, partial interactions at specific oscillatory periods are a possibility, potentially detectable through cross-wavelet analysis. The respective spectra are visualized in Fig. 8 (of the drought indices, results for only SPEI are shown, as the cross-wavelet patterns are very similar for SPI and PDSI); additional results for alternative predictors are provided in Fig. S1 in the Supplement.

While an approximately 11-year oscillation is one of the defining features of the total solar irradiance series (Fig. 7b), the match with similar periodicities in the Czech drought data is limited to just a few short periods, exhibiting mutually quite different phase shifts (Fig. 8a). This outcome supports the conclusions of the regression analysis in Sect. 4.1, indicating the lack of a robust direct link between the central European climate and solar activity variations.

Several noteworthy interaction regions in the time–frequency space seem to exist between the Czech climate descriptors and predictors with distinct inter-decadal oscillations, AMO (Fig. 8b) and PDO (Fig. 8c). These are particularly noticeable in the reconstruction by Mann et al. (2009) and quite similar for PDO and AMO, following on from the resemblance of the two series. More curious, however, is the similarity between cross-oscillatory patterns pertaining to the relation between AMO or PDO and temperature or precipitation; while some differences appear, the general positions of the areas of significant links are quite similar for both series. None of these regions of significant oscillations is, however,

stable throughout the entire period analyzed; a match for periods of ca. 20–30 years appears in the 17th and 18th centuries, as well as during most of the 20th century (albeit with a different phase shift). Another region of high coherence appears for periods of about 70 years from the mid-18th century to the end of the 20th. These features may also be found in the cross-wavelet spectra involving drought indices. However, when the reconstruction by Gray et al. (2004) was used as source for the AMO variability, only oscillations in the ca. 60–100-year range were found to be shared with the Czech drought indices and exhibited profound changes in phase difference throughout the analysis period (Fig. S1a). Similar behavior was also detected for the PDO reconstruction by Shen et al. (2006; Fig. S1c), while no significant periodicity match was found for the PDO data by MacDonald and Case (2005; Fig. S1b).

In contrast to the influence of AMO or PDO, the cross-wavelet spectrum of the Czech climate descriptors vs. ENSO reconstruction by Li et al. (2011) shows no significant coherence regions beyond scattered noise (Fig. 8d). For Mann et al. (2009) inter-decadal ENSO data, there are several discontinuous regions in a period band of 8–16 years, but with highly variable phase shifts, again indicating the lack of a systematic stable relationship (Fig. S1d).

No significant match between the oscillations in the NAO index series and the drought indices was found for the short to mid-periods (it is worthy of note that this result does not imply a lack of relationships as such, merely an absence of common periodicities detectable by the wavelet transform). Regions of possible coherence were, however, detected for the longer timescales. Employing the NAO index reconstruction by Luterbacher et al. (2001), common oscillations with periods of around 70 years were found, especially during the 18th and 19th centuries (Fig. 8e). For the Ortega et al. (2015) winter NAO data, significant common oscillations of ca. 60–100 years appear for temperature throughout most of the analysis period (Fig. S1e). A similar, even stronger pronounced, pattern of similarities at multi-decadal scales was also found for the Trouet et al. (2009) NAO index, owing to the strong resemblance of the long-term components in the Ortega et al. (2015) and Trouet et al. (2009) NAO data.

## 5 Discussion

The results obtained from the analysis of three drought indices in the current paper do generally conform to the conclusions of Brázdil et al. (2015a), highlighting the role of anthropogenic forcing in establishing the long-term Czech drought regime. The general importance of anthropogenic effects in the occurrence and risk of meteorological drought has previously been confirmed by, for example, Gudmundsson and Seneviratne (2016). Based on an observational and climate-model-based assessment, they concluded that anthropogenic emissions have increased the probability of drought years in

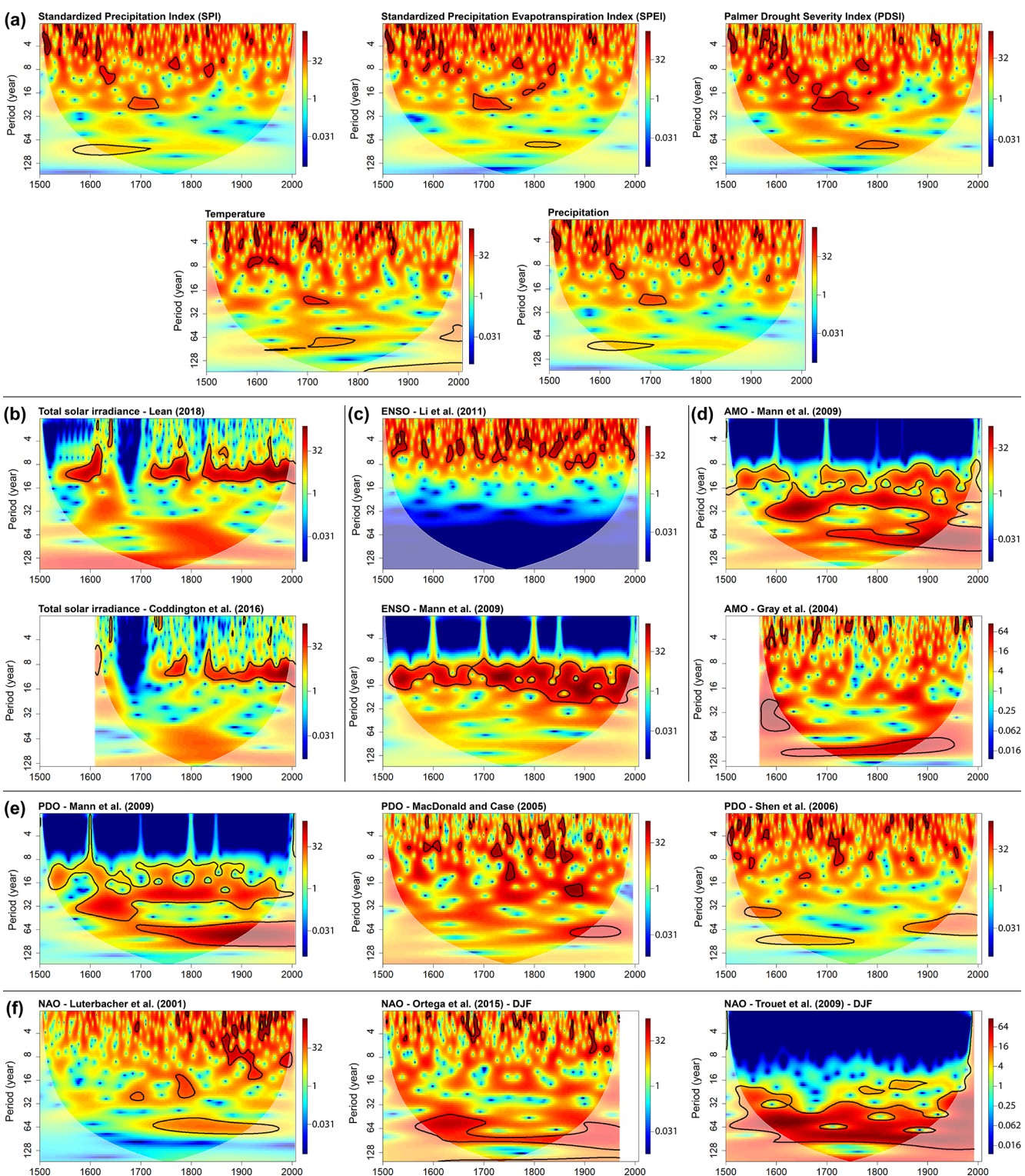

**Figure 7.** Standardized wavelet power spectra of annual series of **(a)** Czech drought indices, temperature and precipitation, as well as of individual explanatory variables with prominent oscillatory component: **(b)** total solar irradiance, **(c)** ENSO, **(d)** AMO, **(e)** PDO and **(f)** NAO. Areas enclosed by a black line correspond to wavelet coefficients statistically significant at the 95 % level, AR(1) process null hypothesis. The lower-contrast areas pertain to the cone of influence, i.e., a region with diminished representativeness of the wavelet spectra due to edge effects.

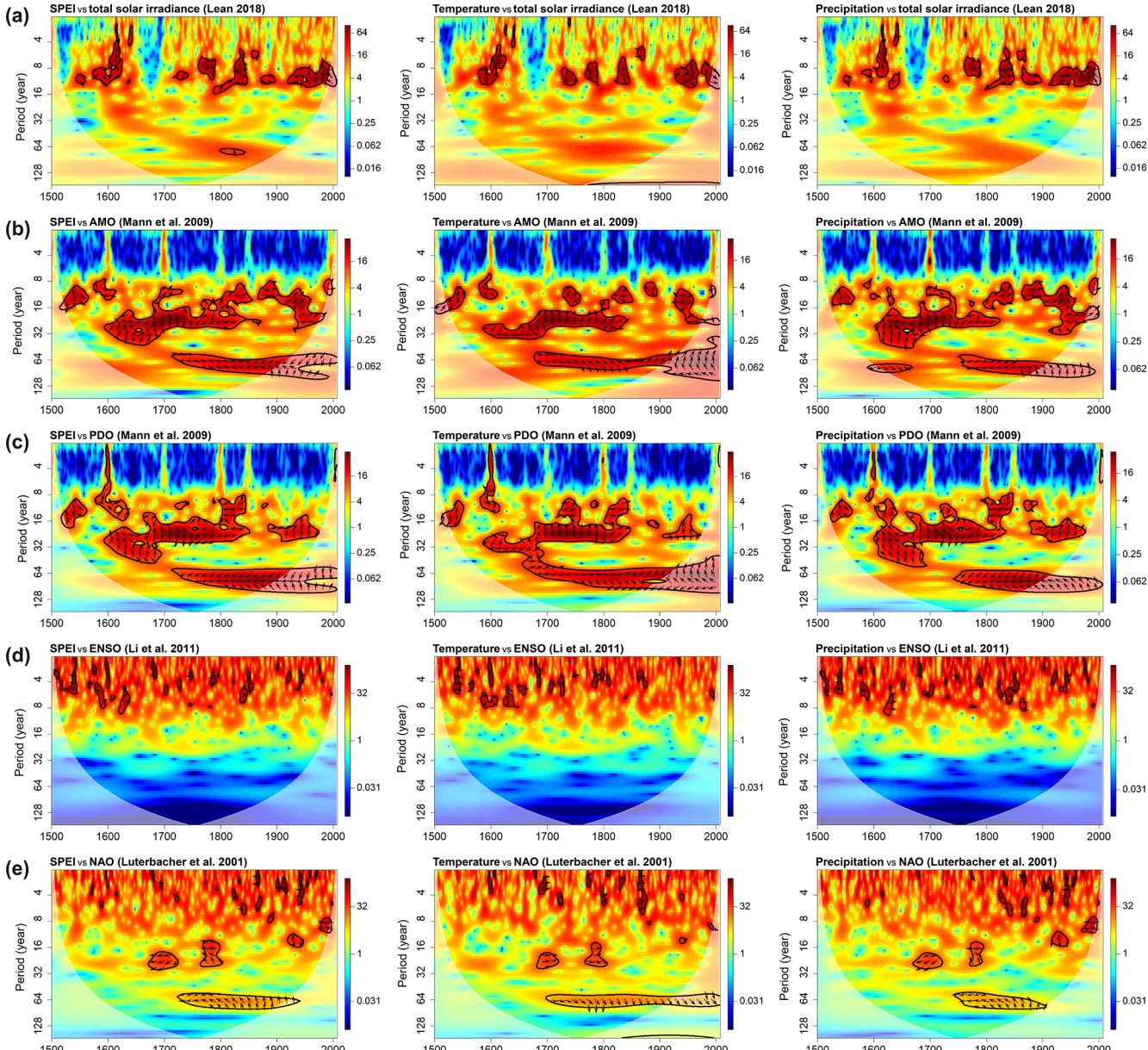

**Figure 8.** Standardized cross-wavelet spectra between series of Czech SPEI, temperature, precipitation and selected explanatory variables with a distinct oscillatory component: **(a)** total solar irradiance, **(b)** AMO, **(c)** PDO, **(d)** ENSO and **(e)** NAO (annual time step; standardized and bias corrected, as per Veleda et al., 2012). Areas enclosed by a black line correspond to cross-wavelet powers statistically significant at the 95 % level, AR(1) process null hypothesis; the arrows indicate local phase difference, with → corresponding to the two signals being in phase and ← indicating a shift of half the period. The lower-contrast areas pertain to the cone of influence, i.e., a region with diminished representativeness of the wavelet spectra due to edge effects.

the Mediterranean and decreased it in northern Europe. The evidence related to central Europe appeared inconclusive. This is consistent with increasing drought severity related to temperature rise in southern Europe (Vicente-Serrano et al., 2014). More recently, Naumann et al. (2018) demonstrated how drought patterns can worsen in many regions of the world (including southern Europe) at a global temperature increase of 1.5, 2 and 3 °C compared with the preindus-

trial era. Our findings show that the increase in the ambient GHGs post 1850 is correlated with the increased probability of temperature-associated droughts in the Czech Republic, while during the pre-instrumental period such a link does not manifest itself. Even so, it should be emphasized that regression (or, more generally, statistical) analysis only reveals formal similarities between target and explanatory variables, and cannot prove the presence of physically meaningful re-

lationships on its own. This is particularly true in the case of signals dominated by simple trends, such as the gradual rise of GHG radiative forcing during the industrial era. Our results should therefore be considered a supportive argument regarding the relationship between the drought regime and the anthropogenic forcing, not proof of a causal link.

The findings of this study for the effects of solar activity (see Fig. 4) are consistent with previous results targeting the instrumental period and reporting only weak, if any, solar links to the European climate (see, e.g., Bice et al., 2012; Mikšovský et al., 2014; Brázdil et al. 2015a) or even to global climate descriptors such as global mean temperature (see Benestad, 2003; Gray et al., 2010, for an overview). Despite using a longer period for analysis, involving prominent features of mid- to long-term solar variability in the form of Maunder and Dalton minima, the absence of consistent significant links suggests that the impacts of solar variations on the drought regime are negligible in central Europe, regardless of the obvious importance of solar radiation as the main source of the energy for the climate system. On the other hand, studying drought conditions in central southern Italy in 1581–2007 based on documentary evidence, Diodato and Bellocchi (2011), reported distinct 11- and 22-year cycles, which could reflect single and double sunspot cycles, albeit not consistently present throughout the period analyzed. They even argued that periods of low sunspot activity, such as the Maunder Minimum, could have more impact on drought than local forcing agents. Schwander et al. (2017) studied the influence of solar variability on the occurrence of central European weather types in the 1763–2009 period. They reported fewer days with westerly and west–southwesterly flow over central Europe under low solar activity and an increase in the occurrence of northerly and easterly types. This could be reflected in precipitation totals and droughts as well.

The effects of ENSO on drought variability in central Europe also appear quite limited. Previously, Pongrácz et al. (2003) demonstrated the influence of ENSO on drought occurrence in Hungary; however, the signal was relatively weak in the statistical sense. Brönnimann et al. (2007) reported the presence of a consistent ENSO signal in various European climate characteristics during late winter and spring, modulated by PDO. Bice et al. (2012) showed a weaker influence of ENSO on temperatures in Croatia. Also weaker and less consistent was the ENSO influence on Croatian winter precipitation, modulated by longer-term PDO cycles. Mikšovský et al. (2014) indicated a weak Southern Oscillation influence on Czech temperature series and none on precipitation series. In contrast, Piervitali and Colacino (2001), analyzing drought events derived from rogation ceremonies for the 1565–1915 period in western Sicily, recorded that a reduction in ENSO events took place in periods when many drought events occurred and vice versa. In the analysis procedure herein, however, the only significant response of the drought indices to ENSO occurred for DJF, and only when Li et al. (2011) data were used as the pre-

dictor. Considering the absence of shorter-scale variability in the Mann et al. (2009) series, it may be speculated that the responses in the seasonal data are tied to interannual rather than decadal variability. On the other hand, both ENSO predictors employed here, by Li et al. (2011) as well as by Mann et al. (2009), have been linked to a tendency towards higher temperature during the positive ENSO phase, although only borderline statistically significant at the 95 % level for the Mann et al. (2009) data when the entire period 1501–2006 is considered. Specific conclusions regarding the nature and reliability of the respective links are, however, difficult to make without a supporting analysis of circulation patterns. Furthermore, comparison with prior results obtained for the instrumental period (such as the seasonal responses reported by Brázdil et al., 2015a, b) is rendered problematic by profound differences in the nature of ENSO-related explanatory variables.

While previous studies of the possible influence of explosive volcanism on Czech droughts reported either no significant connection (Brázdil et al., 2015a) or only a weak and geographically sporadic effect (Mikšovský et al., 2016b), this analysis of more than 5 centuries of data has revealed a more distinct volcanic imprint, suggesting a tendency towards wetter conditions following major eruptions, largely due to temporary temperature decrease, most prominent in summer. Our results conform well to the findings by Fischer et al. (2007), who reported a distinct radiative cooling effect of major tropical eruptions during European summer over the last 5 centuries. The results are also consistent with the analysis by Gao and Gao (2017), who studied European hydroclimatic responses to volcanic eruptions over the past 9 centuries. Applying a superposed epoch analysis, they found a significant wetting response for 31 tropical eruptions (95 % confidence level) in years 0 (the year of eruption) and +1 (the first year after eruption) and a significant drying in year +2. Large high-latitude eruptions in the Northern Hemisphere gave rise to drying responses in western–central Europe occurring in year +2 and shifting southeastwards in years +3 and +4. Similarly, the analysis of the MAM and JJA hydroclimate over Europe and the Mediterranean during the last millennium by Rao et al. (2017) indicated wet conditions occurring in the eruption year and the following 3 years in the western Mediterranean, while northwestern Europe and the British Isles experience dry conditions in response to volcanic eruptions, with the largest moisture deficits in post-eruption years and the Czech Lands being most affected 2 and 3 years after the eruption. Písek and Brázdil (2006) analyzed the imprints of seven large tropical eruptions in four temperature series and three global radiation series in central Europe. They demonstrated that the volcanic signal in regional series is not as strongly expressed as that on a hemispheric scale, owing to varying local effects and circulation patterns. The climatological responses to eruptions in areas closer to central Europe, such as Iceland and the Mediterranean, were identified as more important. This was con-

firmed by a more recent and detailed analysis of the climatological and environmental impacts of the Tambora 1815 eruption on the Czech Lands (Brázdil et al., 2016c) and by its comparison with the Lakagígar 1783 eruption (Brázdil et al., 2017). The presence of a distinct signature of the Tambora eruption was also confirmed in the central European tree-ring chronologies, though overestimated in both intensity and duration of the cooling (Büntgen et al., 2015). Overall, it appears that while the effects of individual volcanic eruptions or their shorter sequences on central European droughts are difficult to isolate from the background of other influences, their existence becomes more noticeable from multi-century series, covering a larger number of powerful volcanic events.

The influence of AMO and PDO on drought variability has already been demonstrated in the results of several papers (e.g., Enfield et al., 2001; Sutton and Hodson, 2005; McCabe et al., 2004; Mohino et al., 2011; Oglesby et al., 2012; Baek et al., 2017). Here, a connection of Czech drought indices to both these oscillations was indicated in the case of the Mann et al. (2009) data, more prominently for PDO. This result is also consistent with the outcomes of an analysis by Mikšovský et al. (2016b), applying linear regression to the seasonal drought index data from several Czech locations in the 1883–2010 period and reporting quite a strong link to the PDO index resulting from an interaction of PDO-correlated components in both precipitation and temperature. However, a potential problem with our analysis procedure stems from the close similarity of the pair of predictors representing AMO and PDO variability. Despite the removal of the long-term temperature component from the original temperature reconstructions by Mann et al. (2009), the Pearson correlation coefficient of the two series is 0.77 over the 1501–2006 period. As a result, the AMO and PDO predictors are competing for the same components in the target signals and the confidence intervals of the resulting regression coefficients are inflated compared to the other explanatory variables (Fig. 4). This similarity is also apparent from the cross-wavelet spectrum between the AMO and PDO series (Fig. S2b), revealing high coherence especially for periods of ca. 20–30 and 60–100 years, with relatively stable phase shifts, especially for the latter band. Considering the relative similarity of magnitude and significance of the regression coefficients for AMO and PDO and their typically opposite signs, it is difficult to assign the variations in the target variables to one or the other. When employed individually (i.e., either AMO or PDO, but not both), the PDO series constitutes a more influential predictor than AMO, with links to SPEI and PDSI statistically significant at a 99 % level over the 1501–2006 period (Fig. 6b). On the other hand, AMO alone produces no significant links to drought indices (Fig. 6b). To investigate this behavior further, the AMO–PDO pair was replaced with their mean value and their difference (this setup formally corresponds to the outcomes of unrotated principal component analysis applied to a bivariate system consisting of the AMO and PDO index series,

with the mean value responsible for 88 % of total variance of the AMO–PDO pair and the difference responsible for the remaining 12 %). Quite surprisingly, a more significant connection to the series of Czech drought indices was indicated for the AMO–PDO difference (all responses statistically significant at the 99 % level) rather than their common value (Fig. 6c). This suggests that the contrast between the temperature anomalies in the northern Atlantic and northern Pacific, in addition to their individual variability, may be potentially influential in the context of the inter-decadal components of central European droughts.

In contrast with the data by Mann et al. (2009), the effects of AMO or PDO on Czech drought indices were less pronounced when other reconstructions were employed. In the case of AMO, the use of the Gray et al. (2004) series revealed tendencies qualitatively similar to the AMO Mann et al. (2009) predictor (higher precipitation and higher drought indices for positive AMO phase and negative temperature anomalies), but with lower statistical significance (Fig. 6f). While both reconstructions are only moderately correlated ($r = 0.29$ over the 1567–1990 period, increasing to $r = 0.39$ when the shorter-term oscillations were removed from the Gray et al., 2004, data by a 11-year running average), there is a distinct similarity in the oscillations in the 60–100-year period band (Fig. S2c).

The similarity of individual reconstructions was even weaker for PDO, with mutual correlations of the three reconstructions (Mann et al., 2009; MacDonald and Case, 2005; Shen et al., 2006) not exceeding $r = 0.17$ in absolute value over their respective overlap periods ($r = -0.17$ being the correlation between the Mann et al., 2009, and MacDonald and Case, 2005, data). Clear differences between the reconstructions are also apparent in the cross-wavelet spectra (Fig. S2c). While some regions of common oscillations exist, especially in the ca. 20–30- and 60–100-year period bands, the phase shifts vary substantially. The contrasts among individual reconstructions should not be surprising in view of the major differences in data inputs employed for their creation (global multi-proxies for Mann et al., 2009, east China summer rainfall for Shen et al., 2006, and North American tree-ring data for MacDonald and Case, 2005). Choice of reconstruction obviously plays a pivotal role in an analysis such as ours; considering that the Mann et al. (2009) PDO series was the only one manifesting statistically significant links to the Czech drought, temperature and precipitation data and that several other series are available as parts of the same dataset (including the mean hemispheric temperature), this dataset can be deemed of particular interest for future analyses concerned with central European climate. Even so, it should be stressed that different reconstructions emphasize different aspects of the climate subsystem in question, and our results should not be considered a proof of the inherent superiority of any of them.

Clear responses of all drought indices as well as temperature and precipitation to NAO were detected for the Luter-

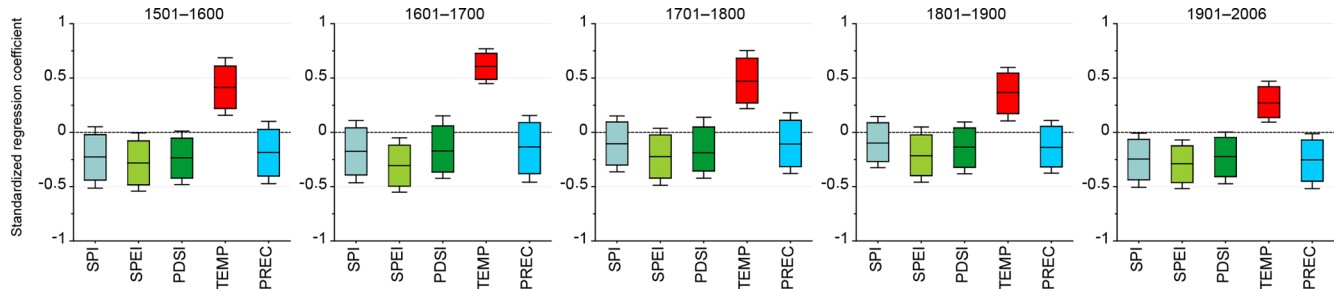

**Figure 9.** Standardized regression coefficients (central line, with 95 % confidence interval shown as the box and 99 % confidence interval as the whiskers), obtained for the NAO index (Luterbacher et al., 2001) in five nonoverlapping subintervals of the 1501–2006 period. Multiple linear regression was employed in a seven-predictor setup identical to the one in Fig. 4 (only coefficients for the NAO index are shown).

bacher et al. (2001) data. The existence of such a relationship is hardly surprising, considering the pivotal role of NAO in establishing the central European climate. The temporal stability of this link may perhaps be of more interest, especially during the pre-instrumental period. While the cross-wavelet transform only suggests potential coherence for periodicities around 70 years, especially for temperature (Fig. 8e), the dominance of relatively fast interannual variability in the NAO data allows for the regression mappings to be split to more segments than in our other tests. In Fig. 9, regression coefficients pertaining to NAO are therefore shown for the individual centuries. While there are some differences among these subperiods, the links are statistically significant for all of them in the case of SPEI and temperature. This outcome not only confirms the presence of links between droughts and NAO, but it also verifies the suitability of the Luterbacher et al. (2001) reconstruction for their analysis, even in the early parts of the data and regardless of a certain degree of heteroscedasticity in the Luterbacher et al. (2001) series (manifested through lower variance in the early parts of the NAO series for all seasons but DJF). The temperature-related results obtained with the Luterbacher et al. (2001) data were partly confirmed by the Ortega et al. (2015) reconstruction for the 1501–1969 period, although statistical significance of the link was generally lower (Fig. 6k); also, just a comparison for DJF was performed, given the winter-specific nature of the Ortega et al. (2015) NAO index series.

It should be mentioned that the fraction of variance explained by the regression mappings ($R^2$) is quite low in some of the cases presented above: it is about 0.06 for SPI or precipitation at an annual resolution in the 1501–2006 period (Fig. 4a). A slightly higher value ($R^2 = 0.11$) was achieved for the temperature-sensitive SPEI, and even higher ($R^2 = 0.28$) for temperature itself. Higher $R^2$ was also indicated for some individual subperiods, especially 1851–2006, with $R^2 = 0.11$ for SPI and $R^2 = 0.21$ for SPEI (Fig. 4c). Even so, the components detected in the drought indices, even when statistically significant, do not constitute a predominant source of total variability; this role appears to be played by interannual variations associated with weather

changes closer to synoptic timescales and tied to local climate dynamics (see Fig. S4 for an illustrative example, visualizing regression-based estimate of the annual SPEI values and the relevant predictor-specific components). Still, as the statistical significance of some of the links in this analysis suggests, the effects of the extra-European climate drivers should not be dismissed, as they appear to contribute substantially to inter-decadal variability (possibly driven, at least partially, by temperature variations in the AMO and PDO regions) or episodic perturbations (volcanic activity). No distinct structures beyond the AR(1)-consistent autocorrelation decay were found in the regression residuals (Fig. S3), with the exception of possible traces of a 22-year cycle in the PDSI residual series and a weak tendency towards positive autocorrelations for temperature. While these may indicate the presence of unaccounted-for effects of a double solar cycle (previously reported for Italian droughts by Diodato and Bellocchi, 2011) and/or an unexplained trend component, the statistical significance of these residual structures is low.

The interactions among the explanatory variables, often manifested through (multi)collinearity of the predictors, are among the potentially critical aspects of multivariable regression analysis. While this study addressed the high correlation between the AMO and PDO predictors based on Mann et al. (2009) data, there are other possible relationships worthy of attention. Additionally, the links may be subject to time-delayed responses of the target variables to the predictors or delayed responses of internal variability modes to external forcings. To investigate these, cross-correlation functions were examined between the target variables and predictors with pronounced interannual variations, as well as between the selected predictors themselves. No prominent short-term time-delayed responses were detected regarding the direct effects of solar or volcanic activity (Fig. S5); if present, distinct extrema of cross-correlations only occurred for concurrent, non-lagged series. Additionally, the presence of long-term components was detected in the solar-related autocorrelation CE2 functions. Since these stem from interaction of long-term trends in the time series and cannot be reliably interpreted via correlation or regression techniques,

they were not taken into account in our analysis. Attention has also been paid to the possibility of delayed responses to volcanic or solar activity in the NAO index, considering the previous reports of positive NAO phase during several years following large volcanic eruptions (Ortega et al., 2015; Sjolte et al., 2018). No clear delay was detected for the analysis setup herein (Fig. S6). In the case of the NAO–TSI cross-correlation, a maximum was indicated for a lag of 2–3 years. While this behavior may be indirectly related to the 5-year delayed circulation response to solar activity reported by Sjolte et al. (2018), its magnitude was rather low in our analysis setup. Finally, to assess whether part of the variability generated by external forcings may be mediated through predictors pertaining to internal climate variability modes, regression analysis was carried out with external forcings only (Fig. 6a). The resulting responses were found to be very similar to the setup with all seven predictors (Fig. 4), suggesting just a limited direct linear imprint of external forcings in the predictors representing influential internal climate variability modes in our analysis.

One of the key questions associated with any analysis of multi-centennial climatic signals is the issue of stability in the patterns and relations observed. While this study attempted to address this subject within its regression-based analysis by investigating two shorter subperiods in the data and an even finer division was employed to study the stability of the NAO-related links, such an approach is problematic for the factors dominated by variations at long periods (such as AMO). In this regard, cross-wavelet transform provided some extra insight; it appears that while periodic oscillations do not dominate Czech drought indices, some specific timescales are involved in the interactions between our target and explanatory variables. However, none of these connections is persistent throughout the entire analysis period, even though some of them (especially the relatively coherent link indicated for periods of around 70 years) span several centuries and appear for multiple target or explanatory variables. To better understand these links and their implications, a transition to more complex regression methods will be desirable in the future. This extension of analytical methods should also be accompanied by a more detailed analysis of the uncertainties in the pre-instrumental data, including intercomparison with other data types (such as dendroclimatic reconstructions).

## 6 Conclusions

The current paper analyzed imprints of climate forcings and large-scale internal variability modes in three multi-centennial series of drought indices (SPI, SPEI, PDSI) derived from documentary and instrumental data after 1501 CE for the Czech Lands. The results confirmed some of the previous findings derived from instrumental data; in other cases, such an extended analysis period facilitated better identification and quantification of the factors responsible for Czech drought regimes and a more complete understanding of how temperature and precipitation mediate the respective links:

i. GHG concentration (and corresponding radiative forcing) matches the long-term trend component in the central European temperature quite well. This warming is then also reflected in a drying tendency in the temperature-sensitive drought indices (SPEI and PDSI), although not always in a statistically significant manner. Even considering that statistical attribution analysis can only reveal formal similarities and cannot verify the causality of the links detected, the dynamics of the relationship during pre-instrumental and instrumental periods and other available evidence (including data from climate simulations) support the existence of an anthropogenically induced drying effect in central Europe, primarily tied to temperature increase rather than precipitation changes.

ii. While the results herein confirmed the lack of a consistent solar variability imprint on the Czech drought series, a signature of temporarily wetter conditions following major stratospheric volcanic eruptions was detected, largely tied to transitory temperature decrease. This behavior appears undetectable from instrumental data alone, probably due to the insufficient number of large volcanic events.

iii. Unlike the mostly nonsignificant response to ENSO, AMO and (especially) PDO appear to be tied to decadal and multi-decadal components in (at least some) drought indices. Even more curiously, more significant drought components appear to be linked to the difference between AMO and PDO phases rather than to their common value. Further validation will, however, be needed to verify whether this behavior is a manifestation of actual physical relationships, especially considering that it only appears for some of the AMO or PDO reconstructions in our analysis.

iv. NAO was reaffirmed as a powerful driver of drought variability in this analysis. For the primary NAO reconstruction in our tests (Luterbacher et al., 2001), not only were links detected for all the drought indices as well as temperature and precipitation, but their presence was confirmed throughout the entire analysis period, including its earliest parts.

Overall, the results herein indicated some potentially prominent, but not completely stable relations between the time series investigated. In the future, these should be investigated more closely, as a better understanding of them is vital to a proper analysis of records spanning many centuries. In this context, the reliability of the reconstructed records needs to be addressed in more detail. Transition to more complex

statistical techniques (possibly nonlinear) may also be desirable, although challenges will have to be overcome related to higher uncertainty and the sometimes limited information content of documentary- and proxy-based data.

**Data availability.** The series of explanatory variables related to external forcings and internal climate variability modes were obtained from public databases (KNMI Climate Explorer – https://climexp.knmi.nl; CRU – http://www.cru.uea.ac.uk/data; NOAA NCEI – https://www.ncdc.noaa.gov/data-access) or from the Supplement to the relevant papers. The central European temperature series is available at https://www.ncdc.noaa.gov/paleo-search/study/9970, while Czech precipitation and drought index series are available from the authors (last access to the above databases: 15 February 2019).

**Supplement.** The supplement related to this article is available online at: https://doi.org/10.5194/cp-15-1-2019-supplement.

**Author contributions.** RB and MT prepared the drought index series. JM adapted the series of explanatory variables and carried out the regression analysis. PP carried out the wavelet and cross-wavelet analysis. All authors participated in interpretation of the results. JM prepared the text and illustrations, with contributions from all coauthors.

**Competing interests.** The authors declare that they have no conflict of interest.

**Special issue statement.** This article is part of the special issue "Droughts over centuries: what can documentary evidence tell us about drought variability, severity and human responses?". It is not associated with a conference.

**Acknowledgements.** This study was supported by the Czech Science Foundation through grant reference 17-10026S. The *biwavelet* package by Tarik C. Gouhier and Aslak Grinsted was used to calculate and visualize the wavelet and cross-wavelet spectra in the R language environment. Tony Long (Svinošice) is acknowledged for English style correction.

**Review statement.** This paper was edited by Jürg Luterbacher and reviewed by three anonymous referees.

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

## Remarks from the language copy-editor

CE1      Please note that it is our house standard to use a numeral and symbol in this context.

CE2      We don't know if this change makes a difference to the meaning. It should be checked with the editor. Could you please provide an explanation why this change is necessary?