# Peer review of "Long-term variability of drought indices in the Czech Lands and effects of external forcings and large-scale climate variability modes"

_Climate of the Past, 2018_

## Referee Comment (RC1) · Anonymous Referee #1 · 22 Jun 2018

The paper analyses the long-term variability of droughts in the Czech lands based on long reconstructions (based on instrumental and documentary data). Time series of drought indices, temperature and precipitation are compared to reconstructions or time series of suspected drivers such as external forcings and oceanic variability modes. Anthropogenic radiative forcing as well as AMO/PDO are identified as influencing factors. The paper is interesting, valuable to the community and within the scope of Climate of the Past. However, I have several comments, which I think the authors should consider, before the paper can be published.

Methods

It is not fully clear which data are monthly or seasonal. Often the text mentions "seasonal and annual" or "monthly, seasonal and annual", which I found confusing. Also,

the drought indices are usually calculated for individual measurement locations or grid cells. Here they are calculated for a large-scale average, as I understand. This should be made clear and explained. In the results section it then becomes clear that the seasons are analysed separately. However, what is the motivation for analysing a autumn or winter drought index?

Multiple linear regression is used to separate individual components, but fully separating external forcing from internal variability (e.g., oceanic modes) is fundamentally difficult. External forcings might operate via altering internal variability modes (e.g. solar and volcanic forcing might change the climate system via AMO or ENSO). Conversely, AMO and PDO have the imprint of global temperature rise. I see that the authors use cross-wavelet spectra, partly to assess the interdependencies, but not systematically. Partial correlation methods could be used to go into more depth here, or different models could be compared. In any case, the interpretations should be phrased very carefully.

The regression model itself is not explained clearly. From the text it becomes clear that different ENSO indices were used, but which model (which ENSO index) is the one shown in Figs. 4 and 5? Furthermore, only very late in the paper we learn that the explained variances are very low, below 5%. Should we even analyse regression models that have no explanatory power? Finally, the effect of reconstruction uncertainty is not discussed.

The paper says little about the mechanisms linking the external and internal drivers to drought and hydroclimatic conditions in general. Obviously a study using reconstructions cannot explicitly address net radiation, soil moisture, temperature effects, land-surface feedbacks, atmospheric circulation effects (blocking), etc. But it would be nice to read the authors' hypotheses. The paper is rather silent about mechanisms. In the introduction Hess-Brezowsky weather types are mentioned, and later the NAO, but the NAO is not incorporated into the analyses and the discussion parts then follows another thread: Doing a PC analysis of AMO/PDO. It would be nice if the Discussion

section could come back to mechanisms at some point.

Minor comments

Abstract, l. 14: "external and internal climate forcings". Please be careful with terminology here and elsewhere. Considering the coupled climate system "forcing" is used for external influences (subdivided into natural and anthropogenic) while internal variability is used for the dynamics of the coupled system even if unforced. When considering only the atmosphere, "oceanic forcing" is sometimes used. In any case, the terms should be defined and used consistently.

P. 2, L. 5: a substantial number of studies: cite

P. 2, L. 5: A lot of work has been done on droughts in the USA. Perhaps before zooming in on Europe, you could mention that.

P. 2, L. 24: instrumental precipitation series

P. 3, L. 10: What time window was used for the SPI?

P. 4, L. 3 and 4: I do not understand this sentence.

P. 4, L. 10: "climate forming agents": rephrase

P. 4, L. 19 and 20: Omit the first part of the sentence, which is unnecessary. Start with "A large part..."

P. 4, L. 26: strong clear?

P. 5, L. 1-3: Perhaps cite Fischer et al. GRL (https://doi.org/10.1029/2006GL027992)

P. 5, L. 19: I am a bit puzzled why the authors use the Mann et al. ENSO series. As the authors write (and other authors have also pointed to that), the reconstruction varies mostly on the 8-20 year time scale. Why use it as an ENSO time series then? I would rather use other ENSO reconstructions. Similarly, for AMO and PDO it would be nice to have two indices for each (e.g. Shen et al. 2006 for the PDO, Gray et al. 2004 for

the AMO).

P. 10, L. 34: Is the tendency for wet conditions after volcanic eruption really due to lower temperatures?

P. 11, L. 34: I am surprised that Sutton and Hodson (2005) paper is not mentioned in context with the AMO effect.

---

## Referee Comment (RC2) · Anonymous Referee #2 · 28 Jun 2018

GENERAL COMMMENTS:

Manuscript under revision is an approach to study of drought in Czech Republic area, taking in consideration previous climatic reconstructions, already published, using these informations to generate drought indices based on instrumental records. Analysis of possible relations with different forcing factors is also made to offer general or initial explanation to drougth mechanisms for this Central Europe study area.

Main effort focused to compare rainfall indices and temperatures for long or complete periods. It's a good first approach to drougth phenomena. It open research to study specific events at higher temporal resolution, impacts and responses, etc.

SPECIFIC COMMENTS

[Figure]

+ Title could include temporal dimension of work of manuscript.

+ Title. Expression "drought" into title is excessively general. A more correct definition of topic developed into manuscript would be "drought indices".

+ Lines 28-30. Seasonal and annual precipitation for 1501-1854 is reconstructed from "document-based precipitation indices". Dobrovolvy et al., 2015. Could explain in a short description general characteristics or contents of these "documents"? How was developed previous analysis. Just to have a connection between original information and present results generated into manuscript. IF drought is analyzed, at least public must know about historical documents used for reconstruction, temporal resolution of information obtained, locations or regions with available information, aspects of natural process and/or and human impacts detected/evaluated....etc.... I understand manuscript can have restrictions of extension, but this short overview would be useful for public.

+ Bibliography used on work is complete and well updated.

+ Effort to offer a background or general overview about drought events is not so complete as we would like find. For example, justification of study of drought. It's a present or potential problem for Czech Republic?, any previous strong event justify this study? How they are drougth conditions in Czech Republic?

+ Historial dimension of drought is not analyzed. Just index values from previous research considered as approach to climatic patterns related to low values of reconstructed precipitation. Drought is not studied by itself as climatic/historic phenomena. This aspect is not negative nor positive. Just it would require any extension of explanations about drougth as climatic pehnomena in introduction of work.

+ No specific drought events are mentioned. No description at least for one event is included into manuscript. Complexity of drought events and related impacts is not described/evaluated. May be authors are preparing other papers with these specific

topics?

+ No explanation about drought as climatic phenomena. How is considered drought in Czech Republic, what criteria are applied, what instrumental thresholds, duration/extension/severity, different concepts/definitions of drougth, affectation of agriculture.... Any explanation would be useful to understand characteristics and effects for public unknowning these specific details.

+ If drougth is defined only from specific indices (SPI, SPEI...), when we work in historical time, out of instrumental data availability, this topic must be taken with more introductory explanations. A more complete and informative approach about how documentary records detect and define droughts, what they record, what transmit....

+ If manuscript is based on previous reconstructions, focused on reconstructed values of mm. rainfall, by total monthly/seasonal/annual resolution, authors must consider they cannot analyze all dimension of droughts. Rainfall indices with positive aspect can cover important drought events, when dry periods are interrupted by strong rainfall events. Knowing what tipe of drougth is under study, these singular aspects could be differenced, generating a better and deeper study.

+ Manuscript doesn't show a clear relation of type of documents and type of information rescued and anallyzed.

+ It would be interesting focuse efforts on variability and extreme events of the same variable before to compare with variability of others proxys.

---

## Referee Comment (RC3) · Anonymous Referee #3 · 16 Jul 2018

**Description**

This paper addresses the understanding of the variability of droughts, temperature and precipitation in the Czech lands 1500-present from the point of view of its dependence on internal drivers (e.g. some specific modes of circulation) and also external forcing factors (volcanic, CO2, etc). For that purpose a multiple linear regression is applied having as predictand variables 3 drought indices and as predictors a set of external and internal drivers.

The purpose of the paper has value and meaningful and solid results in this direction would be worth to be published in CP. If attribution of drought variability or a meaningful step forward in its understanding in the Czech Lands would be attained I think this would be sufficiently valuable in my view to support publication. Therefore, I en-

courage authors to pursue this line of work towards publication. At this state, I would recommend major revision of the manuscript. There are several issues related to the rationale, methodology and description and interpretation of results that in my understanding require revision. I will argue about this in the following points.

General Comments

GC1 General approach to attribution As it is described in the paper, 5 predictand series (3 drought indices and a temperature and a precipitation series) are examined using multiple linear regression as functions of independent predictors, the latter being internal and external in nature. In practice, these are 5 individual multiple regressions. Having that in mind I would suggest to consider the analysis, description and discussion of the: a) selected predictors; b) of the methodological approach; and c) of the residuals of the methodology.

a) Selected predictors. I would argue these are insufficient in both the case of the external and internal subsets.

a.1-Regarding external predictors I have no objections to the ones considered so far but the authors should discuss why important predictors like other greenhouse gases (GHGs), aerosols and particularly land use land cover (LULC) are not considered. For the case of other GHGs than CO2, it would be more elegant either to consider them or to use equivalent CO2. For the case of aerosols some arguments or strategy or implementation should be considered also. For the case of LULC, this would really be an important variable since it can have an impact on drought. If any significant trends are found, how can we attribute them arbitrarily to CO2 or to a mix of the influence of GHGs and aerosols? If there has been progressive changes in LULC in the area, in the context of this manuscript, eluding them would be really misleading for the results of this analysis.

a.2- Regarding internal predictors, the NAO is argued to be important but has not been used. Even if it has been described in previous works, it is relevant to see in

this approach how much variability do ENSO or PDO account for from the residuals once the NAO has been taken into account. Do the results of the analysis concerning the presently used internal predictors change if the NAO index is used? There are some millennium long index reconstructions that would allow for this exercise. I think there is no point in looking only at Pacific indices without considering a potential larger explanatory variable like this one.

b) Methodological issues There are three ideas that I would like to bring here. One is the linear vs non-linear character of the influences that the paper tries to assess. Another one is the power of the approach used herein related to the covariance structure pursued by the analysis in view of the properties of the predictors. Finally, and related, the collinearity of some predictors.

b.1 Regarding the first one, this is commented in the first paragraph of Sec. 3. I have no reservations against the possibility of nonlinear interactions being relevant. I think it is though important and has value, to study the linear relationships. It is also important to study it in a solid way so that we minimize the danger to loosely argue that everything we cannot explain with a linear approach is due to the limited character of its 'linearity' and probably due to nonlinear interactions.

b.2 Regarding the second one, the multiple linear regression is a valid approach to analyze the linear covariance structure in the data. Now, for that purpose, the variables used as predictors like CO2 or, for that case, if additional GHGs+aerosols would (and should) be considered, since these variables present very low variability at high and mid frequencies, one has to be careful in how to handle them in terms of covariance. For instance, a positive coefficient with temperature in the instrumental period means that both temperature and CO2 show positive trends... but any variable showing a positive or a negative trend would show association for that matter. The limited meaning of correlating preindustrial CO2 (+GHGs+aerosols) must be commented and the limited interpretation of correlating trends in the industrial period also should be argued and improved by including other GHGs and aerosols in a meaningful way.

b.3 Some of the predictors (eg. AMO, PDO) show covariability. How is this addressed in the analysis and how does this influence the results? Explaining which type of multiple regression approach would be important for this point.

c) Residuals This is also a rather methodological issue. If the purpose is to statistically describe drought with a multiple linear regression approach, the behavior of the residuals should be discussed. The authors should show estimation of drought variability from the predictor variables, explained variances and some convincing arguments that part of the variability is being reproduced by the predictors used.

I recognize this point, GC1, is rather long. It should probably be treated as independent points. Nevertheless I think it is important and would like to see the arguments for all these. Some specific comments will also follow below.

GC2 Mechanisms As it stands, the approach of the manuscript is to argue on the basis of the regression coefficients. This is quite extreme in its present state. Even in the discussion part, a relatively aseptic account of the results of other authors are provided in this sense. However a more mechanistic based approximation discussing the rationale behind the statistical relationships that may be found is needed. GC3 Temperature and precipitation What does having temperature and precipitation add in this analysis? I don't mean to be unconstructive. . . just that if it is included in the analysis pursuing a more in depth understanding of drought, the reader should understand why they are there. What gain in our understanding do we get from including temperature and precip and analyzing them as predictors? Would it be of use including them in one exercise as predictands and assess their relative influence on drought?

GC4 Section 5: PCA analysis The strategy for the PCA analysis in page12 should be described (already in the methods section), as well as it is purpose and results presented in the text. . . unless the results are rendered invalid or not useful. If the analysis provides some valuable insights within this ms, it should be shown.

GC5 Section 5: discussion The Discussion section provides a wealth of information

on different results from various papers. However, in my opinion it misses a bit some purpose or direction. Actually, it also reports on results (e.g. GC4) that are not shown although they permeate to the conclusions. This is not recommendable. I suggest to pass any results clearly to the parts of the paper to make clear the objectives, methods and analysis of the results. Having a Discussion part or a Conclusion and Discussion makes sense to put the results of the present ms in view of past literature and state clearly what we learn from it. I would advise the authors to modify this section in this sense.

Specific comments

SC1 Title: '. . . large-scale climate drivers' If we understand 'large-scale climate drivers' referring to modes of circulation, shouldn't the title also include those? E.g. ' Long-term variability of droughts in the Czech Lands due to external forcing and large scale climate drivers' ?

SC2 Page 2, l 17: 'Internal forcings' I think the use of this concept is not adequate in the manuscript. We relate forcing factors to changes in the energy of the system and, therefore, external in nature. I agree with using the terms internal/external drivers or external forcings, but not internal forcings.

SC3 Page 3, l 31: 'Missing monthly precipitation figures. . .' I don't understand what is meant here by 'missing' figures in Dobrovolny et al (2015).

SC4 Section 2: Figure 1 I haven't found a reference to Figure 1 in the ms. Check on this please. Regarding this figure and the presentation of drought, in Section 2 there is some description of differences in definitions among the different drought indices used in the text. I think some comment on the available reconstructions are pertinent. There is a paragraph in page 2 (l 21-30) describing the origin of the series. Can the authors provide any thoughts on whether the different definitions really play a role or basically the same information is available, also considering the source data for the reconstructions. Can we anticipate any added value of using these three indices instead of one

in this work?

SC5 Section 2.2: forcings I think it is desirable to place this forcing in the context of PMIP3 and PMIP4 forcings. The authors will find longer reconstructions of this forcing spanning the millennium that have been used to detect solar forcing on temperatures for instance since the 14th century (Schurer et al 2013). Maybe these can be better options for predictors than the one used in the ms (1610-present) Schurer, A., G. Hegerl, M. E. Mann, S. F. B. Tett, and S. J. Phipps, 2013: Separating forced from chaotic climate variability over the past millennium. J. Clim., doi:10.1175/JCLI-D-12-00826.1.

SC6 Section 5 L 30 '... the increase in the ambient $CO_2$ post 1850 is clearly correlated with the increased probability of drought... while during the pre-instrumental period such link does not manifest. The trend in $CO_2$ in the industrial period is just one degree of freedom. Please recall the comments GC1b

Technical corrections, typing errors,etc:

TT1 Page 6, l 8: '... results... standardized regression coefficients...' In an simple regression these would be, by definition, correlation coefficients. How does this differ in this analysis from correlations? Some methodological details on the multiple regression approach taken is advisable. Does it account for covariability in the predictors? Etc... Please provide more explanation of the relevant aspects in the ms.

TT2 Page 2, l 20: '... that increase...' substitute by '... that the increase...' This is just an example. I have found a few of those. I think the text is easy to understand in general. However, I would recommend it would be revised for editing/english

---

## Author Comment (AC1) · 22 Aug 2018

We would like to thank all three anonymous referees for their valuable comments regarding our manuscript. We tried to incorporate the corresponding modifications into our analysis and its presentation in the revised paper, or bring arguments in cases when we were unsure how the proposals could be implemented into the manuscript without too severe changes of its context or aim.

Major proposed changes to the analysis and its presentation in the revised manuscript:

- Results involving effect of the North Atlantic Oscillation (NAO) will be added in a quantitative form (whereas in the original manuscript, only a brief mention of NAO effects was made in the text).
- More reconstructions will be employed to represent the AMO index (2 versions in total) and PDO index (3 versions in total) and comparison with the previous results based on the reconstruction by Mann et al. (2009) will be provided. Predictor representing radiative forcing due to changes in the atmospheric composition will be altered to involve the aggregate effect of multiple greenhouse gases rather than just carbon dioxide. The solar variability predictor will be replaced by the recently published Total Solar Irradiance (TSI) data by Lean (2018; DOI 10.1002/2017EA000357), covering the entire 1501-2006 period.
- Discussion of the (potential) links between predictors will be expanded, in relation to the effects of external forcings on the activity of internal climate variability modes, as well as regarding mutual interactions of individual internal variability modes. Note, however, that these topics are extensive and in some cases (such as the causes, effects and interconnections of the Atlantic Multidecadal Oscillation) still intensely debated and far from concluded. Our manuscript can therefore only provide a limited overview of the related matters. Similarly, discussion of the physical mechanisms linking the drought indices to the explanatory variables will be extended.
- Additional results will be presented to better illustrate the properties of the regression mappings, including graphs of regression-estimated components associated with individual predictors and representations of regression residuals.
- Electronic Supplement will be added to the manuscript to hold extra materials; some of the illustrations originally in the main manuscript may be moved to the Supplement.

Please see below for specific responses to the comments of referees 1, 2 and 3 (R1, R2, R3) and the suggested changes to the manuscript.

**Anonymous Referee #1**

The paper analyses the long-term variability of droughts in the Czech lands based on long reconstructions (based on instrumental and documentary data). Time series of drought indices, temperature and precipitation are compared to reconstructions or time series of suspected drivers such as external forcings and oceanic variability modes. Anthropogenic radiative forcing as well as AMO/PDO are identified as influencing factors. The paper is interesting, valuable to the community and within the scope of Climate of the Past. However, I have several comments, which I think the authors should consider, before the paper can be published.

Methods

It is not fully clear which data are monthly or seasonal. Often the text mentions "seasonal and annual" or "monthly, seasonal and annual", which I found confusing. Also, the drought indices are usually calculated for individual measurement locations or grid cells. Here they are calculated for a large-scale average, as I understand. This should be made clear and explained. In the results section it then becomes clear that the seasons are analysed separately. However, what is the motivation for analysing a autumn or winter drought index?

> **Response R1-1:** The data description (Sect. 2.1) will be modified to make it more clear that the analysis was carried out on either series of annual values (i.e., consecutive values representing means for an entire year) or series of consecutive season-specific means (i.e., one seasonal value for each year). The note regarding monthly series in Sect. 2 pertains to some of the original data sources; monthly values were not directly studied in the current analysis. The nature of the drought indices as area-wide means will be more explicitly stated in the text (Sect. 2.1). Since drought data for all seasons (including fall and winter) were available and analyzed, we present the outcomes for all four seasons, to illustrate the full range of potential climate links, even though for some applications (such as investigation of agricultural droughts) spring and summer conditions may be of greater interest. It should also be considered that recharge of the underground water resources and surface reservoirs depends on the water available during fall and winter and droughts in these periods often induce major hydrological impacts in the following year.

Multiple linear regression is used to separate individual components, but fully separating external forcing from internal variability (e.g., oceanic modes) is fundamentally difficult. External forcings might operate via altering internal variability modes (e.g. solar and volcanic forcing might change the climate system via AMO or ENSO). Conversely, AMO and PDO have the imprint of global temperature rise. I see that the authors use cross-wavelet spectra, partly to assess the interdependencies, but not systematically. Partial correlation methods could be used to go into more depth here, or different models could be compared. In any case, the interpretations should be phrased very carefully.

> **Response R1-2:** Indeed, the problem of separating the strictly external forcings from the internally induced variability is a complicated one, not only at a statistical level, but also with regard to the underlying physical mechanisms. While this was not mentioned in the manuscript, we examined the mutual links between individual predictors with episodic or oscillatory components in terms of Pearson correlation and its time-windowed version.

Although some potentially noteworthy correlations appeared, none of them (other than the AMO-PDO relation) seemed strong and stable enough to warrant a specific treatment of inter-predictor links, at least not in the context of purely linear regression. Therefore, in our analysis eventual external forcing-induced components in the indices of internal climate variability modes were treated as a part of these indices; components attributed by the regression analysis to the forcings themselves were then treated as direct responses. To provide a more complete picture of the potential indirect effects of external forcings manifesting through their influence on the internal climate variability modes, results of analysis carried out with just the predictors representing external forcings (solar, volcanic, anthropogenic) will be added to the revised manuscript. Furthermore, the Discussion will be expanded to provide additional references to works addressing the influence of external forcings on the relevant internal climate variability modes.

In the case of the imprints of global temperature in the AMO and PDO (pseudo)indices, please note that the long-term temperature component (in the form of mean northern hemispheric temperature) has been removed from the data during pre-processing (as described in Sect. 2.2), and the AMO/PDO predictors therefore only encompass oscillatory variations around the hemispheric temperature series. This will be highlighted in the revised version of the text.

Due to the sheer amount of possible combinations, results of the cross-wavelet analysis were only presented for selected pairs of predictors/predictands, either those showing interesting interactions, or those intended to illustrate a similarity or contrast in behavior compared to some other pair of variables.

While we agree that partial correlations can offer additional insight into the interdependencies in a multivariable system, their use does not necessarily solve the ambiguity arising from the existence of a common, physically relevant component within multiple explanatory variables, stemming not from a one-way causality, but rather from a two-or-more-way interaction. Such a component cannot be reliably assigned by purely statistical means and since its origin is typically rather complex, we prefer to deal with its presence and interpretation during the discussion of the results. Note also that in the most prominent case of such collinearity in our analysis, related to the similarity between AMO/PDO predictors by Mann et al. (2009), we addressed this problem by application of a simple version of principal component analysis, to provide a more complete interpretation of the role of individual predictors and the components within.

The regression model itself is not explained clearly. From the text it becomes clear that different ENSO indices were used, but which model (which ENSO index) is the one shown in Figs. 4 and 5? Furthermore, only very late in the paper we learn that the explained variances are very low, below 5%. Should we even analyse regression models that have no explanatory power? Finally, the effect of reconstruction uncertainty is not discussed.

**Response R1-3:** The missing identification of the primary ENSO index will be corrected – it will be explicitly stated that the results in Figs. 4 and 5 are based on the ENSO reconstruction by Li et al. (2011), while the results for the ENSO data by Mann et al. (2009) are only mentioned in the main text.

The seemingly low fraction of variance explained by the regression models ($R^2$) is a result of dominance of inter-annual variability in the predictand series, matched in the

regression mapping against predictors mostly dominated by inter-decadal variation. Formally, higher $R^2$ could be achieved by removing the year-to-year variations, e.g. by smoothing the series by a moving average filter (to give an example, for the period 1501-2006, 21% of variability of the annual SPEI series can be explained by the regression model if the series are smoothed by 11-year moving average; this value increases to 33% when the NAO reconstruction by Luterbacher et al. (2002) is also included as a predictor). However, since some of our explanatory factors (the episodic volcanic activity, 11-year cycle in the solar variability signal, and the NAO index in the revised version of the analysis) do exhibit faster variability, which would be largely erased by the smoothing, we prefer to perform the analysis with the unaltered series. The prominence of individual explanatory factors is evaluated through statistical significance of the respective regression coefficients, regardless of the overall $R^2$ – an approach that we believe to be consistent with our primary aim, i.e. identification of forcings and large-scale factors influential in establishing the drought regime of the Czech Lands (as opposed to an attempt to construct a predictive model reproducing the series with as much variability as possible). To better illustrate the actual magnitude of components associated with individual explanatory factors, time series of regression-generated components corresponding to individual explanatory variables will be included in the Supplement accompanying the revised version of the manuscript.

The effect of uncertainties tied to the results would be rather difficult to quantify reliably, as not all series analyzed come with an uncertainty estimate, and methods of its estimation differ even when such data exist. However, due to increased number of versions of some predictors in the revised version of the analysis, more attention will be paid in the revised manuscript to the robustness of the results based on different reconstruction sources.

The paper says little about the mechanisms linking the external and internal drivers to drought and hydroclimatic conditions in general. Obviously a study using reconstructions cannot explicitly address net radiation, soil moisture, temperature effects, land-surface feedbacks, atmospheric circulation effects (blocking), etc. But it would be nice to read the authors' hypotheses. The paper is rather silent about mechanisms. In the introduction Hess-Brezowsky weather types are mentioned, and later the NAO, but the NAO is not incorporated into the analyses and the discussion parts then follows another thread: Doing a PC analysis of AMO/PDO. It would be nice if the Discussion section could come back to mechanisms at some point.

**Response R1-4:** Note please that our study is dealing with droughts defined through the SPI/SPEI/PDSI indices, shaped by (and calculated from) precipitation and temperature series. Our interpretation of the possible links is therefore focused on the role and eventual interaction of the temperature and precipitation variability in establishing the central European drought regime expressed by the above indices. Also note that some of the responses, while statistically significant, represent rather minor tendencies, difficult to reliably assign to specific mechanisms (especially in our analysis involving pre-instrumental period, as no data exist consistently capturing global large-scale circulation over the last five centuries, making it difficult to evaluate influences related to circulation, blocking, etc.). Even so, we will try to provide more detailed discussion of the possible mechanisms in the revised text, along with additional relevant references (please see also our Response R3-7).

NAO-related effects will be included in the revised version of the manuscript, based on the NAO index reconstruction by Luterbacher et al. (2002) and multidecadal NAO variability reconstruction by Trouet et al. (2009; DOI: 10.1126/science.1166349). The results in the current Figs. 4 and 5 will be updated to show regression coefficients related to NAO in addition to the previously considered predictors; relation between NAO and external forcings (especially volcanic) will be documented and discussed.

Minor comments

Abstract, l. 14: "external and internal climate forcings". Please be careful with terminology here and elsewhere. Considering the coupled climate system "forcing" is used for external influences (subdivided into natural and anthropogenic) while internal variability is used for the dynamics of the coupled system even if unforced. When considering only the atmosphere, "oceanic forcing" is sometimes used. In any case, the terms should be defined and used consistently.

**Response R1-5:** Terminology in the manuscript will be modified to avoid use of the term 'forcing' for factors originating from internal climate dynamics.

P. 2, L. 5: a substantial number of studies: cite
P. 2, L. 5: A lot of work has been done on droughts in the USA. Perhaps before zooming in on Europe, you could mention that.

**Response R1-6:** Both comments accepted, corrected by adding new references as follows: "In addition to a substantial number of studies investigating drought indices for the instrumental period in Europe (e.g. van der Schrier et al., 2007, 2013; Briffa et al., 2009; Sousa et al., 2011; Todd et al., 2013; Spinoni et al., 2015; Haslinger and Blöschl, 2017) and other areas of the world (e.g. Dai, 2011; Spinoni et al., 2014; Ryne and Forest, 2016; Wilhite and Pulwarty, 2018), generally calculated …".
**Added references:**
Briffa, K. R., van der Schrier, G., and Jones, P. D.: Wet and dry summers in Europe since 1750: evidence of increasing drought, Int. J. Climatol., 29, 1894–1905, doi: 10.1002/joc.1836, 2009.

Dai, A.: Characteristics and trends in various forms of the Palmer Drought Severity Index (PDSI) during 1900–2008, J. Geophys. Res., 116, D12115, doi: 10.1029/2010JD015541, 2011.

Haslinger, K. and Blöschl, G.: Space-time patterns of meteorological drought events in the European Greater Alpine Region over the past 210 year, Water Resour. Res., 53, 9807–9823, doi: 10.1002/2017WR020797, 2017.

Ryne, S. and Forest, K.: Evidence for increasing variable Palmer Drought Severity Index in the United States since 1895, Sci. Tot. Env., 544, 792–796, doi: 10.1016/j.scitotenvv.2015.11.167.

Sousa, P. M., Trigo, R. M., Aizpurua, P., Nieto, R., Gimeno, L., and García-Herrera, R.: Trends and extremes of drought indices of drought indices throughout the 20th century in the Mediterranean, Nat. Hazards Earth Syst. Sci., 11, 33–51, doi: 10.5194/nhess-11-33-2011, 2011.

Spinoni, J., Naumann, G., Carrao, H., Barbosa, P., and Vogt, J.: World drought frequency, duration, and severity for 1951–2010. Int. J. Climatol., 34, 2792–2804, doi: 10.1002/joc.3875, 2014.

Spinoni, J., Naumann, G., Vogt, J., and Barbosa, P.: European drought climatologies and trends based on a multi-indicator approach. Glob. Plan. Change, 127, 50–57, doi: 10.1016/j.gloplacha.2015.01.012, 2015.

Todd, B., Macdonald, N., Chiverrell, R. C., Caminade, C., and Hooke, J. M.: Severity, duration and frequency of drought in SE England from 1697 to 2011. Clim. Change, 121, 673–687, doi: 10.1007/s10584-013-0970-6, 2013.

van der Schrier, G., Barichivich, J., Briffa, K. R., and Jones, P. D.: A scPDSI-based global dataset of dry and wet spells for 1901–2009. J. Geoph. Res., 118, 4025–4048, doi: 10.1002/jgrd.50355, 2013.

van der Schrier, G., Efthymiadis, D., Briffa, K. R., and Jones, P. D.: European Alpine moisture variability 1800–2003. Int. J. Climatol., 27, 415–427, 10.1002/joc.1411, 2007.

Wilhite, D. A. and Pulwarty, R. S.: Drought as hazard: Understanding the natural and social context. In: Wilhite, D. A. and Pulwarty, R. S., eds.: Drought and Water Crises. Integrating Science, Management, and Policy. CRC Press, Taylor & Francis Group, Boca Bayton, 3–20, 2018.

P. 2, L. 24: instrumental precipitation series

**Response R1-7:** Corrected as "The authors demonstrated the importance …"

P. 3, L. 10: What time window was used for the SPI?

**Response R1-8:** The time window used for the SPI/SPEI calculation was chosen to match the type of the series used as predictand, i.e. 12 months for the annual data, 3 months for seasonal data.

P. 4, L. 3 and 4: I do not understand this sentence.

**Response R1-9:** The sentence will be changed to: 'As a result, a 100-member ensemble of distributions of monthly precipitation totals for each season and the year was obtained. These distributions were then applied for calculation of indices for every year in the 1501–1803 period.'. Note, please, that this is just a substantially simplified description of the data preparation process, and full explanation can be found in Brázdil et al. (2016a).

P. 4, L. 10: "climate forming agents": rephrase

**Response R1-10:** Reformulated to 'climate-defining factors'

P. 4, L. 19 and 20: Omit the first part of the sentence, which is unnecessary. Start with "A large part..."

**Response R1-11:** Accepted

P. 4, L. 26: strong clear?

**Response R1-12:** Changed to 'clear'

P. 5, L. 1-3: Perhaps cite Fischer et al. GRL (https://doi.org/10.1029/2006GL027992)

**Response R1-13:** The reference to Fisher et al. (2007; DOI 10.1029/2006GL027992) will be added to the revised manuscript.

P. 5, L. 19: I am a bit puzzled why the authors use the Mann et al. ENSO series. As the authors write (and other authors have also pointed to that), the reconstruction varies mostly on the 8-20 year time scale. Why use it as an ENSO time series then? I would rather use other ENSO reconstructions. Similarly, for AMO and PDO it would be nice to have two indices for each (e.g. Shen et al. 2006 for the PDO, Gray et al. 2004 for the AMO).

**Response R1-14:** Please note that ENSO reconstruction by Li et al. (2011) was used as the primary descriptor of ENSO phase, and a basis for the results shown in Figs. 4 and 5. ENSO index derived from the Mann et al. (2009) data was only used as an alternative ENSO descriptor and was not found to be associated with any the statistically responses in the drought data. This will be more explicitly stated in the revised text; regression outcomes for Mann et al. ENSO data will be included in the Supplement.

Results based on the PDO reconstruction by MacDonald and Case (2005; DOI 10.1029/2005GL022478) and Shen et al. (2006; DOI 10.1029/2005GL024804) and AMO reconstruction by Gray et al. (2004; DOI 10.1029/2004GL019932) will be included in the revised manuscript and discussed along with the outcomes of the analysis utilizing the originally employed PDO/AMO data by Mann et al. (2009). Regression coefficients will be presented for each version of the predictors; their similarity (or lack thereof, as is the case for the PDO reconstructions) will be discussed with regard to the robustness of the results and the associated uncertainties.

P. 10, L. 34: Is the tendency for wet conditions after volcanic eruption really due to lower temperatures?

**Response R1-15:** This formulation is meant to reflect the fact that the tendency towards higher values of the drought indices during periods with higher amounts of volcanic aerosol coincides with significant drop of temperature, while no statistically significant change in precipitation is indicated.

P. 11, L. 34: I am surprised that Sutton and Hodson (2005) paper is not mentioned in context with the AMO effect.

**Response R1-16:** The reference to Sutton and Hodson (2005; DOI 10.1126/science.1109496) will be added to the revised manuscript and discussed.

**Anonymous Referee #2**

GENERAL COMMMENTS:

Manuscript under revision is an approach to study of drought in Czech Republic area, taking in consideration previous climatic reconstructions, already published, using these informations to generate drought indices based on instrumental records. Analysis of possible relations with different forcing factors is also made to offer general or initial explanation to drougth mechanisms for this Central Europe study area.

Main effort focused to compare rainfall indices and temperatures for long or complete periods. It's a good first approach to drougth phenomena. It open research to study specific events at higher temporal resolution, impacts and responses, etc.

SPECIFIC COMMENTS

+ Title could include temporal dimension of work of manuscript.

> **Response R2-1:** Because we are using the expression "long-term", it is probably not necessary to extend the title for the time span used.

+ Title. Expression "drought" into title is excessively general. A more correct definition of topic developped into manuscript would be "drought indices".

> **Response R2-2:** Accepted and also with respect to a comment of Referee 3 changed to "Long-term variability of drought indices in the Czech Lands and effects of external forcings and large-scale climate variability modes"

+ Lines 28-30. Seasonal and annual precipitation for 1501-1854 is reconstructed from "document-based precipitation indices". Dobrovolvy et al., 2015. Could explain in a short description general characteristics or contents of these "documents"? How was developed previous analysis. Just to have a connection between original information and present results generated into manuscript. IF drought is analyzed, at least public must know about historical documents used for reconstruction, temporal resolution of information obtained, locations or regions with available information, aspects of natural process and/or and human impacts detected/evaluated....etc.... I understand manuscript can have restrictions of extension, but this short overview would be useful for public.

> **Response R2-3:** This comment and several following remarks of Referee 2 concern details related to the documentary data used. We would like to stress that the primary aim of the analysis is the study of forcings and large-scale climate drivers reflected in series of drought indices, described in detail in the paper by Brázdil et al. (2016a). Because their calculations are based on reconstructed temperature (Dobrovolný et al., 2010) and precipitation (Dobrovolný et al., 2015) series, in which a detailed information of documentary data used with their types, examples and critical evaluation are given (as well as the reconstruction uncertainties), it would bring not too much new information to the merit of this article. But looking at the comments of the Referee 2, we included several additional sentences in this direction with hope to fulfill at least partly these requests by the change of the fifth paragraph of Section 2.1 as follows:

"Long-term seasonal and annual series of these three indices, dating from AD 1501 in the Czech Lands (Brázdil et al., 2016a) were used in the current study. They were derived from 500-year temperature and precipitation reconstructions based on a combination of documentary data and instrumental measurements. Documentary data were represented by descriptions of weather and related phenomena coming from different documentary evidence of individual as well as institutional character as annals, chronicles and memories, weather diaries (non-instrumental observations), financial and economic records, religious records, newspaper and journals, epigraphic sources etc. Corresponding data in the Czech Lands cover with a different density particularly the period from AD 1501 to the mid-19th century, but they continue even to the recent time. Also the spatial density of such data is changing with time depending on the availability and extraction of existing documentary sources over the Czech territory. All collected data were critically evaluated with respect to possible errors in dating or spatial attribution and were used for interpretation of monthly weighted temperature and precipitation indices in the 7-degree scale, from which series of seasonal and annual indices were created (for more details of the use of documentary data, their critics, analysis and interpretation as well as creation of indices series in historical climatology see Brázdil et al., 2005, 2010). Such data were further used as a basic tool for temperature/precipitation reconstructions. Firstly, Dobrovolný et al. (2010) reconstructed monthly, seasonal and annual central European temperature series, partly based on temperature indices derived from documentary data for Germany, Switzerland and the Czech Lands in the 1501–1854 period and partly on homogenized instrumental temperature series from 11 meteorological stations in central Europe (Germany, Austria, Switzerland, Bohemia) from 1760 onwards. This temperature series is fully representative of the Czech Lands. Subsequently, seasonal and annual precipitation series for the Czech Lands were reconstructed from documentary-based precipitation indices in the 1501–1854 period and from mean precipitation series calculated from measured precipitation totals in the Czech Lands after 1804 (Dobrovolný et al., 2015)."

+ Bibliography used on work is complete and well updated.

**Response R2-4:** Thank you.

+ Effort to offer a background or general overview about drought events is not so complete as we would like find. For example, justification of study of drought. It's a present or potential problem for Czech Republic?, any previous strong event justify this study? How they are drougth conditions in Czech Republic?

**Response R2-5:** To fulfill this comment, the first paragraph of Introduction will be changed as follows: "Droughts, among the most prominent manifestations of extreme weather and climate anomalies, are not only of great climatological interest but also constitute an essential factor to be considered in the assessment of the impacts of climate change (Stocker et al., 2013; Trnka et al., 2018; Wilhite and Pulwarty, 2018). This is valid also for the territory of the Czech Republic where droughts, besides floods, represent the most important natural disaster with significant impacts in different national economy sectors as, for example, agriculture, forestry, water management, or tourism/recreation. Since the Czech Republic lays on a continental divide with rivers flowing out of its territory, it depends on just the

atmospheric precipitation for its water supply. Although we know some extreme droughts with important socio-economic and political impacts from the past, such as drought of 1947 (Brázdil et al., 2016c), the studies from the recent years show increasing dryness of the Czech climate in the past 2-3 decades expressed in higher frequency of extreme droughts with significant consequences (e.g. Brázdil et al., 2015b; Zahradníček et al., 2015). The abundance of long-term instrumental meteorological observations in the European area has provided a basis for a number of recent drought-focused studies, revealing complex regional drought patterns and a richness of features observed at various spatial and temporal scales (e.g., van der Schrier, 2006, 2007; Brázdil et al., 2009; Briffa et al., 2009; Dubrovský et al., 2009; Sousa et al., 2011; Spinoni et al., 2015). Along with more rapid variations, these also include long-term variability, such as a distinct trend towards drier conditions, prominent especially during the late 20th and early 21st centuries (e.g., Trnka et al., 2009a; Brázdil et al., 2015b). "

**Added references:**

Brázdil, R., Raška, P., Trnka, M., Zahradníček, P., Valášek, H., Dobrovolný, P., Řezníčková, L., Treml, P., Stachoň, Z.: The Central European drought of 1947: causes and consequences, with particular reference to the Czech Lands. Climate Research, 70, 161–178, doi: 10.3354/cr01387, 2016c.

Zahradníček, P., Trnka, M., Brázdil, R., Možný, M., Štěpánek, P., Hlavinka, P., Žalud, Z., Malý, A., Semerádová, D., Dobrovolný, P., Dubrovský, M., Řezníčková, L.: The extreme drought episode of August 2011–May 2012 in the Czech Republic. International Journal of Climatology, 35, 3335–3352. DOI: 10.1002/joc.4211, 2015.

+ Historial dimension of drought is not analyzed. Just index values from previous research considered as approach to climatic patterns related to low values of reconstructed precipitation. Drought is not studied by itself as climatic/historic phenomena. This aspect is not negative nor positive. Just it would require any extension of explanations about drougth as climatic pehnomena in introduction of work.

> **Response R2-6:** As explained above, our manuscript concentrates on the explanation of effects of external forcings and large-scale climate drivers on long-term drought indices variability in the Czech Lands. This means that we are really not analyzing "historial dimension of drought" as the referee correctly states because it does not fit to the concept of this paper.

+ No specific drought events are mentioned. No description at least for one event is included into manuscript. Complexity of drought events and related impacts is not described/evaluated. May be authors are preparing other papers with these specific topics?

> **Response R2-7:** The description of any "specific drought event" does not fit to the paper context, analyzing rather effects of external forcings and large-scale climate drivers in long-term drought indices series. Descriptions of specific drought events in the Czech Lands can be found, for example, in Brázdil et al. (2013) or Brázdil et al. (2015b). Moreover, we are currently preparing a paper with working title "Extreme droughts and their human responses in the Czech Lands in the pre-instrumental period" for another journal.

+ No explanation about drought as climatic phenomena. How is considered drought in Czech Republic, what criteria are applied, what instrumental thresholds, duration/extension/severity,

different concepts/definitions of drought, affectation of agriculture.... Any explanation would be useful to understand characteristics and effects for public unknowning these specific details.

> **Response R2-8:** We would like to stress that we are not concentrating in this paper primarily on "drought as climatic phenomena" or "what criteria are applied, what instrumental thresholds, duration/extension/severity, different concepts/definitions of drought, affectation of agriculture", because it was analysed already in many other papers related to the territory of the Czech Republic (for comprehensive overview see e.g. Brázdil et al., 2015b). We are just trying to find how fluctuations in series of drought indices in the Czech Lands are influenced by external forcings or large-scale climate drivers.

+ If drought is defined only from specific indices (SPI, SPEI...), when we work in historical time, out of instrumental data availability, this topic must be taken with more introductory explanations. A more complete and informative approach about how documentary records detect and define droughts, what they record, what transmit....

> **Response R2-9:** We would like to stress that drought indices are not primarily derived (calculated) from documentary data, but from temperature/precipitation reconstructions based on documentary-based indices series and overlapping instrumental series. For this reason we are of the opinion that comment "A more complete and informative approach about how documentary records detect and define droughts, what they record, what transmit...." could be difficult to follow in the recent concept of our paper.

+ If manuscript is based on previous reconstructions, focused on reconstructed values of mm. rainfall, by total monthly/seasonal/annual resolution, authors must consider they cannot analyze all dimension of droughts. Rainfall indices with positive aspect can cover important drought events, when dry periods are interrupted by strong rainfall events. Knowing what tipe of drougth is under study, these singular aspects could be differenced, generating a better and deeper study.

> **Response R2-10:** We agree with the opinion of referee 2 but we are not analyzing drought on the base of precipitation indices. Precipitation reconstruction was used only as one of two basic series which were used to calculate series of drought indices.

+ Manuscript doesn't show a clear relation of type of documents and type of information rescued and anallyzed.

> **Response R2-11:** As mentioned in Section 2.1, we analyse effects of external forcings and large-scale climate drivers in long-term variability of drought indices series, calculated from reconstructed series of temperatures (Dobrovolný et al., 2010) and precipitation (Dobrovolný et al., 2015). Both these papers contain detailed information about types of documents and information rescued and analysed. Calculation of drought indices was explained in detail in the paper by Brázdil et al. (2016a). From these reasons we do not see as necessary to repeat in detail all these aspects in the present paper.

+ It would be interesting focuse efforts on variability and extreme events of the same variable before to compare with variability of others proxys.

**Response R2-12:** Aspects reported by the referee (variability, extreme events, …) were dealt in a great detail already in the paper by Brázdil et al. (2016a). We feel it redundant to repeat it here again because it does not fit to the context of the present article.

**Anonymous Referee #3**

Description

This paper addresses the understanding of the variability of droughts, temperature and precipitation in the Czech lands 1500-present from the point of view of its dependence on internal drivers (e.g. some specific modes of circulation) and also external forcing factors (volcanic, CO2, etc). For that purpose a multiple linear regression is applied having as predictand variables 3 drought indices and as predictors a set of external and internal drivers.

The purpose of the paper has value and meaningful and solid results in this direction would be worth to be published in CP. If attribution of drought variability or a meaningful step forward in its understanding in the Czech Lands would be attained I think this would be sufficiently valuable in my view to support publication. Therefore, I encourage authors to pursue this line of work towards publication. At this state, I would recommend major revision of the manuscript. There are several issues related to the rationale, methodology and description and interpretation of results that in my understanding require revision. I will argue about this in the following points.

General Comments

GC1 General approach to attribution As it is described in the paper, 5 predictand series (3 drought indices and a temperature and a precipitation series) are examined using multiple linear regression as functions of independent predictors, the latter being internal and external in nature. In practice, these are 5 individual multiple regressions.

Having that in mind I would suggest to consider the analysis, description and discussion of the: a) selected predictors; b) of the methodological approach; and c) of the residuals of the methodology.

a) Selected predictors. I would argue these are insufficient in both the case of the external and internal subsets.

a.1-Regarding external predictors I have no objections to the ones considered so far but the authors should discuss why important predictors like other greenhouse gases (GHGs), aerosols and particularly land use land cover (LULC) are not considered. For the case of other GHGs than CO2, it would be more elegant either to consider them or to use equivalent CO2. For the case of aerosols some arguments or strategy or implementation should be considered also. For the case of LULC, this would really be an important variable since it can have an impact on drought. If any significant trends are found, how can we attribute them arbitrarily to CO2 or to a mix of the influence of GHGs and aerosols? If there has been progressive changes in LULC in the area, in the context of this manuscript, eluding them would be really misleading for the results of this analysis.

> **Response R3-1:** It is true that using just $CO_2$ concentration as an approximation of anthropogenic influence oversimplifies the setup. In the revised version of the analysis, aggregate radiative forcing of multiple GHGs (including $CO_2$, $CH_4$ and $N_2O$) will therefore be used instead. As for the inclusion of the effects of (tropospheric) aerosols, their regional effect is difficult to consider in an analysis such as ours, due to high temporal and spatial variability of their concentrations, differences in behavior of different aerosol species and still high uncertainties regarding their effects. Note also that from a standpoint of a

regression analysis, the predictors with and without the aerosol forcing are usually almost identical, as the respective time series are very strongly correlated. For instance, using the Meinshausen et al. (2011; DOI 10.1007/s10584-011-0156-z) global annual concentration and forcing data over the 1765-2005 period, the $CO_2$ concentration series is correlated with total anthropogenic forcing (representing the aggregated effect of various greenhouse gases as well as aerosols) at 0.995. There would therefore be almost no change of the regression results if different versions of the predictor representing anthropogenic forcing were applied (despite the obvious differences in the physical effects involved). This will be mentioned in the revised manuscript, along with a more detailed discussion of the correlation between GHG forcing and the drought indices and the caveats of its interpretation.

Regarding the Land use land cover (LULC): We are working with drought indices for the whole Czech Lands calculated from reconstructed temperature and precipitation series. The calculation procedure of none of these indices includes information about LULC. Although it can be an important factor deciding about drought severity and particularly its impacts, effects of LULC on country-wide temperature and precipitation should be limited. In this study oriented on long-term temporal changes it seems to be not an important factor helping us as predictor in the regression analysis of drought indices series.

a.2- Regarding internal predictors, the NAO is argued to be important but has not been used. Even if it has been described in previous works, it is relevant to see in this approach how much variability do ENSO or PDO account for from the residuals once the NAO has been taken into account. Do the results of the analysis concerning the presently used internal predictors change if the NAO index is used? There are some millennium long index reconstructions that would allow for this exercise. I think there is no point in looking only at Pacific indices without considering a potential larger explanatory variable like this one.

**Response R3-2:** Our original intention was to concentrate on mid-to-long-range variability in the drought series, i.e. oscillations typically slower than the dominant variability of NAO. Moreover, the strong relation between central European drought regime and NAO phase has been established by various prior studies, hence we considered it to be less interesting for the current analysis. Since both Referees 1 and 3 expressed their interest in the NAO-related effects, in the revised version of the paper, results involving NAO reconstructions by Luterbacher et al. (2002) and Trouet (2009; DOI 10.1126/science.1166349) will be included and discussed (please see also our Response R1-4).

b) Methodological issues There are three ideas that I would like to bring here. One is the linear vs non-linear character of the influences that the paper tries to assess. Another one is the power of the approach used herein related to the covariance structure pursued by the analysis in view of the properties of the predictors. Finally, and related, the collinearity of some predictors.

b.1 Regarding the first one, this is commented in the first paragraph of Sec. 3. I have no reservations against the possibility of nonlinear interactions being relevant. I think it is though important and has value, to study the linear relationships. It is also important to study it in a solid way so that we minimize the danger to loosely argue that everything we cannot explain with a linear approach is due to the limited character of its 'linearity' and probably due to nonlinear interactions.

**Response R3-3:** We seem to be in agreement with the referee; the mention of nonlinear approach was meant to provide a methodological context while also giving rationale for using linear version of regression.

b.2 Regarding the second one, the multiple linear regression is a valid approach to analyze the linear covariance structure in the data. Now, for that purpose, the variables used as predictors like CO2 or, for that case, if additional GHGs+aerosols would (and should) be considered, since these variables present very low variability at high and mid frequencies, one has to be careful in how to handle them in terms of covariance. For instance, a positive coefficient with temperature in the instrumental period means that both temperature and CO2 show positive trends... but any variable showing a positive or a negative trend would show association for that matter. The limited meaning of correlating preindustrial CO2 (+GHGs+aerosols) must be commented and the limited interpretation of correlating trends in the industrial period also should be argued and improved by including other GHGs and aerosols in a meaningful way.

**Response R3-4:** This is definitely true, and admittedly under-explained in the original manuscript. The inclusion of a trend-like variable ($CO_2$ concentration in the original version of the manuscript, composite GHG forcing in the revised one) was meant to provide a predictor potentially approximating long-term evolution observed in the drought indices. Naturally, despite the similarity in shape (and thus statistical significance of the link detected for some of the drought indices), the formal relationship does not prove causal relation. While we commented on this in the original version of the text ('Even considering that statistical attribution analysis can only reveal formal similarities and cannot verify the causality of the links detected ...' in the Conclusions, page 13, lines 13-17), and referenced supporting evidence pointing to a physical link between droughts and anthropogenic forcing (the second paragraph of Discussion), the potential for mis-attribution will be more explicitly emphasized in the revised manuscript.

b.3 Some of the predictors (eg. AMO, PDO) show covariability. How is this addressed in the analysis and how does this influence the results? Explaining which type of multiple regression approach would be important for this point.

**Response R3-5:** For the AMO and PDO representations based on the Mann et al. temperature reconstruction, this was actually addressed (in the Discussion section) by employing a simple form of principal component analysis, allowing to better assess the role of the common component in the predictors and of their difference. In the revised manuscript, the respective results will be shown in more detail, including the graphs of the regression coefficients and cross-wavelet spectra illustrating the relation between the drought indices and the principal components.

Please note also that (multi)collinearity of the predictors results in increased variance inflation factor for the regression coefficients (and thus wider confidence intervals). Since this is an inherent feature of multivariable regression, we did not comment on it specifically; the effect can, however, be seen from Figs. 4 and 5 in the original manuscript.

c) Residuals This is also a rather methodological issue. If the purpose is to statistically describe drought with a multiple linear regression approach, the behavior of the residuals should be discussed. The authors should show estimation of drought variability from the predictor variables, explained variances and some convincing arguments that part of the variability is being reproduced by the predictors used.

I recognize this point, GC1, is rather long. It should probably be treated as independent points. Nevertheless I think it is important and would like to see the arguments for all these. Some specific comments will also follow below.

> **Response R3-6:** The analysis of regression residuals was performed when designing the optimum setup for the moving-block bootstrap. The only noteworthy feature (aside from the approximately AR1-consistent persistence structure) was a presence of a weak and rather unstable 22-year-period oscillation (possibly an imprint of the 22-year cycle in solar activity, but inconsistently present throughout our analysis period). This will be mentioned and discussed in the revised version. Graphs illustrating the residual variability will be included in the Supplement, along with residual wavelet spectra.
>
> As for the explained variances and evaluation of the regression results, additional results will be included and discussed in the revised manuscript, including graphs of regression-estimated components pertaining to individual predictors – please see also the second paragraph of our Response R1-3.

GC2 Mechanisms As it stands, the approach of the manuscript is to argue on the basis of the regression coefficients. This is quite extreme in its present state. Even in the discussion part, a relatively aseptic account of the results of other authors are provided in this sense. However a more mechanistic based approximation discussing the rationale behind the statistical relationships that may be found is needed.

> **Response R3-7:** In the revised version, we will pay more attention to the (potential) mechanisms behind the observed links. Note, however, that most of the connections highlighted in our analysis represent rather weak (albeit statistically significant) tendencies, which cannot be unambiguously assigned to specific mechanisms (especially considering that no observational data exist that could be used for analysis of circulation patterns over the period of the last five centuries, and that dynamical models are still rather unreliable in capturing some of the relevant factors, including the sources of multidecadal climate variability). Still, we will try to suggest plausible mechanisms that can be used as initial hypothesis that could be tested by follow up studies, e.g. in the regions (periods) with more comprehensive datasets.

GC3 Temperature and precipitation What does having temperature and precipitation add in this analysis? I don't mean to be unconstructive... just that if it is included in the analysis pursuing a more in depth understanding of drought, the reader should understand why they are there. What gain in our understanding do we get from including temperature and precip and analyzing them as predictors? Would it be of use including them in one exercise as predictands and assess their relative influence on drought?

**Response R3-8:** Temperature and precipitation data were used for calculation of the drought indices themselves (as explained in Sect. 2.1), and their behavior is therefore crucial when discussing their combined effect in the drought descriptors.

GC4 Section 5: PCA analysis The strategy for the PCA analysis in page12 should be described (already in the methods section), as well as it is purpose and results presented in the text... unless the results are rendered invalid or not useful. If the analysis provides some valuable insights within this ms, it should be shown.

**Response R3-9:** We did not mention PCA in the Methods section, as it was only used as a supporting technique in a small part of our analysis (and we assumed its general principles to be a common knowledge, thus not requiring introduction). In the revised version, use of PCA will be mentioned in the Methods section; the results based on analysis of principal components will be presented directly instead of just mentioned in the text (please see also Response R3-5).

GC5 Section 5: discussion The Discussion section provides a wealth of information on different results from various papers. However, in my opinion it misses a bit some purpose or direction. Actually, it also reports on results (e.g. GC4) that are not shown although they permeate to the conclusions. This is not recommendable. I suggest to pass any results clearly to the parts of the paper to make clear the objectives, methods and analysis of the results. Having a Discussion part or a Conclusion and Discussion makes sense to put the results of the present ms in view of past literature and state clearly what we learn from it. I would advise the authors to modify this section in this sense.

**Response R3-10:** The results originally just mentioned (but not shown) in the manuscript will now be included fully, either in the main paper or in its Supplement. The Introduction and Discussion will be modified to paint a clearer picture of our main objectives: to assess the existence of links between Czech drought indices and climate forcings or activity of large-scale internal variability modes, and to investigate the properties of the existing reconstructions (for both the target and explanatory variables), especially with regard to the uncertainties tied to these series in the pre-instrumental era. To this end, additional results and references will also be included in the revised manuscript (some of them detailed in Responses R1-4, R1-14 and R3-6).

Specific comments

SC1 Title: '... large-scale climate drivers' If we understand 'large-scale climate drivers' referring to modes of circulation, shouldn't the title also include those? E.g. ' Longterm variability of droughts in the Czech Lands due to external forcing and large scale climate drivers' ?

**Response R3-11:** Based on this comment and suggestions of Referee 1, we changed the title to "Long-term variability of drought indices in the Czech Lands and effects of external forcings and large-scale climate variability modes"

SC2 Page 2, l 17: 'Internal forcings' I think the use of this concept is not adequate in the manuscript. We relate forcing factors to changes in the energy of the system and, therefore, external in nature. I agree with using the terms internal/external drivers or external forcings, but not internal forcings.

> **Response R3-12:** The terminology will be changed in the revised version of the text.

SC3 Page 3, l 31: 'Missing monthly precipitation figures …' I don't understand what is meant here by 'missing' figures in Dobrovolny et al (2015).

> **Response R3-13:** Corrected to "Missing monthly precipitation totals in …"

SC4 Section 2: Figure 1 I haven't found a reference to Figure 1 in the ms. Check on this please. Regarding this figure and the presentation of drought, in Section 2 there is some description of differences in definitions among the different drought indices used in the text. I think some comment on the available reconstructions are pertinent. There is a paragraph in page 2 (l 21-30) describing the origin of the series. Can the authors provide any thoughts on whether the different definitions really play a role or basically the same information is available, also considering the source data for the reconstructions. Can we anticipate any added value of using these three indices instead of one in this work?

> **Response R3-14:** In order to express various sides of droughts, there exists a great number of different drought indices. SPI, SPEI and PDSI represent those which are used in description and quantification of droughts most frequently, and they are also used for estimating impacts of agricultural and hydrological droughts. While there are obvious similarities between the respective time series (due to precipitation sums being the key factor shaping all of them), each of the indices represents slightly different approach. As mentioned by the referee, these are briefly summarized in Sect. 2.1, along with references to more comprehensive sources. Based on the differences found during our regression analysis, their individuality seems strong enough to justify inclusion of all three indices.
>
> Reference to Fig. 1 will be added to the text, to the Drought indices section.

SC5 Section 2.2: forcings I think it is desirable to place this forcing in the context of PMIP3 and PMIP4 forcings. The authors will find longer reconstructions of this forcing spanning the millennium that have been used to detect solar forcing on temperatures for instance since the 14th century (Schurer et al 2013). Maybe these can be better options for predictors than the one used in the ms (1610-present) Schurer, A., G. Hegerl, M. E. Mann, S. F. B.+Tett, and S. J. Phipps, 2013: Separating forced from chaotic climate variability over the past millennium. J. Clim., doi:10.1175/JCLI-D-12-00826.1.

> **Response R3-15:** We are grateful for the suggestions; in the revised version of the analysis, a recently published TSI reconstruction by Lean (2018; DOI 10.1002/2017EA000357) will be used to represent solar activity (providing full coverage for the 1501-2006 period). The previously employed TSI data by Coddington et al. (2016) will be retained as an alternative solar-related predictor.

SC6 Section 5 L 30 '… the increase in the ambient $CO_2$ post 1850 is clearly correlated with the increased probability of drought… while during the pre-instrumental period such link does not

manifest. The trend in CO2 in the industrial period is just one degree of freedom. Please recall the comments GC1b

> **Response R3-16:** True, but note please that while we mentioned the existence of a correlation, we did not interpret it as a causal relation, only noted the possibility of one (please see also our Response R3-4).

Technical corrections, typing errors,etc:
TT1 Page 6, l 8: '...results...standardized regression coefficients...' In an simple regression these would be, by definition, correlation coefficients. How does this differ in this analysis from correlations? Some methodological details on the multiple regression approach taken is advisable. Does it account for covariability in the predictors? Etc...Please provide more explanation of the relevant aspects in the ms.

> **Response R3-17:** Indeed, in simple linear regression, standardized regression coefficient corresponds to correlation between predictor and predictand (and, in absolute value, to square root of the coefficient of determination). In multiple regression, however, no such straightforward relation exists for individual predictors. Standardization of the coefficients is used to make them more comparable mutually (among different predictors as well as among predictands), thought, admittedly, this representation does not directly convey information about the magnitude of the responses. In the revised version, the responses will therefore be also shown in the form of predictor-specific time series generated for selected regression configurations (the respective graphs will be included in the Supplement of the paper).
>
> As for covariance of the predictors, its effect is reflected in the size of the confidence intervals for individual predictors (please see also response R3-5). This aspect of multiple linear regression will be emphasized in the revised version of the manuscript, within the expanded discussion of the effects of (multi)collinearity of the explanatory variables.

TT2 Page 2, l 20: '...that increase...' substitute by '...that the increase...' This is just an example. I have found a few of those. I think the text is easy to understand in general. However, I would recommend it would be revised for editing/English

> **Response R3-18:** Selected example corrected. The English of the manuscript was checked and corrected by a native speaker, Mr. Tony Long. The language correction will be repeated in the revised text once again.

---

## Author Response (AR1)

We would like to thank all three anonymous referees for their valuable comments regarding our manuscript. We tried to incorporate the corresponding modifications into our analysis and its proposed presentation in the revised paper, or bring arguments in cases when we were unsure how the proposals could be implemented into the manuscript without too severe changes of its context or aim.

Main changes to the analysis and its presentation in the revised manuscript:

- Results involving effect of the North Atlantic Oscillation (NAO) have been added in a quantitative form (whereas in the original manuscript, only a brief mention of NAO effects was made in text).
- More reconstructions have been employed to represent the AMO index (2 versions in total) and PDO index (3 versions in total) and comparison with the previous results based just on the reconstruction by Mann et al. (2009) has been provided. Predictor representing radiative forcing due to changes in the atmospheric composition has been altered to involve the aggregate effect of multiple greenhouse gases rather than just carbon dioxide. The solar variability predictor has been replaced by the recently published Total Solar Irradiance (TSI) data by Lean (2018), covering the entire 1501-2006 period, whereas the data by Coddington et al. (2016) are now used as an alternative solar activity descriptor.
- Discussion of the results has been expanded to pay more attention to the potential interactions between individual predictors.
- Additional results have been added to better illustrate properties of the regression mappings, including values of the coefficient of determination, samples of regression-estimated components associated with individual predictors, and graphs of regression residuals and their autocorrelation functions.
- Electronic Supplement has been added to the manuscript to hold extra materials illustrating outcomes of the supporting analyses.

Please see below for specific responses to the comments of referees 1, 2 and 3 (R1, R2, R3) and description of the corresponding changes to the manuscript.

The marked-up version of the manuscript, detailing the changes made, is attached at the end of this document.

**Anonymous Referee #1**

The paper analyses the long-term variability of droughts in the Czech lands based on long reconstructions (based on instrumental and documentary data). Time series of drought indices, temperature and precipitation are compared to reconstructions or time series of suspected drivers such as external forcings and oceanic variability modes. Anthropogenic radiative forcing as well as AMO/PDO are identified as influencing factors. The paper is interesting, valuable to the community and within the scope of Climate of the Past. However, I have several comments, which I think the authors should consider, before the paper can be published.

Methods

It is not fully clear which data are monthly or seasonal. Often the text mentions "seasonal and annual" or "monthly, seasonal and annual", which I found confusing. Also, the drought indices are usually calculated for individual measurement locations or grid cells. Here they are calculated for a large-scale average, as I understand. This should be made clear and explained. In the results section it then becomes clear that the seasons are analysed separately. However, what is the motivation for analysing a autumn or winter drought index?

> **Response R1-1:** The data description (Sect. 2.1, 2nd paragraph) should now make it more clear that the analysis was carried out on either series of true annual values (i.e., consecutive values representing means for an entire year) or series of season-specific means (i.e., one seasonal value for each year). The note regarding monthly series in Sect. 2 pertains to some of the original data sources; monthly values were not directly studied in the current analysis. The nature of the drought indices as area-wide means is now more explicitly stated in the text (Sect. 2.1, 1st paragraph). Since drought data for all seasons (including autumn and winter) were available and analyzed, we present the outcomes for all four seasons, to illustrate the full range of potential climate links. It should also be considered that even though for some applications (such as investigation of agricultural droughts) spring and summer conditions may be of greater interest, recharge of the underground water resources and surface reservoirs depends on the water available during autumn and winter and droughts in these periods often induce major hydrological impacts in the following year.

Multiple linear regression is used to separate individual components, but fully separating external forcing from internal variability (e.g., oceanic modes) is fundamentally difficult. External forcings might operate via altering internal variability modes (e.g. solar and volcanic forcing might change the climate system via AMO or ENSO). Conversely, AMO and PDO have the imprint of global temperature rise. I see that the authors use cross-wavelet spectra, partly to assess the interdependencies, but not systematically. Partial correlation methods could be used to go into more depth here, or different models could be compared. In any case, the interpretations should be phrased very carefully.

> **Response R1-2:** Indeed, the problem of separating the strictly external forcings from the internally induced variability is a complicated one, not only at a statistical level, but also with

regard to the underlying physical mechanisms. While this was not mentioned in the manuscript, we examined the mutual links between individual predictors with episodic or oscillatory components in terms of Pearson (cross-)correlation. Although some potentially noteworthy correlations appeared, none of them (other than the AMO-PDO relation) seemed strong and stable enough to warrant a specific treatment of inter-predictor links, at least not in the context of purely linear regression. Therefore, in our analysis eventual external forcing-induced components in the indices of internal climate variability modes were treated as a part of these indices; components attributed by the regression analysis to the forcings themselves were then treated as direct responses. To provide a more complete picture of the potential indirect effects of external forcings manifesting through their influence on the internal climate variability modes, results of analysis carried out with just the predictors representing external forcings (solar, volcanic, anthropogenic) have been added to the revised manuscript (Fig. S1a in the Supplement), and they are mentioned in the main text (p. 16, l. 18+). The effect of solar and volcanic forcing on NAO is also discussed (paragraph starting at p. 16, l. 3) and documented in the Supplement (Fig. S7). Furthermore, the Discussion has been expanded to provide additional references to some works addressing the influence of external forcings on NAO (Ortega et al., 2015; Sjolte et al., 2018). We did, however, not attempt to extend our analysis to involve the effects of external forcings on long-term components in the internal climate variability modes (particularly AMO and PDO), as this issue would require a considerably different methodological approach.

In the case of the imprints of global temperature in the AMO and PDO indices derived from data by Mann et al. (2009), please note that the long-term temperature component (in the form of mean northern hemispheric temperature) has been removed from the data during pre-processing (as described in Sect. 2.2), and the AMO/PDO predictors therefore only encompass oscillatory variations around the hemispheric temperature series. This is stated in the respective paragraphs of Sect. 2.2.

Due to the sheer amount of possible combinations, results of the cross-wavelet analysis were only presented for selected pairs of predictors/predictands, either those showing interesting interactions, or those intended to illustrate a similarity or contrast in behavior compared to some other pair of variables. In the revised version, the interactions between SPEI, temperature and precipitation and the primary predictors have been retained in the manuscript (Fig. 7); results for the alternative predictors are given in the Supplement (Fig. S2). Selected additional cross-wavelet spectra have also been added to the Supplement, illustrating interactions between predictands (Fig. S3a) and between different predictors and their versions (Figs. S3b, S3c).

While we agree that partial correlations can offer additional insight into the interdependencies in a multivariable system, their use does not necessarily solve the ambiguity arising from the existence of a common, physically relevant component within multiple explanatory variables, stemming not from a one-way causality, but rather from a two-or-more-way interaction. Such a component cannot be reliably assigned by purely statistical means and since its origin is typically rather complex, we prefer to deal with its presence and interpretation during the discussion of the results. Note also that in the most prominent case of such collinearity in our analysis, related to the similarity between AMO/PDO predictors by Mann et al.

(2009), we addressed this problem by carrying out regression with mean value and difference of the AMO and PDO series; the outcomes are now shown explicitly (Fig. S1c) rather than just mentioned in the main text. Results for regression involving AMO-only and PDO-only configuration of predictors are now also included (Fig. S1b) instead of just discussed.

The regression model itself is not explained clearly. From the text it becomes clear that different ENSO indices were used, but which model (which ENSO index) is the one shown in Figs. 4 and 5? Furthermore, only very late in the paper we learn that the explained variances are very low, below 5%. Should we even analyse regression models that have no explanatory power? Finally, the effect of reconstruction uncertainty is not discussed.

> **Response R1-3:** The missing identification of the primary ENSO index has been corrected – it is now explicitly stated that the results in Figs. 4 and 5 are based on the ENSO reconstruction by Li et al. (2011). Furthermore, due to the inclusion of alternative predictors in the revised version of the manuscript, individual data sources are now systematically specified in the individual figures whenever more than one version of the predictor exists.
>
> The seemingly low fraction of variance explained by the regression models ($R^2$) is a result of dominance of inter-annual variability in the predictand series, matched in the regression mapping against predictors mostly dominated by inter-decadal variation. Formally, higher $R^2$ could be achieved by removing the year-to-year variations, e.g. by smoothing the series by a moving average filter (to give an example, for the period 1501-2006, 21% of variability of the annual SPEI series can be explained by the regression model if the series are smoothed by 11-year moving average; this value increases to 33% when the NAO reconstruction by Luterbacher et al. (2002) is also included as a predictor). However, since some of our explanatory factors (the episodic volcanic activity, 11-year cycle in the solar variability signal, and the NAO index in the revised version of the analysis) do exhibit faster variability, which would be largely erased by the smoothing, we prefer to perform the analysis with the unaltered series. The prominence of individual explanatory factors is evaluated through statistical significance of the respective regression coefficients, regardless of the overall $R^2$ – an approach that we believe to be consistent with our primary aim, i.e. identification of forcings and large-scale factors influential in establishing the drought regime of the Czech Lands (as opposed to an attempt to construct a predictive model reproducing the series with as much variability as possible). To better illustrate the actual magnitude of components associated with individual explanatory factors, a sample of time series of regression-generated components corresponding to individual explanatory variables has been included in the Supplement (Fig. S5); furthermore, values of $R^2$ have been added to Figs. 4 and 5 for each of the regression configurations.
>
> The effect of uncertainties tied to the results would be rather difficult to quantify reliably, as not all series analyzed come with an uncertainty estimate, and methods of its estimation differ even when such data exist. However, due to increased number of versions of some predictors in the revised version of the analysis, more attention is paid in the revised manuscript to the robustness of the results based on different reconstruction sources.

The paper says little about the mechanisms linking the external and internal drivers to drought and hydroclimatic conditions in general. Obviously a study using reconstructions cannot explicitly address net radiation, soil moisture, temperature effects, land-surface feedbacks, atmospheric circulation effects (blocking), etc. But it would be nice to read the authors' hypotheses. The paper is rather silent about mechanisms. In the introduction Hess-Brezowsky weather types are mentioned, and later the NAO, but the NAO is not incorporated into the analyses and the discussion parts then follows another thread: Doing a PC analysis of AMO/PDO. It would be nice if the Discussion section could come back to mechanisms at some point.

> **Response R1-4:** Note please that our study is dealing with droughts defined through the SPI/SPEI/PDSI indices, shaped by (and calculated from) precipitation and temperature series. Our interpretation of the possible links is therefore focused on the role and eventual interaction of the temperature and precipitation variability in establishing the central European drought regime expressed by the above indices. Also note that some of the responses, while statistically significant, represent rather minor tendencies, difficult to reliably assign to specific mechanisms (especially in our analysis involving pre-instrumental period, as no data exist consistently capturing global large-scale circulation over the last five centuries, making it difficult to evaluate influences related to circulation, blocking, etc.).
>
> NAO-related effects have been included in the revised version of the manuscript, based on the NAO index reconstructions by Luterbacher et al. (2002) and Ortega et al. (2015) as well as multidecadal NAO variability reconstruction by Trouet et al. (2009). The results in Figs. 4 and 5 have been updated to show regression coefficients related to NAO in addition to the previously considered predictors; the Discussion has been expanded to include analysis of the NAO-related links.

Minor comments

Abstract, l. 14: "external and internal climate forcings". Please be careful with terminology here and elsewhere. Considering the coupled climate system "forcing" is used for external influences (subdivided into natural and anthropogenic) while internal variability is used for the dynamics of the coupled system even if unforced. When considering only the atmosphere, "oceanic forcing" is sometimes used. In any case, the terms should be defined and used consistently.

> **Response R1-5:** Terminology in the manuscript has been modified to avoid use of the term 'forcing' for factors originating from internal climate dynamics.

P. 2, L. 5: a substantial number of studies: cite
P. 2, L. 5: A lot of work has been done on droughts in the USA. Perhaps before zooming in on Europe, you could mention that.

> **Response R1-6:** Both comments accepted, corrected by adding new references as follows (p. 2, l. 15 in the revised manuscript): "In addition to a substantial number of studies investigating drought indices for the instrumental period in Europe (e.g. van der Schrier et al., 2007, 2013;

Briffa et al., 2009; Sousa et al., 2011; Todd et al., 2013; Spinoni et al., 2015; Haslinger and Blöschl, 2017) and other areas of the world (e.g. Dai, 2011; Spinoni et al., 2014; Ryne and Forest, 2016; Wilhite and Pulwarty, 2018), generally calculated …".

P. 2, L. 24: instrumental precipitation series

**Response R1-7:** Corrected as "The authors demonstrated the importance …"

P. 3, L. 10: What time window was used for the SPI?

**Response R1-8:** The time window used for the SPI/SPEI calculation was chosen to match the type of the series used as predictand, i.e. 12 months for the annual data, 3 months for seasonal data.

P. 4, L. 3 and 4: I do not understand this sentence.

**Response R1-9:** The sentence has been changed to (p. 4. l. 26): 'As a result, a 100-member ensemble of distributions of monthly precipitation totals for each season and the year was obtained. These distributions were then applied for calculation of indices for every year in the 1501–1803 period.'. Note, please, that this is just a substantially simplified description of the data preparation process, and full explanation can be found in Brázdil et al. (2016a).

P. 4, L. 10: "climate forming agents": rephrase

**Response R1-10:** Reformulated to 'climate-defining factors'

P. 4, L. 19 and 20: Omit the first part of the sentence, which is unnecessary. Start with "A large part…"

**Response R1-11:** Accepted

P. 4, L. 26: strong clear?

**Response R1-12:** Changed to 'clear'

P. 5, L. 1-3: Perhaps cite Fischer et al. GRL (https://doi.org/10.1029/2006GL027992)

**Response R1-13:** The reference to Fisher et al. (2007; DOI 10.1029/2006GL027992) has been added to the revised manuscript.

P. 5, L. 19: I am a bit puzzled why the authors use the Mann et al. ENSO series. As the authors write (and other authors have also pointed to that), the reconstruction varies mostly on the 8-20 year time scale. Why use it as an ENSO time series then? I would rather use other ENSO reconstructions. Similarly, for AMO and PDO it would be nice to have two indices for each (e.g. Shen et al. 2006 for the PDO, Gray et al. 2004 for the AMO).

**Response R1-14:** Please note that ENSO reconstruction by Li et al. (2011) was used as the primary descriptor of ENSO, and a basis for the results shown in Figs. 4 and 5. ENSO index derived from the Mann et al. (2009) data was only used as an alternative ENSO descriptor. This should now be more clear from the revised text, as more thorough identification of individual data sources is given throughout the text. Regression outcomes for Mann et al. (2009) ENSO data are now included in the Supplement (Fig. S1i).

Results based on the PDO reconstructions by MacDonald and Case (2005) and Shen et al. (2006) and AMO reconstruction by Gray et al. (2004) have been included in the revised manuscript and discussed along with the outcomes of the analysis utilizing the originally employed PDO/AMO data by Mann et al. (2009). Regression coefficients are presented for each version of the predictors (some of them in the Supplement, Fig. S1); their similarity (or lack thereof, as is the case for the PDO reconstructions) is now discussed with regard to the robustness of the results and the associated uncertainties (in the relevant sections of the Discussion).

P. 10, L. 34: Is the tendency for wet conditions after volcanic eruption really due to lower temperatures?

**Response R1-15:** This formulation is meant to reflect the fact that the tendency towards higher values of the drought indices during periods with higher amounts of volcanic aerosol coincides with significant drop of temperature, while no statistically significant change in precipitation is indicated.

P. 11, L. 34: I am surprised that Sutton and Hodson (2005) paper is not mentioned in context with the AMO effect.

**Response R1-16:** The reference to Sutton and Hodson (2005) has been added to the revised manuscript (p. 13, l. 31).

**Anonymous Referee #2**

GENERAL COMMMENTS:

Manuscript under revision is an approach to study of drought in Czech Republic area, taking in consideration previous climatic reconstructions, already published, using these informations to generate drought indices based on instrumental records. Analysis of possible relations with different forcing factors is also made to offer general or initial explanation to drougth mechanisms for this Central Europe study area.

Main effort focused to compare rainfall indices and temperatures for long or complete periods. It's a good first approach to drougth phenomena. It open research to study specific events at higher temporal resolution, impacts and responses, etc.

SPECIFIC COMMENTS

+ Title could include temporal dimension of work of manuscript.

> **Response R2-1:** Because we are using the expression "long-term", it is probably not necessary to extend the title for the time span used.

+ Title. Expression "drought" into title is excessively general. A more correct definition of topic developped into manuscript would be "drought indices".

> **Response R2-2:** Accepted and also with respect to a comment of Referee 3 changed to "Long-term variability of drought indices in the Czech Lands and effects of external forcings and large-scale climate variability modes"

+ Lines 28-30. Seasonal and annual precipitation for 1501-1854 is reconstructed from "document-based precipitation indices". Dobrovolvy et al., 2015. Could explain in a short description general characteristics or contents of these "documents"? How was developed previous analysis. Just to have a connection between original information and present results generated into manuscript. IF drought is analyzed, at least public must know about historical documents used for reconstruction, temporal resolution of information obtained, locations or regions with available information, aspects of natural process and/or and human impacts detected/evaluated....etc.... I understand manuscript can have restrictions of extension, but this short overview would be useful for public.

> **Response R2-3:** This comment and several following remarks of Referee 2 concern details related to the documentary data used. We would like to stress that the primary aim of the analysis is the study of forcings and large-scale climate drivers reflected in series of drought indices, described in detail in the paper by Brázdil et al. (2016a). Because their calculations are based on reconstructed temperature (Dobrovolný et al., 2010) and precipitation (Dobrovolný et al., 2015) series, in which a detailed information of documentary data used with their types, examples and critical evaluation are given (as well as the reconstruction uncertainties), it would

bring not too much new information to the merit of this article. But looking at the comments of the Referee 2, we included several additional sentences in this direction with hope to fulfill at least partly these requests by the change of the fifth paragraph of Section 2.1 as follows (p. 4, l. 3):

"Long-term seasonal and annual series of these three indices, dating from AD 1501 in the Czech Lands (Brázdil et al., 2016a) were used in the current study. They were derived from 500-year temperature and precipitation reconstructions based on a combination of documentary data and instrumental measurements. Documentary data comprised descriptions of weather and related phenomena from a variety of documentary evidence, some of it individual, some of it of an institutional character, such as annals, chronicles and memoirs, weather diaries (non-instrumental observations), financial and economic accounts, religious records, newspaper and journals, epigraphic sources, and more. Such data in the Czech Lands cover particularly, at varying degrees of density, the period from AD 1501 to the mid-19th century, but continue even to the present time. The spatial density of such data changes over time, depending on the availability and extraction of existing documentary sources. All the data collected were critically evaluated with respect to possible errors in dating or spatial attribution and were used for interpretation of monthly-weighted temperature and precipitation indices on a 7-degree scale, from which series of seasonal and annual indices were created (for more details of the use of documentary data, its critics, analysis and interpretation, as well as creation of series of indices in historical climatology, see Brázdil et al., 2005, 2010). Such data were further used as a basic tool for temperature/precipitation reconstructions. Firstly, Dobrovolný et al. (2010) reconstructed monthly, seasonal and annual central European temperature series, partly based on temperature indices derived from documentary data for Germany, Switzerland and the Czech Lands in the 1501–1854 period and partly on homogenized instrumental temperature series from 11 meteorological stations in central Europe (Germany, Austria, Switzerland, Bohemia) from 1760 onwards. This temperature series is fully representative of the Czech Lands. Subsequently, seasonal and annual precipitation series for the Czech Lands were reconstructed from documentary-based precipitation indices in the 1501–1854 period and from mean precipitation series calculated from measured precipitation totals in the Czech Lands after 1804 (Dobrovolný et al., 2015)."

+ Bibliography used on work is complete and well updated.

> **Response R2-4:** Thank you.

+ Effort to offer a background or general overview about drought events is not so complete as we would like find. For example, justification of study of drought. It's a present or potential problem for Czech Republic?, any previous strong event justify this study? How they are drought conditions in Czech Republic?

> **Response R2-5:** To fulfill this comment, the first paragraph of Introduction has been changed as follows: "Droughts, among the most prominent manifestations of extreme weather and climate anomalies, are not only of great climatological interest but also constitute an essential factor to

be considered in the assessment of the impacts of climate change (Stocker et al., 2013; Trnka et al., 2018; Wilhite and Pulwarty, 2018). This is also valid for the territory of the Czech Republic where droughts, apart from floods, constitute the most important natural disasters, with significant impacts upon various sectors of the national economy, such as agriculture, forestry, water management, and tourism/recreation. Since the Czech Republic lies on a continental divide with rivers flowing out of its territory, it depends on atmospheric precipitation alone for its water supply. Although certain extreme droughts with important socio-economic and political impacts are known from the past, such as the drought of 1947 (Brázdil et al., 2016b), studies performed in recent years show the Czech climate has become increasingly dry in the past 2–3 decades, expressed in terms of higher frequency of extreme droughts with significant consequences (e.g. Brázdil et al., 2015b; Zahradníček et al., 2015). The abundance of long-term instrumental meteorological observations has provided a basis for a number of recent drought-focused studies, revealing complex regional drought patterns and a richness of features observed at various spatial and temporal scales, in the European area (e.g., van der Schrier, 2006, 2007; Brázdil et al., 2009; Briffa et al., 2009; Dubrovský et al., 2009; Sousa et al., 2011; Spinoni et al., 2015) as well as other areas of the world (e.g. Dai, 2011; Spinoni et al., 2014; Ryne and Forest, 2016; Wilhite and Pulwarty, 2018). Along with more rapid variations, these also include long-term variability, such as a distinct trend towards drier conditions, prominent especially during the late 20th and early 21st centuries (e.g., Trnka et al., 2009a; Brázdil et al., 2015b). "

+ Historial dimension of drought is not analyzed. Just index values from previous research considered as approach to climatic patterns related to low values of reconstructed precipitation. Drought is not studied by itself as climatic/historic phenomena. This aspect is not negative nor positive. Just it would require any extension of explanations about drougth as climatic pehnomena in introduction of work.

> **Response R2-6:** As explained above, our manuscript concentrates on the explanation of effects of external forcings and large-scale climate drivers on long-term drought indices variability in the Czech Lands. This means that we are really not analyzing "historial dimension of drought" as the referee correctly states because it does not fit to the concept of this paper.

+ No specific drought events are mentioned. No description at least for one event is included into manuscript. Complexity of drought events and related impacts is not described/evaluated. May be authors are preparing other papers with these specific topics?

> **Response R2-7:** The description of any "specific drought event" does not fit to the paper context, analyzing rather effects of external forcings and large-scale climate drivers in long-term drought indices series. Descriptions of specific drought events in the Czech Lands can be found, for example, in Brázdil et al. (2013) or Brázdil et al. (2015b). Moreover, the paper "Extreme droughts and human responses to them: the Czech Lands in the pre-instrumental period" by Brázdil et al. was currently submitted to Climate of the Past (https://doi.org/10.5194/cp-2018-135).

+ No explanation about drought as climatic phenomena. How is considered drought in Czech Republic, what criteria are applied, what instrumental thresholds, duration/extension/severity, different concepts/definitions of drougth, affectation of agriculture.... Any explanation would be useful to understand characteristics and effects for public unknowning these specific details.

> **Response R2-8:** We would like to stress that we are not concentrating in this paper primarily on "drought as climatic phenomena" or "what criteria are applied, what instrumental thresholds, duration/extension/severity, different concepts/definitions of drought, affectation of agriculture", because it was analysed already in many other papers related to the territory of the Czech Republic (for comprehensive overview see e.g. Brázdil et al., 2015b). We are just trying to find how fluctuations in series of drought indices in the Czech Lands are influenced by external forcings or large-scale climate drivers.

+ If drougth is defined only from specific indices (SPI, SPEI...), when we work in historical time, out of instrumental data availability, this topic must be taken with more introductory explanations. A more complete and informative approach about how documentary records detect and define droughts, what they record, what transmit….

> **Response R2-9:** We would like to stress that drought indices are not primarily derived (calculated) from documentary data, but from temperature/precipitation reconstructions based on documentary-based indices series and overlapping instrumental series. For this reason we are of the opinion that comment "A more complete and informative approach about how documentary records detect and define droughts, what they record, what transmit...." could be difficult to follow in the recent concept of our paper.

+ If manuscript is based on previous reconstructions, focused on reconstructed values of mm. rainfall, by total monthly/seasonal/annual resolution, authors must consider they cannot analyze all dimension of droughts. Rainfall indices with positive aspect can cover important drought events, when dry periods are interrupted by strong rainfall events. Knowing what tipe of drought is under study, these singular aspects could be differenced, generating a better and deeper study.

> **Response R2-10:** We agree with the opinion of referee 2 but we are not analyzing drought on the base of precipitation indices. Precipitation reconstruction was used only as one of two basic series which were used to calculate series of drought indices.

+ Manuscript doesn't show a clear relation of type of documents and type of information rescued and anallyzed.

> **Response R2-11:** As mentioned in Section 2.1, we analyse effects of external forcings and large-scale climate drivers in long-term variability of drought indices series, calculated from reconstructed series of temperatures (Dobrovolný et al., 2010) and precipitation (Dobrovolný et al., 2015). Both these papers contain detailed information about types of documents and information rescued and analysed. Calculation of drought indices was explained in detail in the

paper by Brázdil et al. (2016a). From these reasons we do not see as necessary to repeat in detail all these aspects in the present paper.

+ It would be interesting focuse efforts on variability and extreme events of the same variable before to compare with variability of others proxys.

**Response R2-12:** Aspects reported by the referee (variability, extreme events, …) were dealt in a great detail already in the paper by Brázdil et al. (2016a). We feel it redundant to repeat it here again because it does not fit to the context of the present article.

**Anonymous Referee #3**

Description

This paper addresses the understanding of the variability of droughts, temperature and precipitation in the Czech lands 1500-present from the point of view of its dependence on internal drivers (e.g. some specific modes of circulation) and also external forcing factors (volcanic, CO2, etc). For that purpose a multiple linear regression is applied having as predictand variables 3 drought indices and as predictors a set of external and internal drivers.

The purpose of the paper has value and meaningful and solid results in this direction would be worth to be published in CP. If attribution of drought variability or a meaningful step forward in its understanding in the Czech Lands would be attained I think this would be sufficiently valuable in my view to support publication. Therefore, I encourage authors to pursue this line of work towards publication. At this state, I would recommend major revision of the manuscript. There are several issues related to the rationale, methodology and description and interpretation of results that in my understanding require revision. I will argue about this in the following points.

General Comments

GC1 General approach to attribution As it is described in the paper, 5 predictand series (3 drought indices and a temperature and a precipitation series) are examined using multiple linear regression as functions of independent predictors, the latter being internal and external in nature. In practice, these are 5 individual multiple regressions.

Having that in mind I would suggest to consider the analysis, description and discussion of the: a) selected predictors; b) of the methodological approach; and c) of the residuals of the methodology.

a) Selected predictors. I would argue these are insufficient in both the case of the external and internal subsets.

a.1-Regarding external predictors I have no objections to the ones considered so far but the authors should discuss why important predictors like other greenhouse gases (GHGs), aerosols and particularly land use land cover (LULC) are not considered. For the case of other GHGs than CO2, it would be more elegant either to consider them or to use equivalent CO2. For the case of aerosols some arguments or strategy or implementation should be considered also. For the case of LULC, this would really be an important variable since it can have an impact on drought. If any significant trends are found, how can we attribute them arbitrarily to CO2 or to a mix of the influence of GHGs and aerosols? If there has been progressive changes in LULC in the area, in the context of this manuscript, eluding them would be really misleading for the results of this analysis.

> **Response R3-1:** It is true that using just $CO_2$ concentration as an approximation of anthropogenic influence oversimplifies the setup. In the revised version of the analysis, aggregate radiative forcing of multiple GHGs (including $CO_2$, $CH_4$ and $N_2O$) is therefore used instead. As for the inclusion of the effects of (tropospheric) aerosols, their regional effect is difficult to consider in

an analysis such as ours, due to high temporal and spatial variability of their concentrations and differences in behavior of different aerosol species. Note also that from a standpoint of a regression analysis, the predictors with and without the aerosol forcing are usually quite similar, as the respective time series are very strongly correlated. For instance, using the Meinshausen et al. (2011) global annual concentration and forcing data over the 1765-2005 period, the $CO_2$ concentration series is correlated with total anthropogenic forcing (representing the aggregated effect of various greenhouse gases as well as aerosols) at 0.995. There would therefore be almost no change of the regression results if different versions of the predictor representing anthropogenic forcing were applied (despite the obvious differences in the physical effects involved). This is now explicitly mentioned in the manuscript (p. 8, l. 14+)

Regarding the Land use land cover (LULC): We are working with drought indices for the whole Czech Lands calculated from reconstructed temperature and precipitation series. The calculation procedure of none of these indices includes information about LULC. Although it can be an important factor deciding about drought severity and particularly its impacts, effects of LULC on country-wide temperature and precipitation should be limited. In this study oriented on long-term temporal changes it seems to be not an important factor helping us as predictor in the regression analysis of drought indices series.

a.2- Regarding internal predictors, the NAO is argued to be important but has not been used. Even if it has been described in previous works, it is relevant to see in this approach how much variability do ENSO or PDO account for from the residuals once the NAO has been taken into account. Do the results of the analysis concerning the presently used internal predictors change if the NAO index is used? There are some millennium long index reconstructions that would allow for this exercise. I think there is no point in looking only at Pacific indices without considering a potential larger explanatory variable like this one.

**Response R3-2:** Our original intention was to concentrate on mid-to-long-range variability in the drought series, i.e. oscillations typically slower than the dominant variability of NAO. Moreover, the strong relation between central European drought regime and NAO phase has been established by various prior studies, hence we considered it to be less interesting for the current analysis. Since both Referees 1 and 3 expressed their interest in the NAO-related effects, in the revised version of the paper, results involving NAO reconstructions by Luterbacher et al. (2002), Ortega et al. (2015) and Trouet (2009) have been included. The results in Figs. 4 and 5 have been updated to show regression coefficients related to NAO in addition to the previously considered predictors; the Results, Discussion and Conclusions sections have been expanded to include analysis of the NAO-related links.

b) Methodological issues There are three ideas that I would like to bring here. One is the linear vs non-linear character of the influences that the paper tries to assess. Another one is the power of the approach used herein related to the covariance structure pursued by the analysis in view of the properties of the predictors. Finally, and related, the collinearity of some predictors.

b.1 Regarding the first one, this is commented in the first paragraph of Sec. 3. I have no reservations against the possibility of nonlinear interactions being relevant. I think it is though important and has value, to study the linear relationships. It is also important to study it in a solid way so that we minimize the danger to loosely argue that everything we cannot explain with a linear approach is due to the limited character of its 'linearity' and probably due to nonlinear interactions.

> **Response R3-3:** We seem to be in agreement with the referee; the mention of nonlinear approach was meant to provide a methodological context while also giving rationale for using linear version of regression for analysis of links in an inherently nonlinear system.

b.2 Regarding the second one, the multiple linear regression is a valid approach to analyze the linear covariance structure in the data. Now, for that purpose, the variables used as predictors like CO2 or, for that case, if additional GHGs+aerosols would (and should) be considered, since these variables present very low variability at high and mid frequencies, one has to be careful in how to handle them in terms of covariance. For instance, a positive coefficient with temperature in the instrumental period means that both temperature and CO2 show positive trends... but any variable showing a positive or a negative trend would show association for that matter. The limited meaning of correlating preindustrial CO2 (+GHGs+aerosols) must be commented and the limited interpretation of correlating trends in the industrial period also should be argued and improved by including other GHGs and aerosols in a meaningful way.

> **Response R3-4:** This is definitely true, and admittedly under-explained in the original manuscript. The inclusion of a trend-like variable ($CO_2$ concentration in the original version of the manuscript, composite GHG forcing in the revised one) was meant to provide a predictor potentially approximating long-term evolution observed in the drought indices. Naturally, despite the similarity in shape (and thus statistical significance of the link detected for some of the drought indices), the formal relationship does not prove causal relation. While we commented on this in the original version of the text ('Even considering that statistical attribution analysis can only reveal formal similarities and cannot verify the causality of the links detected ...' in the Conclusions), and referenced supporting evidence pointing to a physical link between droughts and anthropogenic forcing (the second paragraph of Discussion), the potential for mis-attribution has now been more explicitly emphasized in the revised manuscript (2nd paragraph of the Discussion, p. 12, l. 2+).

b.3 Some of the predictors (eg. AMO, PDO) show covariability. How is this addressed in the analysis and how does this influence the results? Explaining which type of multiple regression approach would be important for this point.

> **Response R3-5:** For the AMO and PDO representations based on the Mann et al. (2009) temperature reconstruction, this was actually addressed (in the Discussion section) by employing a simple form of principal component analysis, allowing to better assess the role of the common component in these predictors and of their difference. In the revised manuscript, the respective

results are now shown in more detail, including the graphs of the regression coefficients pertaining to predictor configurations involving only AMO or only PDO (Figs. S1b, S1c).

Please note also that (multi)collinearity of the predictors results in increased variance inflation factor for the regression coefficients (and thus wider confidence intervals). Since this is an inherent feature of multivariable regression, we did not comment on it specifically; the effect can, however, be seen from Figs. 4 and 5, and it is mentioned in the context of the AMO/PDO collinearity (p. 14, l. 5+).

c) Residuals This is also a rather methodological issue. If the purpose is to statistically describe drought with a multiple linear regression approach, the behavior of the residuals should be discussed. The authors should show estimation of drought variability from the predictor variables, explained variances and some convincing arguments that part of the variability is being reproduced by the predictors used.
I recognize this point, GC1, is rather long. It should probably be treated as independent points. Nevertheless I think it is important and would like to see the arguments for all these. Some specific comments will also follow below.

**Response R3-6:** The analysis of regression residuals was performed when designing the optimum setup for the moving-block bootstrap. The only noteworthy feature (aside from the approximately AR(1)-consistent persistence structure) was a presence of a weak and rather unstable 22-year-period oscillation (possibly an imprint of the 22-year cycle in solar activity, but inconsistently present throughout our analysis period). This is now mentioned and discussed in the revised version (Discussion, p. 15, l. 32+). Graphs illustrating residual variability have been included in the Supplement (Fig. S4), along with charts of the residual autocorrelations.

As for the explained variances and evaluation of the regression results, $R^2$ values have been added to the results in Figs. 4 and 5 in the revised manuscript, and sample graphs illustrating regression-estimated components pertaining to individual predictors have been included in the Supplement, as Fig. S5. Regarding the reproduction of variability by our predictors: Please note that prominence of individual explanatory factors is evaluated through statistical significance of the respective regression coefficients, regardless of the overall $R^2$ – an approach that we believe to be consistent with our primary aim, i.e. identification of forcings and large-scale factors influential in establishing the drought regime of the Czech Lands (as opposed to an attempt to construct a predictive model reproducing the series with as much variability as possible).

GC2 Mechanisms As it stands, the approach of the manuscript is to argue on the basis of the regression coefficients. This is quite extreme in its present state. Even in the discussion part, a relatively aseptic account of the results of other authors are provided in this sense. However a more mechanistic based approximation discussing the rationale behind the statistical relationships that may be found is needed.

**Response R3-7:** Note, please, that most of the connections highlighted in our analysis represent rather weak (albeit sometimes statistically significant) tendencies, which are difficult to assign unambiguously to specific mechanisms (especially considering that no observational data exist that could be used for analysis of global circulation patterns over the full period of the last five

centuries, and that dynamical models are still rather unreliable in capturing some of the relevant factors, including the sources of multidecadal climate variability). This is further complicated by often mutually inconsistent or contradicting accounts regarding the effects and mechanisms of some of the relevant forcings/variability modes in the existing literature. Assessment of the possible factors behind our results would therefore be quite speculative on our part. Future, more topically focused (and methodically wider) analysis may bring better understanding of the relevant questions; such an effort would however go beyond the intended scope and aim of our present study.

GC3 Temperature and precipitation What does having temperature and precipitation add in this analysis? I don't mean to be unconstructive... just that if it is included in the analysis pursuing a more in depth understanding of drought, the reader should understand why they are there. What gain in our understanding do we get from including temperature and precip and analyzing them as predictors? Would it be of use including them in one exercise as predictands and assess their relative influence on drought?

> **Response R3-8:** Temperature and precipitation data were used for calculation of the drought indices themselves (as explained in Sect. 2.1), and their behavior is therefore crucial when discussing their combined effect in the drought descriptors.

GC4 Section 5: PCA analysis The strategy for the PCA analysis in page12 should be described (already in the methods section), as well as it is purpose and results presented in the text... unless the results are rendered invalid or not useful. If the analysis provides some valuable insights within this ms, it should be shown.

> **Response R3-9:** We did not mention PCA in the Methods section, as it was only employed as a supporting technique in a small part of our analysis (and we assumed its general principles to be well-enough known to not require introduction). In the revised version, the transformation of the AMO/PDO pair is introduced simply as a calculation of their mutual mean and difference; PCA is now only mentioned as an alternative way to produce the same result. The results based on analysis of principal components are now presented directly instead of just mentioned in the text (Fig. S1c in the Supplement).

GC5 Section 5: discussion The Discussion section provides a wealth of information on different results from various papers. However, in my opinion it misses a bit some purpose or direction. Actually, it also reports on results (e.g. GC4) that are not shown although they permeate to the conclusions. This is not recommendable. I suggest to pass any results clearly to the parts of the paper to make clear the objectives, methods and analysis of the results. Having a Discussion part or a Conclusion and Discussion makes sense to put the results of the present ms in view of past literature and state clearly what we learn from it. I would advise the authors to modify this section in this sense.

> **Response R3-10:** The results originally just mentioned (but not shown) in the manuscript are now included fully, either in the main paper or in its Supplement. The Introduction and

Discussion have been modified to paint a clearer picture of our main objectives: to assess the existence of links between Czech drought indices and climate forcings or activity of large-scale internal variability modes, and to investigate the properties of the existing reconstructions.

Specific comments

SC1 Title: '… large-scale climate drivers' If we understand 'large-scale climate drivers' referring to modes of circulation, shouldn't the title also include those? E.g. ' Longterm variability of droughts in the Czech Lands due to external forcing and large scale climate drivers' ?

> **Response R3-11:** Based on this comment and suggestions of Referee 2, we changed the title to "Long-term variability of drought indices in the Czech Lands and effects of external forcings and large-scale climate variability modes"

SC2 Page 2, l 17: 'Internal forcings' I think the use of this concept is not adequate in the manuscript. We relate forcing factors to changes in the energy of the system and, therefore, external in nature. I agree with using the terms internal/external drivers or external forcings, but not internal forcings.

> **Response R3-12:** The terminology has been changed in the revised version of the text to avoid use of the term 'forcing' for factors originating from internal climate dynamics.

SC3 Page 3, l 31: 'Missing monthly precipitation figures …' I don't understand what is meant here by 'missing' figures in Dobrovolny et al (2015).

> **Response R3-13:** Changed to "Missing monthly precipitation totals in …"

SC4 Section 2: Figure 1 I haven't found a reference to Figure 1 in the ms. Check on this please. Regarding this figure and the presentation of drought, in Section 2 there is some description of differences in definitions among the different drought indices used in the text. I think some comment on the available reconstructions are pertinent. There is a paragraph in page 2 (l 21-30) describing the origin of the series. Can the authors provide any thoughts on whether the different definitions really play a role or basically the same information is available, also considering the source data for the reconstructions. Can we anticipate any added value of using these three indices instead of one in this work?

> **Response R3-14:** In order to express various sides of droughts, there exists a great number of different drought indices. SPI, SPEI and PDSI represent those which are used in description and quantification of droughts most frequently, and they are also used for estimating impacts of agricultural and hydrological droughts. While there are obvious similarities between the respective time series (due to precipitation sums being the key factor shaping all of them), each of the indices represents slightly different approach. As mentioned by the referee, these are briefly summarized in Sect. 2.1, along with references to more comprehensive sources. Based on the differences found during our analysis, their individuality seems strong enough to justify inclusion of all three indices.

Reference to Fig. 1 has been added to the text, to the first paragraph of Sect. 2.1.

SC5 Section 2.2: forcings I think it is desirable to place this forcing in the context of PMIP3 and PMIP4 forcings. The authors will find longer reconstructions of this forcing spanning the millennium that have been used to detect solar forcing on temperatures for instance since the 14th century (Schurer et al 2013). Maybe these can be better options for predictors than the one used in the ms (1610-present) Schurer, A., G. Hegerl, M. E. Mann, S. F. B.+Tett, and S. J. Phipps, 2013: Separating forced from chaotic climate variability over the past millennium. J. Clim., doi:10.1175/JCLI-D-12-00826.1.

**Response R3-15:** We are grateful for the suggestions; in the revised version of the analysis, a recently published TSI reconstruction by Lean (2018) has been used to represent solar activity (providing full coverage for the 1501-2006 period). The previously employed TSI data by Coddington et al. (2016) have been retained as an alternative solar-related predictor.

SC6 Section 5 L 30 '… the increase in the ambient CO2 post 1850 is clearly correlated with the increased probability of drought… while during the pre-instrumental period such link does not manifest. The trend in CO2 in the industrial period is just one degree of freedom. Please recall the comments GC1b

**Response R3-16:** True, but note please that while we mentioned the existence of a correlation, we did not interpret it as a causal relation, only noted the possibility of one (please see also our Response R3-4).

Technical corrections, typing errors,etc:
TT1 Page 6, l 8: '…results…standardized regression coefficients…' In an simple regression these would be, by definition, correlation coefficients. How does this differ in this analysis from correlations? Some methodological details on the multiple regression approach taken is advisable. Does it account for covariability in the predictors? Etc…Please provide more explanation of the relevant aspects in the ms.

**Response R3-17:** Indeed, in simple (univariable) linear regression, standardized regression coefficient corresponds to correlation between predictor and predictand (and, in absolute value, to square root of the coefficient of determination). In multiple regression, however, no such straightforward relation exists for individual predictors. Standardization of the coefficients is used to make them more comparable mutually (among different predictors as well as among predictands), thought, admittedly, this representation does not directly convey information about the magnitude of the responses. In the revised version, the responses are therefore also shown in the form of predictor-specific time series generated for a selected regression configuration (Fig. S5 in the Supplement).

As for covariance of the predictors, its effect is reflected in the size of the confidence intervals for individual predictors (please see also our response R3-5).

TT2 Page 2, l 20: '…that increase…' substitute by '…that the increase…' This is just an example. I have found a few of those. I think the text is easy to understand in general. However, I would recommend it would be revised for editing/English

**Response R3-18:** Selected example corrected. The English of the manuscript was checked and corrected by a native speaker, Mr. Tony Long. The language correction was repeated in the revised text once again.

[revised manuscript text omitted]

---

## Referee Report (RR1)

Description

The paper addresses the identification and quantitative attribution of drought variability over the Czech lands in terms of three drought indices spanning over the last 5 centuries. It targets drought-climate links comparing drought indices with climate (temperature and precipitation) reconstructions, modes of internal variability and external forcing parameters.

As I stated in my first revision, I think the purpose of the paper has value and steps in the attribution of drought variability over the Czech Lands or progress in understanding mechanisms would be valuable in my view to support publication. The revised version of the manuscript includes new analysis like NAO related links and other aspects not considered in the initial version. It also considers more reconstructions of specific indices and provides a more clear view of the difficulties to relate, at least with this technique, the variability in external forcings and large scale climate drivers with Czech drought during the last few centuries. I think the authors have made an effort to provide a more clear manuscript and the discussion of their results is fair and honest in declaring the understanding that can derive from this analysis and its limitations. I think that there are still a number of issues that have to be taken care of. In general I would say that individually taken they are not major but there is a number of them. I will leave it to the consideration of the Editor whether a new revision cycle will be needed.

General comments.

GC 1.  Overall I would invite the authors to really think about what we gain from this analysis and have clear statements about the confidence and reliability of the links they report on, and report on this assessment in the conclusions as a take-home message for the reader. Some of the resulting coefficients and relationships stem from the (sometimes clearly, sometimes marginally clear) analysis of some of the reconstructions and not from the others. I think the authors do an honest job in highlighting this (and some of my subsequent comments go in this direction), although I would suggest really making an effort for a very clear assessment of how much confidence we can have on these results.

I think it is important to minimize the danger that results are cherry picked in the future using this manuscript as a reference for clear links between a mode of circulation (say PDO or AMO) and drought when in fact, it is not, and the relationship may be very much dependent on the reconstruction considered. I think this is particularly the case when the study provides coefficients resulting from a multiple regression analysis that considers data in different periods but there is no insight about the mechanisms that may support such relationships. Perhaps a message that needs to be clearly stated is that there is too much uncertainty and that even if one technique may provide relatively clear results in depicting some level of relationship (very small R2) between drought and large scale drivers and forcings, there is too much uncertainty for

having confidence on the purported relationships as other reconstructions do not provide that clear link.

GC2 Would the authors have arguments to believe that any specific reconstruction from the ones considered for ENSO, PDO, AMO or NAO is better or more reliable than the others? If so, this should be clearly stated. It is actually done so to some level. At a first stage the reconstructions are considered alike to derive relationships to regional drought and then the relationships are used to decide that one reconstruction is more trustworthy or reliable or better than the other because it shows a more intense relationship with drought. I think this is a dangerous path. Indeed the manuscript steps onto this ground and I would strongly suggest revising these types of arguments. After all I think that if this manuscript gets to publication, its value for the community should reside as much in the quality of the arguments as in the strength of the statistical results.

GC3 A great deal of the material of the manuscript at this stage refers to the Supplementary Material. Actually there are more plots in the SM than in the main document. Much of the discussion part relies on this bulk of material. I really wonder if some of the results should not be promoted to the main text.

Specific comments

SC1 Section 2.1.
Various indices to characterize drought at seasonal and annual timescales are used in the paper and introduced here. The last paragraph indicates that these indices are derived in Brázdil et al (2016a). The first paragraphs of Sec 2.1 indicate the differences in the predictor variables that lead to different definitions of SPI, SPEI and PDSI. I suggest that it may be good to include here some sentences of the different information that using these three indices instead of a single one can provide in the light not only of a priori definitions but also of the results in Brázdil et al (2016a) or on what may be expected to obtain later on. This information may be of interest for the reader to have some understanding of the interpretation of the indices for this specific area based on previous experience of the authors. Are the indices very different? Do any of the three reflect any specific features? Indeed the low frequency variability in fig 1 looks very similar for the three of them. I suggest providing the correlations between each pair of the three indices available. This will allow the reader for knowing two what extent having 3 series instead of one adds information. Otherwise the reader misses some, probably justified, rationale for this set up.

SC2 Section 2.2. Page 5, Line 15-17. '…extended back to AD 1501 using $CO_2$, $CH_4$… concentrations obtained…'
Who did this extension? If obtained by the authors indicate reference. If developed by the authors, please, explain how.
The reference to the web is perhaps better in the figure caption? A preferable alternative is always a reference.

SC3    Section 2.2. Page 5, Line 15-17
       Why wouldn't aerosols also be considered? This is an important
       anthropogenic component. The authors have discussed this in the
       response to the previous review. Please consider arguing here why
       including aerosols and LULC is not worth.

SC4    Figure 1
       o  Figure caption: suggest changing 'Fluctuations in the annual series
          of…' by 'Annual series of…'
       o  References: they are clearly exposed in the legends. I would rather
          suggest having them in the caption, but this is not critical. The
          Meinshausen one, extended… a note on this can better be incorporated
          to the caption, as any other feature related to the construction of the
          figure or source of data.
       o  Scale: the volcanic forcing should be negative. Units would be more
          clearly stated in the axis labels than in the legends.

SC5    Figure 2
       o  Figure caption: suggest changing 'Fluctuations in reconstructed series
          of…' by 'Reconstructed series of…'
       o  The Luterbacher et al (2002) series looks strangely flat. Can the authors
          please check on that one? Previous representations of this series show
          more low frequency variability. The resolution of the plotted series is not
          indicated and is confusing after reading the text (monthly, seasonal,
          annual… see MC7.
       o  Any technical details in the construction of the series added by the
          authors like filtering low frequency components by subtraction in the
          Mann et al series can, for the sake of clarity, be mentioned in the
          caption or a note to the main text be made.
       o  The reconstruction of Mann et al (2009) and MacDonald and Case
          (2005) seem to be in phase opposition. Maybe the authors should
          consider commenting on this in the main text as it can have implications
          for the subsequent analysis.

SC6    Figure 6
       o  In the logics of the text, this should rather be Figure 4.
       o  Please check on the Luterbacher NAO index wavelet (see SC5).
       o  Consider making a technical short note on the cone of influence in the
          caption. Also for the subsequent cross-wavelet plots.
       o  The numbers and labels in this figure are too small. Check size of
          characters also in subsequent plots.

SC7    Section 4.1, page 8.
       •  Line 24-25. 'A statistically significant solar related signal was also
          absent in all individual seasons except for SOM'
          Right, and also additionally a somewhat marginal link in JJA, however
          they are negative! I suggest being really careful with these things.
          Otherwise statistical links are highlighted but they may have little

physical basis. What can be the reason for a negative relationship of temperature with solar variability?

- Regarding ENSO, AMO and PDO. It would be good if some mechanistic explanation, linking to other literature, can be provided to support the confidence on these correlations. For instance the positive correlations with wetter DJF or drier SOM… It is desirable to provide some support for these relationships on the basis of mechanisms and/or similar relationships in other studies.

  The same applies for the wetter DJF and SOM AMO situations or the influence of the PDO to dry conditions. In this last one, why would PDSI be more sensitive according to the experience of the authors?

  Regarding the PDO, I would be interested in having some assessment by the authors on the confidence on these results, since a) the relation is found only with the Mann et al data, and b) this reconstruction seems to have a very different behavior to Macdonal and Case and Shen, sometimes in phase opposition.

  Finally, according to Fig 4 and 5, the influence of the PDO seems to be the largest. It is important to have some assessment on confidence on these results on the basis of previous literature and the results herein, as a) these results rely only on one reconstruction and b) the resulting coefficients are even larger than the NAO. Would the authors then support a larger influence of Pacific variability on Czech drought than that of the North Atlantic?

  When reading subsequent parts of the text this is not the case, but the numbers play in this direction at this stage and some comments on this may be advisable.

SC8    Fig. 4 and 5

$R^2$: The explained variances shown through Fig 4,5 seem to be very weak in general. This means the bulk of drought variability is not explained by these indices. Perhaps the fraction of low frequency variability explained is larger?

Perhaps it would be advisable to do the same exercise on purely instrumental indices (ENSO, PDO, AMO and NAO), not reconstructions and have that as a benchmark of what should be expected in the frame of the reconstructions. This should be viable in terms of assessing interanual variability in the instrumental period and would place a more realistic perspective on the level of expectations we can have on the reconstructions. After all, most of the variability the study is addressing is interannual to multi-decadal, well represented in the instrumental period.

SC9    Section 4.2

- In general I agree with the description of Section 4.2. I have reservations regarding talking about periodicities. Talking about periods or frequencies in a wavelet or spectrum is fine, but I would suggest avoiding conveying the message of stable periods/cycles. Otherwise, prediction based on cyclic memory would be possible. I would rather talk about timescales of variability. Having said that, I leave that to the criteria/taste of the authors.

- The sentence in page 11 ' No significant match between the oscillations in the NAO index series and the drought indices was found … (it is worthy of note that this result does not imply a lack of relationship as such, merely an abscense of common periodicities…'
I would disagree with this statement. If there is a relationship (linear) it must be appreciated in the covariability shown by crosswavelet or crosspectra. Perhaps I misunderstood the statement, but please, reconsider it, since this can be a very misleading one.

SC10 Section 5, page 12. Lines >2. 'Even so it should be emphasized that regression … does only reveal formal similarities… . This is particularly true in the case of signals dominated by simple trends, such as the gradual rise of GHG radiative forcing… Our results should be considered a supportive argument regarding the relationship between the drought regime and the anthropogenic forcing, not a definitive proof of the causal link.'

Page 17, line 7: 'GHGs concentration … matches the long-term trend component in the temperature sensitive drought indices quite well… Even considering that statistical attribution analysis can only reveal formal similarities… the relationship during pre-instrumental and instrumental periods and other available evidence… support the existence of an anthropogenic induced drying effect in central Europe…'

Please check the consistency of the level of reassurance of these statements with the results of the paper. The coefficients in Figs 4 and 5 somewhat support the role of GHGs, mostly in the industrial period for SPEI and PDSI and for temperature in the whole period. Seasonally, temperature is clearly positive and SPEI shows some negative response in JJA and SON. However: are temperature and SPEI and PDSI trends significant themselves? See Figure 1. Temperature trends seem to stand out of the background envelope of variability, but I would not be able to ascertain this is the case for SPEI and PDSI. Please think how to formulate attributing statistical relationships to trends that …may not be significant? First ascertain they are (detection) and then try going further.

SC11 Section 5, page 13. Lines >5. 'While previous studies… of explosive volcanism this analysis of more than five centuries of data has revealed a more distinct volcanic imprint suggesting a tendency to wetter conditions following major eruptions… most prominent in summer.'

Consider also page 8 line 31: 'The volcanism effect … precipitation is non significant… As a result the volcanism-attributed component is negligible in precipitation-only SPI, but somewhat more prominent (even still non significant) in temperature sensitive SPEI and PDSI. The season specific… during summer, when a borderline statistically significant response also appears for precipitation and both SPI and SPEI.

Page 17, line13: 'A distinct signature of temporarily wetter conditions following major … eruptions…was detected.'

These statements suggest different levels of reassurance of the relationship to volcanic activity. I think the 'distinct signature' statements overstate the relationships found with drought indices. In Fig. 4 none of the drought indices or the precipitation show coefficients that significantly stand out of 0. In Fig. 5 this is also the case except for summer when SPEI, SPI and precipitation tend to show values larger than 0… but can we call that a 'distinct signature'? Please evaluate the level of confidence on the relationships found and make sure the statements are really supported by the data and the relationships found. This applies also in general to other statements of the manuscript. See also next comment.

SC12     Regarding the volcanic response. The authors report that a lagged analysis bears no clear results. This is not strange in terms of covariance/correlations. A more appropriate analysis can be a simple epoch analysis in which the authors would synchronize the most important volcanic events in the last few centuries and the corresponding values of drought indices, temperature and precipitation. I think this would be a meaningful complementary plot to the ones shown here and would fit well to the discussion. In the summer it may show a clearer signal even.

SC13     Section 5. The discussion in pages 14 to 16 is well organized regarding the structure and the use of literature. I quite like that. There are however, two additional features that I find odd and would advise differently.

    a) One is the systematic use of supplementary material. A considerable bulk of supporting evidence relies on it. It can be a matter of style but having more figures in the SM than in the main text and these figures being so relevant for the interpretation of results suggests to me that some of these figures should be included in the main document.

    b) There is an issue with the interpretation and discussion of different reconstructions and how they provide or not evidence for variability of regional drought. One specific case is that of the AMO PDO indices and their differences. I understand this is somewhat an evolution of the PCA analysis in the first version of the manuscript. I would not say it is wrong, but as it is presented it reads like playing with numbers. The other reconstructions do not support that and all of a sudden the differences between two specific reconstructions of the same type (that have already been through a considerable filtering process) renders some correlations. It is hard to have confidence on these results and bear they are really representing some differences between Atlantic and Pacific variability. I think overstating those numbers is dangerous. This results permeates to the conclusions, with cautious phrasing, that is true… but I do not think there is good ground for it if it is not supported by some serious

rationale based on literature or mechanism based arguments. How can the authors provide some confidence that these are not numbers obtained just by chance?

SC14    Section 5. Page 15, Lines > 21. See also SC8. '… this role appears to be played by interannual variations associated with weather changes closer to synoptic time scales and tied to local climate…'
Still, it is strange that only NAO would not for instance account for a larger percentage of variability. And if there are other (European) local modes that account for more variability, shouldn't these actually the ones that should be considered then in this analysis?
I would advocate for having a benchmark of correlations with instrumental period indices that would then support to look at the indices selected in longer timescales.

Minor comments

MC1    Section 2.2, page 5, line 9: '…with notable oscillatory components in Fig 6.
This is the first time Fig 6 is mentioned. The second in Page 6, Line 8. The previous figure to be cited is Fig 3. I think that the logical sequence of the text asks for moving Fig. 6 to the 4$^{th}$ position. This would make a more logical flow in the text.

MC2    Section 2.2, page 5, line 9: '…with notable oscillatory components in Fig 6.'
Why 'with notable oscillatory components'?. Better indicate why wavelet spectra are used… for actually all series.

MC3    Section 2.2, page 5, line 18. 'Variations in solar activity typically  leave no clear imprint on the climatic conditions of the lower stratosphere'
Check consistency with detection/attribution chapter in IPCC 2013.

MC4    Section 2.2, page 5, line 24. 'The effects of major volcanic eruptions … but exhibiting  just inconclusive local imprints during the instrumental period'
Really?. To what area does this statement refer to? Please, check consistency with detection/attribution chapter in IPCC 2013.

MC5    Section 2.2, page 6, line 3-4. 'Since the primary focus... oscillatory behavior associated with internal climate variability…'
This may read a bit misleading because the focus of the study is also considering external forcings that influence drought. Perhaps would it be a better argument here that the external forcing signal is disregarded from the Mann et al series by subtracting the 70 yr moving average of the NH mean temperatures?

MC6    Section 2.2, page 6, line 3-4. 'Since the primary focus... oscillatory behavior associated with internal climate variability…'

Line 15-16. 'Again, due to the presence of a strong trend component in the Mann et al series, detrending…. '
Did the authors in this paper do this or was the detrended series obtained from elsewhere? If so, please include a reference.

MC7    Section 2.2, page 6, line 24-32. 'For the purposes of this study, it was also analyzed in the form of annual NAO index values, extended to the year 2006 by… Jones et al (1997)'
I found the last comment regarding Jones et al (1997) confusing, but maybe it was my misunderstanding. I suggest that the text includes clear statements on the strategy to address drought for different seasons/timescales in coordination/correspondence with those of the predictors used. I see that more clear statements are included in Section 3, page 7, lines ~10. I just suggest making this as clear as possible to the reader.

MC8    Section 4.1, page 8, line 15. '… or by total anthropogenic forcing including the effects of man-made aerosols'
Is this really so? Typically the effect of aerosols delays that of GHG because of their relative cooling. Thus a better correspondence between temperatures and anthropogenic forcing can be achieved when aerosols are considered. Check IPCC 2013 and perhaps rephrase argument.
In any case I understand the trends of drought are quite small and the effect would be difficult to discern between GHG and aerosols, as also the authors have commented in their response.

MC9    Section 5, page 12, line 31. '… it may be speculated that the responses in the seasonal data are tied to inter-annual…'
Wouldn't this be evident in the crosswavelet analysis?

---

## Author Response (AR2)

We would like to thank both referees for their valuable comments regarding our manuscript. We tried to incorporate their suggestions into the revised paper, or bring arguments in cases when we were unsure how the proposals could be implemented into the manuscript without too severe changes of its context or aim. In particular, there were several very relevant observations by reviewer #2 regarding interpretation and reliability of our results. We have tried to modify our presentation to address these; however, some of the suggested extensions of the analysis (such as incorporation of analysis more focused on the instrumental period) would result in substantial extensions our analysis, beyond its intended scope.

Please see below for individual comments of both referees, our responses and the corresponding changes to the manuscript.

**REVIEWER 1**

This is the revised version of a paper which I had reviewed earlier. The authors have done a thorough revision of the paper and have taken the main criticism into account. Their replies and iterations were done in a careful manner. Now the Discussion seems overly long - but I guess having asked for revisions, we now have to live with that. So I am happy with the revisions by the authors and have only very minor comments (e.g., Fisher -> Fischer, "borderline significant" -> perhaps better give p-value, conform -> confirm etc.; there are several more - please have another careful read).

> Thank you - we re-read the manuscript and made a few corrections/changes, hopefully improving the text. With regard to the specific suggestions above:
> - The spelling of 'Fischer' has been corrected.
> - Confidence level is now given when 'borderline significant' results are mentioned; we would however prefer to preserve the term itself, to underscore results that were just at the edge of statistical significance (unfortunately we do not have p-values themselves calculated from our significance tests).
> - The term 'conform' was used intentionally, not as a synonym/misspelling of 'confirm'

**REVIEWER 2**

**Description**
The paper addresses the identification and quantitative attribution of drought variability over the Czech lands in terms of three drought indices spanning over the last 5 centuries. It targets drought-climate links comparing drought indices with climate (temperature and precipitation) reconstructions, modes of internal variability and external forcing parameters.

As I stated in my first revision, I think the purpose of the paper has value and steps in the attribution of drought variability over the Czech Lands or progress in understanding mechanisms would be valuable in my view to support publication. The revised version of the manuscript includes new analysis like NAO related links and other aspects not considered in the initial version. It also considers more reconstructions of specific indices and provides a more clear view of the difficulties to relate, at least with this technique, the variability in external forcings and large scale climate drivers with Czech drought during the last few centuries. I think the authors have made an effort to provide a more clear manuscript and the discussion of their results is fair and honest in declaring the understanding that can derive from this analysis and its limitations. I think that there are still a number of issues that have to be taken care of. In general I would say that individually taken they are not major but there is a number of them. I will leave it to the consideration of the Editor whether a new revision cycle will be needed.

**General comments.**
GC 1. Overall I would invite the authors to really think about what we gain from this analysis and have clear statements about the confidence and reliability of the links they report on, and report on this assessment in the conclusions as a take-home message for the reader. Some of the resulting coefficients and relationships stem from the (sometimes clearly, sometimes marginally clear) analysis of some of the reconstructions and not from the others. I think the authors do an honest job in highlighting this (and some of my subsequent comments go in this direction), although I would suggest really making an effort for a very clear assessment of how much confidence we can have on these results.

I think it is important to minimize the danger that results are cherry picked in the future using this manuscript as a reference for clear links between a mode of circulation (say PDO or AMO) and drought when in fact, it is not, and the relationship may be very much dependent on the reconstruction considered. I think this is particularly the case when the study provides coefficients resulting from a multiple regression analysis that considers data in different periods but there is no insight about the mechanisms that may support such relationships. Perhaps a message that needs to be clearly stated is that there is too much uncertainty and that even if one technique may provide relatively clear results in depicting some level of relationship (very small R2) between drought and large scale drivers and forcings, there is too much uncertainty for having confidence on the purported relationships as other reconstructions do not provide that clear link.

> We agree that there are still questions not resolved by our analysis, as well as uncertainties and interpretational caveats that should be clearly communicated to the reader. While we have already tried to highlight them in the previous iteration of the

manuscript, we have now further extended and restated some formulations (especially) in the Results, Discussion and Conclusions sections, to emphasize the potential problems even more clearly, and to better match our interpretation to the results shown. This was done in a large part in response to individual comments and suggestions of the reviewer - please see the points below for specific details.

GC2 Would the authors have arguments to believe that any specific reconstruction from the ones considered for ENSO, PDO, AMO or NAO is better or more reliable than the others? If so, this should be clearly stated. It is actually done so to some level. At a first stage the reconstructions are considered alike to derive relationships to regional drought and then the relationships are used to decide that one reconstruction is more trustworthy or reliable or better than the other because it shows a more intense relationship with drought. I think this is a dangerous path. Indeed the manuscript steps onto this ground and I would strongly suggest revising these types of arguments. After all I think that if this manuscript gets to publication, its value for the community should reside as much in the quality of the arguments as in the strength of the statistical results.

The problem of reliability of individual reconstructions is certainly an important one, but one difficult to resolve. There are indeed substantial contrasts between our predictors: in particular, the three PDO-approximating signals are almost uncorrelated, sometimes actually slightly anti-correlated (with Mann et al. and MacDonald&Case series correlated at $r$ = -0.17). We do not dare to explicitly mark any of the reconstructions as inherently "more trustworthy or reliable", as such assessment would have to be backed by a deeper analysis of the source proxies and their processing. It would perhaps be possible to argue that a multi-proxy reconstruction (such as the one by Mann et al. 2009) might be superior in general reliability to reconstructions based on specific data from a particular region. Then again, perhaps individual reconstructions should be treated as reflections of different aspects of the systems studied (and therefore not completely mutually interchangeable, with each of them potentially suitable for a different purpose). Ultimately, to resolve the question which (if any) of the reconstructions should be considered superior would require a separate (and quite sizable) analysis of its own.

To communicate the above more clearly in our paper, we changed the formulation in the part of the Discussion dealing with PDO and differences between its individual representation (paragraph at p. 15, l. 12+). Please note that while we do present results for the 'best fitting' reconstruction (i.e. Mann et al. data for AMO/PDO) as a part of our core results, the results for the alternative versions of the predictors are also given (even though sometimes in a reduced form, as the amount of illustrations would have to be substantially increased otherwise). The differences and uncertainties are mentioned and discussed, especially in various parts of Discussion and Conclusions.

GC3 A great deal of the material of the manuscript at this stage refers to the Supplementary Material. Actually there are more plots in the SM than in the main document. Much of the discussion part relies on this bulk of material. I really wonder if some of the results should not be promoted to the main text.

Figs. S1 and S8 have now been moved to the main manuscript as Figs. 6 and 9, respectively. We would prefer to leave the rest of the extra materials in the Supplement, as they are either related to variables showing just limited influence in our analysis, or they are not essential to our key messages (and the main text has already grown rather considerably compared to our original concept).

**Specific comments**

SC1 Section 2.1.

Various indices to characterize drought at seasonal and annual timescales are used in the paper and introduced here. The last paragraph indicates that these indices are derived in Brázdil et al (2016a). The first paragraphs of Sec 2.1 indicate the differences in the predictor variables that lead to different definitions of SPI, SPEI and PDSI. I suggest that it may be good to include here some sentences of the different information that using these three indices instead of a single one can provide in the light not only of a priori definitions but also of the results in Brázdil et al (2016a) or on what may be expected to obtain later on. This information may be of interest for the reader to have some understanding of the interpretation of the indices for this specific area based on previous experience of the authors. Are the indices very different? Do any of the three reflect any specific features? Indeed the low frequency variability in fig 1 looks very similar for the three of them. I suggest providing the correlations between each pair of the three indices available. This will allow the reader for knowing two what extent having 3 series instead of one adds information. Otherwise the reader misses some, probably justified, rationale for this set up.

We have added the values of mutual correlations in the text, along with a brief a rationale for using three indices rather than a single one (end of Sect. 2.1):

"Note also that despite the profound similarity in temporal variations of individual drought indices, manifesting in their strong mutual correlation (Pearson correlation coefficient r = 0.93 for the annual values of SPI-SPEI, r = 0.83 for SPI-PDSI, r = 0.88 for SPEI-PDSI over the 1501-2006 period), each of them emphasizes different aspects of meteorological droughts, be they simple precipitation sums captured by SPI, evapotranspiration variations considered by SPEI, or longer-term soil moisture status embodied by PDSI. Hereinafter, results for multiple indices are therefore presented and discussed, with regard to their common features as well as mutual contrasts."

Regarding the details on differences in their definitions: please note that a brief version of this information is provided when introducing the drought indices in Sect. 2.2, along with links to more exhaustive sources. When relevant, contrasts between outcomes stemming from use of different indices are also highlighted in the Results and Discussion sections (for instance, when the behavior of SPI (a purely precipitation-based index) differs from the behavior of SPEI/PDSI (indices considering temperature in addition to precipitation)).

SC2 Section 2.2. Page 5, Line 15-17. '…extended back to AD 1501 using $CO_2$, $CH_4$… concentrations obtained…' Who did this extension? If obtained by the authors indicate

reference. If developed by the authors, please, explain how. The reference to the web is perhaps better in the figure caption? A preferable alternative is always a reference.

Approximate formulas from IPCC (2001) were used for the extension of the radiative forcing series; this is now mentioned in Sect. 2.2.

The web link points to an online data archive; we are not aware of a matching traditional reference.

SC3 Section 2.2. Page 5, Line 15-17
Why wouldn't aerosols also be considered? This is an important anthropogenic component. The authors have discussed this in the response to the previous review. Please consider arguing here why including aerosols and LULC is not worth.

Please note that we mention the effects of possible inclusion of (globally averaged) aerosol forcing at the end of the first paragraph of Sect. 4.1:

"It is also worthy of note that, due to very strong correlation between the respective time series, very similar results would have been obtained if the GHG forcing series was replaced with a predictor representing just $CO_2$-related effects, or by total anthropogenic forcing including the effects of man-made aerosols."

This explanation states that within the scope of our methodology, almost no change would result from using aerosol-inclusive anthropogenic forcing; we do not dispute their physical relevance, but since their inclusion would not change our results, we do not include discussion of possible aerosol effects either.

As for the LULC data, we do not use or mention them, primarily due to non-existence of a usable (homogeneous, representative, and five-centuries-long) reconstruction of a time series quantifying LULC changes.

SC4 Figure 1
o Figure caption: suggest changing 'Fluctuations in the annual series of…' by 'Annual series of…'

Changed.

o References: they are clearly exposed in the legends. I would rather suggest having them in the caption, but this is not critical. The Meinshausen one, extended… a note on this can better be incorporated to the caption, as any other feature related to the construction of the figure or source of data.

In this case, we would prefer to leave the references directly in the figures rather than in their captions, since names of the author(s) of the relevant publications serve as the only identifiers of different reconstructions of the explanatory variables, and they are used as such everywhere else in the manuscript. The method of extending the GHG forcing data is given in the respective part of Sect. 2.2 (second paragraph), we would prefer to not duplicate it in the figure caption.

o Scale: the volcanic forcing should be negative. Units would be more clearly stated in the axis labels than in the legends.

> Please note that the data in Fig. 2 (as well as the time series used in our analysis) represent stratospheric aerosol optical depth (and thus non-negative values), not forcing as such.
> As for the units placement, we would prefer to keep them as a part of the horizontal subtitles (it seems more consistent graphically, considering that the drought indices are dimensionless and unit is thus not given).

SC5 Figure 2
o Figure caption: suggest changing 'Fluctuations in reconstructed series of…' by 'Reconstructed series of…'

> Changed.

o The Luterbacher et al (2002) series looks strangely flat. Can the authors please check on that one? Previous representations of this series show more low frequency variability. The resolution of the plotted series is not indicated and is confusing after reading the text (monthly, seasonal, annual… see MC7.

> As far as we can tell, the series is consistent with its representation in the source paper by Luterbacher et al. (2002) – please see Fig. R1 below (DJF season, since only winter data are visualized in the original 2002 paper).

[Figure]

**Figure R1:** (top) DJF NAO index used in our analysis, derived from Luterbacher et al. (2002) data, (bottom) the original graph of the same series as presented by Luterbacher et al. (2002, Fig. 3).

o Any technical details in the construction of the series added by the authors like filtering low frequency components by subtraction in the Mann et al series can, for the sake of clarity, be mentioned in the caption or a note to the main text be made.

The detrending procedure is described in Sect. 2.2, when introducing the ENSO/AMO/PDO series; we would prefer to not repeat its description in the figure caption.

o The reconstruction of Mann et al (2009) and MacDonald and Case (2005) seem to be in phase opposition. Maybe the authors should consider commenting on this in the main text as it can have implications for the subsequent analysis.

Indeed, the correlation between the two series is negative, albeit relatively weak (-0.17 for the series of annual values), which is still rather underwhelming value for indices supposedly pertaining to the variability of the same climate oscillatory system. We have added a note of this fact to the Discussion, to complement the previous remarks pointing to the uncertainties stemming from contrasts between individual reconstructions. However, as also discussed for GC2/SC7, more detailed analysis of the reasons for these differences is beyond the scope of our paper, as it would require a detailed assessment of the data and methodology involved in preparation of each reconstruction.

SC6 Figure 6
o In the logics of the text, this should rather be Figure 4.

Our intention was to keep the wavelet and cross-wavelet illustrations close to each other, and close to the discussion of the wavelet-related results in Sect. 4.2. The reference to Fig. 6 in the Data section is only minor and we would therefore prefer to keep the wavelet charts as Fig. 6 (Fig. 7 in the current revised version).

o Please check on the Luterbacher NAO index wavelet (see SC5).

While the wavelet analysis is able to separate oscillations pertaining to different parts of the time-frequency space, their statistical significance is subject to the overall properties of the time series (in this case to the structure of the corresponding AR(1) model, upon which testing of statistical significance is based). For this reason, statistical significance of some portions of the wavelet spectra may be lower than it would be for a smoother signal, less dominated by high-frequency components.

o Consider making a technical short note on the cone of influence in the caption. Also for the subsequent cross-wavelet plots.

Done: Figs. 7 & 8 captions have been extended with: "The lower-contrast areas pertain to the cone of influence, i.e. region with diminished representativeness of the wavelet spectra due to edge effects. ".

o The numbers and labels in this figure are too small. Check size of characters also in subsequent plots.

The font for sub-titles and color scale has been enlarged in all figures containing wavelet and cross-wavelet spectra.

SC7 Section 4.1, page 8.
● Line 24-25. 'A statistically significant solar related signal was also absent in all individual seasons except for SOM' Right, and also additionally a somewhat marginal link in JJA, however they are negative! I suggest being really careful with these things. Otherwise statistical links are

highlighted but they may have little physical basis. What can be the reason for a negative relationship of temperature with solar variability?

> The problem here might lie with the 1501-1609 part of the Lean (2018) data, which is reconstructed in a different manner than the 1610+ part (for which sunspot data were employed, and which can thus arguably be considered more reliable). For this reason, we also included analysis results covering the 1610-2006 period only (Figs. 6d, 6e) – these show solar-related coefficients to be closer to zero and far from statistical significance, albeit still negative (which is likely just an effect of sample variance, not indication of actual cooling). We reordered the text in Sect. 4.1, 2$^{nd}$ paragraph, to make the reasoning for our claim of solar non-relevance more clear.

● Regarding ENSO, AMO and PDO. It would be good if some mechanistic explanation, linking to other literature, can be provided to support the confidence on these correlations. For instance the positive correlations with wetter DJF or drier SOM… It is desirable to provide some support for these relationships on the basis of mechanisms and/or similar relationships in other studies. The same applies for the wetter DJF and SOM AMO situations or the influence of the PDO to dry conditions. In this last one, why would PDSI be more sensitive according to the experience of the authors? Regarding the PDO, I would be interested in having some assessment by the authors on the confidence on these results, since a) the relation is found only with the Mann et al data, and b) this reconstruction seems to have a very different behavior to Macdonal and Case and Shen, sometimes in phase opposition.
Finally, according to Fig 4 and 5, the influence of the PDO seems to be the largest. It is important to have some assessment on confidence on these results on the basis of previous literature and the results herein, as a) these results rely only on one reconstruction and b) the resulting coefficients are even larger than the NAO. Would the authors then support a larger influence of Pacific variability on Czech drought than that of the North Atlantic? When reading subsequent parts of the text this is not the case, but the numbers play in this direction at this stage and some comments on this may be advisable.

> As also discussed in relation to GC2: While it would certainly be beneficial to expand the explanation of the mechanisms behind the formal links pointed out by our analysis, our options are limited. In the absence of a dataset covering our five-centuries-long analysis period and allowing for investigation of the circulation patterns (such as a reanalysis), it is difficult to ascertain/validate the nature of the teleconnections.  As for comparison with pre-existing studies, there seems to be a great deal of disparity between the character of the explanatory variables we used and the (largely observational) data employed in comparably focused studies (specifically, for PDO, the Mann et al. data provide long series, but virtually devoid of shorter-term variability, contrasting with the usually employed observational PDO indices, relatively rich in inter-annual (and sometimes even inter-monthly) variations). As a result, different time scales and processes likely play role in establishing the relevant teleconnections, making them difficult to compare between our study and the others.

To give a specific example: Baek et al. (2017; doi: 10.1175/JCLI-D-16-0766.1) performed a correlation analysis of PDSI with regard to various climate variability modes, including PDO. They identified a relatively large positive correlation between PDO index and PDSI in Europe – a result formally opposite to ours in terms of the correlation sign. On closer inspection, there appears to be an anticorrelation between Mann et al. PDO temperatures and the JISAO PDO index used by Baek et al. (http://research.jisao.washington.edu/pdo/PDO.latest.txt), valued at r = -0.37 for the JISAO index smoothed by 11-year moving average (note that this difference does not arise from our pre-processing of the Mann et al. data, although the anti-correlation becomes somewhat stronger after the detrending). This raises a question of definition and calibration of individual signals (including perhaps their sign when PCA is involved in the index definition) – an issue that would require a deep analysis of various definitions and/or reconstruction approaches for the explanatory variables, probably sizable enough to warrant a separate study. Only after this, results of the attribution-based studies could be meaningfully compared.

As for the weaker-than-expected influence of NAO, it can arguably be explained by annual averaging, at least partly - please see our response to SC14. (note also that NAO's influence in Czech precipitation – the key factor shaping all the drought indices – is smaller in the Czech Republic (CR) than it is in some other parts of Europe, due to CR's geographic position near the transition from positive to negative precipitation-to-NAO response.)

SC8 Fig. 4 and 5
$R^2$: The explained variances shown through Fig 4,5 seem to be very weak in general. This means the bulk of drought variability is not explained by these indices. Perhaps the fraction of low frequency variability explained is larger?
Perhaps it would be advisable to do the same exercise on purely instrumental indices (ENSO, PDO, AMO and NAO), not reconstructions and have that as a benchmark of what should be expected in the frame of the reconstructions. This should be viable in terms of assessing interanual variability in the instrumental period and would place a more realistic perspective on the level of expectations we can have on the reconstructions. After all, most of the variability the study is addressing is interannual to multi-decadal, well represented in the instrumental period.

Indeed, if only low-frequency variability was considered (e.g. after both the target and explanatory variables have been smoothed by a moving-average filter), the fraction of variance explained would be higher. However, such an approach would not allow to directly consider the faster-variable explanatory variables (episodic volcanism, 11-year cycle in the solar forcing or sub-decadal variations in the climate indices). Since our aim was not to construct a predictive model (and thus explain as much variability as possible), but rather to identify factors with potentially relevant links to our target variables, our interpretation is not centered around $R^2$ values, but is based on values and statistical significance of the regression coefficients.

SC9 Section 4.2
● In general I agree with the description of Section 4.2. I have reservations regarding talking about periodicities. Talking about periods or frequencies in a wavelet or spectrum is fine, but I would suggest avoiding conveying the message of stable periods/cycles. Otherwise, prediction based on cyclic memory would be possible. I would rather talk about timescales of variability. Having said that, I leave that to the criteria/taste of the authors.

The reviewer is correct to point out that stable periodicities would be an indication of simple, straightforward predictability, something our analysis does not support. However, note please that whenever a stable periodicity is mentioned in our text, the formulation actually points to its absence, not presence.

● The sentence in page 11 ' No significant match between the oscillations in the NAO index series and the drought indices was found … (it is worthy of note that this result does not imply a lack of relationship as such, merely an abscense of common periodicities…' I would disagree with this statement. If there is a relationship (linear) it must be appreciated in the covariability shown by crosswavelet or crosspectra. Perhaps I misunderstood the statement, but please, reconsider it, since this can be a very misleading one.

Please note that the lowest period detectable by the (cross-)wavelet analysis corresponds to twice the sampling time, i.e. 2 years (the Nyquist frequency). Much of the variability of the NAO index happens on year-to-year basis, and the high-frequency component can almost be considered white noise (lag-1-year autocorrelation of the NAO series is just 0.07). As a result, this variability (as well as the respective links to the drought indices/temperature/precipitation) is not captured in the cross-wavelet spectra, even in the presence of a clear linear relationship between the series. (please see also our response to MC9)

SC10
Section 5, page 12. Lines >2. 'Even so it should be emphasized that regression … does only reveal formal similarities… . This is particularly true in the case of signals dominated by simple trends, such as the gradual rise of GHG radiative forcing… Our results should be considered a supportive argument regarding the relationship between the drought regime and the anthropogenic forcing, not a definitive proof of the causal link.'

Page 17, line 7: 'GHGs concentration … matches the long-term trend component in the temperature sensitive drought indices quite well… Even considering that statistical attribution analysis can only reveal formal similarities… the relationship during pre-instrumental and instrumental periods and other available evidence… support the existence of an anthropogenic induced drying effect in central Europe…'

Please check the consistency of the level of reassurance of these statements with the results of the paper. The coefficients in Figs 4 and 5 somewhat support the role of GHGs, mostly in the industrial period for SPEI and PDSI and for temperature in the whole period. Seasonally,

temperature is clearly positive and SPEI shows some negative response in JJA and SON. However: are temperature and SPEI and PDSI trends significant themselves? See Figure 1. Temperature trends seem to stand out of the background envelope of variability, but I would not be able to ascertain this is the case for SPEI and PDSI. Please think how to formulate attributing statistical relationships to trends that …may not be significant? First ascertain they are (detection) and then try going further.

> In this case, our evaluation of the GHG-related components in the drought indices is based on the role of temperature in addition to the formal (non-)significance of individual coefficients themselves. In other words, a positive significant response in temperature is considered a justification for considering the negative response in temperature-sensitive indices (SPEI, PDSI) to be relevant and interpretable, even at $p > 0.05$. We modified the related formulations somewhat (especially in the Conclusions), to better convey our reasoning.

SC11
Section 5, page 13. Lines >5. 'While previous studies… of explosive volcanism this analysis of more than five centuries of data has revealed a more distinct volcanic imprint suggesting a tendency to wetter conditions following major eruptions… most prominent in summer.' Consider also page 8 line 31: 'The volcanism effect … precipitation is non significant… As a result the volcanism-attributed component is negligible in precipitation-only SPI, but somewhat more prominent (even still non significant) in temperature sensitive SPEI and PDSI. The season specific… during summer, when a borderline statistically significant response also appears for precipitation and both SPI and SPEI.
Page 17, line13: 'A distinct signature of temporarily wetter conditions following major … eruptions…was detected.'
These statements suggest different levels of reassurance of the relationship to volcanic activity. I think the 'distinct signature' statements overstate the relationships found with drought indices. In Fig. 4 none of the drought indices or the precipitation show coefficients that significantly stand out of 0. In Fig. 5 this is also the case except for summer when SPEI, SPI and precipitation tend to show values larger than 0… but can we call that a 'distinct signature'? Please evaluate the level of confidence on the relationships found and make sure the statements are really supported by the data and the relationships found. This applies also in general to other statements of the manuscript. See also next comment.

> Again, our interpretation reflects the relatively clear volcanic signal in temperature as a justification for considering the components in the temperature-sensitive drought indices (SPEI, PDSI) more physically meaningful, even when statistically non-significant. It is true, however, that the formulation regarding the drought indices may have been overly strong: in the revised version, 'a distinct signature' was changed to 'a signature', plus some other minor changes have been made to the related statements.

SC12

Regarding the volcanic response. The authors report that a lagged analysis bears no clear results. This is not strange in terms of covariance/correlations. A more appropriate analysis can be a simple epoch analysis in which the authors would synchronize the most important volcanic events in the last few centuries and the corresponding values of drought indices, temperature and precipitation. I think this would be a meaningful complementary plot to the ones shown here and would fit well to the discussion. In the summer it may show a clearer signal even.

> One problem with simple epoch analysis is the lack of information about intensity and interaction of individual eruptions, as they are treated as simple yes/no events. Our analysis primarily suggests a presence of volcanic effects for the temperature series and, in turn, for the temperature-sensitive SPEI and PDSI series (albeit not always in a statistically significant form). Since the analysis of European temperature data over a period quite similar to ours was already done by Fischer et al. (2007), and they reported results rather consistent with ours (summer temperature drop following major volcanic eruptions), we prefer to reference this study instead of implementing the epoch analysis ourselves.

SC13

Section 5. The discussion in pages 14 to 16 is well organized regarding the structure and the use of literature. I quite like that. There are however, two additional features that I find odd and would advise differently.

a) One is the systematic use of supplementary material. A considerable bulk of supporting evidence relies on it. It can be a matter of style but having more figures in the SM than in the main text and these figures being so relevant for the interpretation of results suggests to me that some of these figures should be included in the main document.

> As also discussed for GC3: Figs. S1 and S8 have now been moved to the main manuscript as Figs. 6 and 9, respectively.

b) There is an issue with the interpretation and discussion of different reconstructions and how they provide or not evidence for variability of regional drought. One specific case is that of the AMO PDO indices and their differences. I understand this is somewhat an evolution of the PCA analysis in the first version of the manuscript. I would not say it is wrong, but as it is presented it reads like playing with numbers. The other reconstructions do not support that and all of a sudden the differences between two specific reconstructions of the same type (that have already been through a considerable filtering process) renders some correlations. It is hard to have confidence on these results and bear they are really representing some differences between Atlantic and Pacific variability. I think overstating those numbers is dangerous. This results permeates to the conclusions, with cautious phrasing, that is true… but I do not think there is good ground for it if it is not supported by some serious rationale based on literature or mechanism based arguments. How can the authors provide some confidence that these are not numbers obtained just by chance?

Our inclusion of the common AMO/PDO component and of their difference was motivated by the strong collinearity of the PDO a AMO signals (which, by the way, was even stronger before the respective series were detrended, due to their shared long-term component – see Fig. 1 in Mann et al. 2009). Because of this strong correlation between the AMO and PDO series, linear regression cannot reliably distinguish between their influences; using PCA (or, in this case, its primitive 'mean vs. difference' interpretation) allows for a better defined setup of regressors. Since we found it interesting that the AMO-PDO difference constitutes a more influential predictor, despite it being responsible for just a small fraction of the combined variance in the original AMO/PDO pair, we mentioned this outcome. Note, please, that we avoided an explicit interpretation of this results as a proof an actual link between AMO-PDO contrast and our target variables and stressed a need for future validation (Conclusions, point (iii)).  We have now also slightly modified the respective sections of the text to make this even more clear.

SC14

Section 5. Page 15, Lines > 21. See also SC8. '… this role appears to be played by interannual variations associated with weather changes closer to synoptic time scales and tied to local climate…'

Still, it is strange that only NAO would not for instance account for a larger percentage of variability. And if there are other (European) local modes that account for more variability, shouldn't these actually the ones that should be considered then in this analysis?

I would advocate for having a benchmark of correlations with instrumental period indices that would then support to look at the indices selected in longer timescales.

The seemingly less-than-expected dominance of NAO likely stems from several factors. First, while there are profound inter-annual variations in the NAO index series, a substantial part of its variability (and thus influence on local climate) takes place at sub-annual time scales; the annual averaging therefore somewhat diminishes NAO's relative prominence. Furthermore, annual averaging also aggregates strong NAO influence in winter with its weaker effects in other seasons. Note, please, that the NAO influence is more pronounced in the DJF season (Fig. 5a) in the temperature series (due to tighter temporal focus, as well as winter conditions being more affected by NAO than the rest of the year). (You may also notice almost non-existent effect of NAO on SPEI in winter, despite its links to both temperature and precipitation – this seems to be due to substantial positive correlation between central European winter temperature and precipitation, the influences of which effectively cancel-out each other in the SPEI series.)

Regarding other potentially influential variability modes (such as Eastern Atlantic / Western Russia Pattern or local central European circulation indices): while they are certainly something to be considered in a analysis focusing on recent climate variability, their usability for our study is constrained by the lack of reconstructions pre-dating the instrumental era.

As for extending our analysis to the instrumental-era inputs: This would shift our intended focus substantially, and would require introduction of multiple additional datasets, as well as major additions to the methodology applied (e.g., adding circulation

patterns analysis, unfeasible and thus unused for our five-centuries-long-period, but possible and desirable for recent records). While we would prefer to not do this in the present study, we are currently preparing a follow-up analysis specifically targeting the possible links and teleconnection for just the 20$^{th}$ and 21$^{st}$ centuries, in a wider geographical and physical perspective.

**Minor comments**

MC1 Section 2.2, page 5, line 9: '…with notable oscillatory components in Fig 6. This is the first time Fig 6 is mentioned. The second in Page 6, Line 8. The previous figure to be cited is Fig 3. I think that the logical sequence of the text asks for moving Fig. 6 to the 4th position. This would make a more logical flow in the text.

As also discussed with regard to comment SC6: Numbering of the figures was chosen to keep illustrations with wavelet and cross-wavelet spectra close to each other and to the sections discussing them.

MC2 Section 2.2, page 5, line 9: '…with notable oscillatory components in Fig 6.' Why 'with notable oscillatory components'?. Better indicate why wavelet spectra are used… for actually all series.

This formulation is merely meant to emphasize that wavelet analysis was only applied to signals with oscillatory behavior consistent with variability in the wavelets employed (Morlet), i.e. not to trend-dominated series (anthropogenic forcing) or to series reflecting aperiodic time-asymmetric episodic events (volcanic activity).

MC3 Section 2.2, page 5, line 18. 'Variations in solar activity typically leave no clear imprint on the climatic conditions of the lower stratosphere' Check consistency with detection/attribution chapter in IPCC 2013.

The sentence was meant to convey the lack of distinct solar-related components reported by most comparable previous studies involving time series analysis (though it may also be mentioned that even the solar signal reported by the IPCC (2013) report is quite weak, and the respective aggregate confidence interval starts at zero for solar-induced temperature change since the pre-industrial period). Formulation in our manuscript has been revised to: "While variations in solar activity typically leave no or just weak imprint in lower tropospheric observational time series during the instrumental era..."

MC4 Section 2.2, page 5, line 24. 'The effects of major volcanic eruptions … but exhibiting just inconclusive local imprints during the instrumental period'
Really?. To what area does this statement refer to? Please, check consistency with detection/attribution chapter in IPCC 2013.

Similarly to our statement regarding the solar activity above, this formulation was meant to emphasize difficulty of reliably extracting the imprint of volcanic activity from the

(relatively short) time-series covering the instrumental period. The sentence has now been slightly changed to: "... but exhibiting just largely inconclusive local imprints in local observational temperatures during the instrumental period ... ".

MC5 Section 2.2, page 6, line 3-4. 'Since the primary focus... oscillatory behavior associated with internal climate variability…' This may read a bit misleading because the focus of the study is also considering external forcings that influence drought. Perhaps would it be a better argument here that the external forcing signal is disregarded from the Mann et al series by subtracting the 70 yr moving average of the NH mean temperatures?

Thank you for the suggestion – the respective sentence has been replaced with: "To limit the presence of long-term trends in the Mann et al. data, largely reflecting external forcing rather than manifestation of internal climate dynamics, the series has been detrended by subtracting the 70-year moving average of the northern hemisphere mean temperature, also provided by Mann et al. (2009); ... "

MC6 Section 2.2, page 6, line 3-4. 'Since the primary focus... oscillatory behavior associated with internal climate variability…' Line 15-16. 'Again, due to the presence of a strong trend component in the Mann et al series, detrending…. ' Did the authors in this paper do this or was the detrended series obtained from elsewhere? If so, please include a reference.

This is something we did ourselves – the procedure (i.e., a simple subtraction of smoothed northern hemispheric temperature) is described in the fifth and sixth paragraphs of Sect. 2.2, when introducing the series approximating ENSO, AMO and PDO.

MC7 Section 2.2, page 6, line 24-32. 'For the purposes of this study, it was also analyzed in the form of annual NAO index values, extended to the year 2006 by… Jones et al (1997)'
I found the last comment regarding Jones et al (1997) confusing, but maybe it was my misunderstanding. I suggest that the text includes clear statements on the strategy to address drought for different seasons/timescales in coordination/correspondence with those of the predictors used. I see that more clear statements are included in Section 3, page 7, lines ~10. I just suggest making this as clear as possible to the reader.

The comment referring to the Jones et al. (1997) NAO index merely explains that the Luterbacher et al. (2002) NAO reconstruction, available until 2001, was extended by the observational Jones et al. data for the period 2002-2006 (so that full 1501-2006 period could be used for multiple linear regression). As the reviewer states, specification of the (sub)periods used in our analysis is then given in Sect. 3, as a part of the methodology overview.

MC8 Section 4.1, page 8, line 15. '… or by total anthropogenic forcing including the effects of man-made aerosols' Is this really so? Typically the effect of aerosols delays that of GHG because of their relative cooling. Thus a better correspondence between temperatures and

anthropogenic forcing can be achieved when aerosols are considered. Check IPCC 2013 and perhaps rephrase argument.

In any case I understand the trends of drought are quite small and the effect would be difficult to discern between GHG and aerosols, as also the authors have commented in their response.

> Indeed, from a physical point of view, the effects of aerosols are essential (albeit still quite uncertain regarding their magnitude). However, from the perspective of linear regression, the standardized regression coefficients (presented in our analysis) would be almost identical for two versions of a given predictor strongly resembling each other in shape. Specifically, the GHG-only and GHG+aerosols radiative forcing series are very strongly correlated (their Pearson correlation is 0.995 for the annual Meinshausen et al. (2011) data over the 1765-2010 period). Use of a forcing series involving aerosols would therefore produce results very similar to the those presented in our manuscript – this is mentioned in the first paragraph of Sect. 4.1.

MC9 Section 5, page 12, line 31. '… it may be speculated that the responses in the seasonal data are tied to inter-annual…'
Wouldn't this be evident in the crosswavelet analysis?

> Not necessarily, since the links may not pertain to a specific frequency or frequency band, strong enough in both signals to produce a statistically significant response in the cross-wavelet spectrum (a regression mapping may be more sensitive in this regard, as it does not consider individual frequencies separately and does not rely on a specific shape of the wavelet). (To be honest, we did not actually examine the cross-wavelets for season-specific series, partly because of the higher noise levels compared to the annual versions, and partly because of the high amount of possible combinations.)

[revised manuscript text omitted]

---

## Author Response (AR3)

dear authors

**I now got review comments back on your revised version. One of the referee is happy how you have addressed, the 2nd one is not yet satisfied as a couple of points from the first review have not been adequately addressed. Please work on those, shorten the paper significantly and address the open issues. I have noted minor, but in fact I think you need to put more effort than minor changes but did not want to put major revisions. Please check again also the queries of the 3rd reviewer carefully.**

**with my best wishes**

**Jürg**

Greetings,

and thank both you and the referee for your suggestions. The revised text has now been reduced by about 500 words, by removing several passages in the Data, Methods and Discussion sections, as well as by implementing some smaller modifications to various formulations. However, please note that manuscript's total size results from several additions and extensions, largely reflecting the comments and recommendations of the referees during the previous review rounds. These included use and discussion of additional predictors (particularly NAO, which was not directly considered in the original manuscript but was included upon remarks/requests from referees during the first round of reviews), employment of alternative reconstructions for most predictors (also included after the first round of review) as well as several supplementary analyses. While we tried to limit the revisional size bloat (e.g. by placing most of the additional materials in the Supplement rather than in the main text), the resulting text has grown quite sizable due to encompassing a substantial number of individual elements, none of which is arguably too large on its own, but which are many in number. Additional reductions would therefore require elimination of actual results or of the key elements of their discussion – something we would prefer to avoid.

Please see below for our responses to the comments by Referee #1.

Best regards
On behalf of the authors
Jiri Miksovsky

**REFEREE #1:**

**This is the revised version of a paper I have reviewed before. The paper is for a special issue on "Droughts over centuries: what can documentary evidence tell us about drought variability, severity and human responses?", and I this fits of course very well into the topic of thew Special Issue and the Journal. Overall I am not extremely excited about the paper, but I think it is worth publishing.**

**As said in my previous review, I find the paper unnecessarily long, and it has gotten even a bit longer, not shorter. The results and discussion section fail to focus, separate the important from the less important findings and guide the reader through the text.**

In shortening of the manuscript (by about 500 words), we removed several passages meant to provide context and additional details to the methodological or interpretational aspects of our analysis, but not essential to the core presentation. We would, however, prefer to not remove the results themselves; note also that most of the size inflation is due to accommodating the requests of the referees in the previous review rounds.

**Some relatively well known aspects (NAO) could be shortened.**

The sections devoted to NAO are relatively brief and additional shortening would mean removing some of the actual results (please note that we did not include NAO in the original version of the manuscript – it was added after the first round of reviews upon the suggestion of the referees, especially Referee #3).

**I am a bit surprised that p-values cannot be stated. How do you know, then, whether something is "borderline significant" or not?**

The statements about significance of the responses are based on the confidence intervals (CIs) for the regression coefficients, presented for the 95% and 99% levels in Figs. 4, 5, 6 and 9. This way, not only can reader assess statistical significance of the coefficients (by looking at the relative position of the CIs to the zero line), but additional information about symmetry of the CIs or the effects of (multi)collinearity can be seen in the visualizations presented.

**Also, something which goes back to my first review, but I still would like to bring it up in the context of shortening: the Mann et al. NINO3 lacks interannual variability and the correlation between the PDO indices is even negative. Stating this is important, but then no long discussion of the results is necessary.**

As for the ENSO variability based on Mann et al. data: As stated in our responses in the prior revision rounds, this series was only used as an alternative reconstruction, while data by Li et al. served as our primary ENSO predictor (as such, they are used in Figs. 4 and 5, summarizing regression outcomes). While we would prefer to not remove the Mann-et-al-based results altogether (since they serve as an approximation of longer-term variability of the ENSO system), changes were made in a few places of the text to further highlight the fact that the Mann et al. ENSO data only carry inter-decadal variability, and thus information substantially different from the Li et al. ENSO reconstruction.

Regarding PDO indices: It would be rather inadequate to mention the differences between the indices, but not to attempt to discuss their implications (note, please, that this discussion has been included to accommodate a suggestion of Referee #2 in the 2nd review round, and we tried to keep it rather brief).

[revised manuscript text omitted]

---

## Author Response (AR4)

**dear authors**

**thanks very much for addressing the reviewers comments.**
**I have only three small points I wish you can implement/change**
**In the section of ENSO and the role for climate in Europe, can I please ask you to include and cite also Brönnimann et al. (2007).**
**Brönnimann, S., Xoplaki, E., Casty, C., Pauling, A. and Luterbacher, J., 2007: ENSO influence on Europe during the last centuries, Clim. Dyn., 28, 181-197.**

**Concerning ENSO indices, please also cite the new paper by Dätwyler et al. 2019**
**Dätwyler, C., Abram, N.J., Grosjean, M., Wahl, E.R., . & Neukom, R. (2019). ENSO variability, teleconnection changes and responses to large volcanic eruptions since AD 1000. International Journal of Climatology, https://doi.org/10.1002/joc.5983**

**Luterbacher et al. (2002) should be changed to 2001**

**with my best wishes and thanks very much for those small points to be implemented. Afterwards I am happy to accept the paper**

**Jürg**

Greetings,

and thank you very much for your suggestions, which have now been incorporated into the manuscript:

- the effects of ENSO on European climate, reported by Brönnimann et al. (2007), are now mentioned in the *Data* section as well as in the *Discussion*
- the paper by Dätwyler et al. (2019) is now referenced in the *Data* section as another source of information on ENSO teleconnections (though we did not explicitly mention/discuss their ENSO reconstruction, since we do not work with it in our analysis)
- the reference Luterbacher et al. (2002) has been changed to Luterbacher et al. (2001) in the text as well as in the figures (the original dating was based on the year given in the PDF version of the paper on the publisher's site, as well as in the source of the respective time series (CRU online database)).

Aside from the above changes, the manuscript has been extended with the *Data availability*, *Author contributions* and *Competing interests* sections, as per the guide on the journal's page; the format of DOIs in the list of references has been modified to full http form.

Best regards
On behalf of the authors
Jiri Miksovsky

[revised manuscript text omitted]